# Multi-Reward Best Policy Identification

**Alessio Russo**
Ericsson AB
Stockholm, Sweden

**Filippo Vannella**
Ericsson Research
Stockholm, Sweden

## Abstract

Rewards are a critical aspect of formulating Reinforcement Learning (RL) problems; often, one may be interested in testing multiple reward functions, or the problem may naturally involve multiple rewards. In this study, we investigate the *Multi-Reward Best Policy Identification* (MR-BPI) problem, where the goal is to determine the best policy for all rewards in a given set $\mathcal{R}$ with minimal sample complexity and a prescribed confidence level. We derive a fundamental instance-specific lower bound on the sample complexity required by any Probably Correct (PC) algorithm in this setting. This bound guides the design of an optimal exploration policy attaining minimal sample complexity. However, this lower bound involves solving a hard non-convex optimization problem. We address this challenge by devising a convex approximation, enabling the design of sample-efficient algorithms. We propose `MR-NaS`, a PC algorithm with competitive performance on hard-exploration tabular environments. Extending this approach to Deep RL (DRL), we also introduce `DBMR-BPI`, an efficient algorithm for model-free exploration in multi-reward settings.

## 1 Introduction

Reinforcement Learning (RL) provides a powerful framework for sequential decision-making, which has achieved a number of breakthroughs in a wide range of applications, including mastering video games [1, 2], accelerating algorithm design [3], optimizing fusion control systems [4, 5], and enhancing autonomous robotic control [6, 7, 8].

Rewards are a critical aspect of formulating an RL problem. Yet, designing a reward function can require numerous iterations of reward engineering [9]. Furthermore, in many practical scenarios, the reward signal is not necessarily a scalar, and the agent may seek to optimize performance across a set of reward functions [10, 11] (e.g., goal reaching applications in robotics [12], recommender systems [13], intent-based radio network optimization tasks [14, 15], etc.).

In this work, we investigate how to efficiently identify optimal policies across multiple rewards in a given set $\mathcal{R}$, a setting we refer to as *Multi-Reward Best Policy Identification* (MR-BPI). This setting is a variant of traditional Best Policy Identification (BPI) with fixed confidence [16, 17, 18, 19, 20], where the goal is to identify optimal policies using the least number of samples and with a prescribed confidence level. Exploration algorithms for BPI typically aim to find the optimal policy for a single reward function, which is known and fixed in advance. However, as previously mentioned, the agent may face uncertainty about which reward function is the most appropriate for a given task, or may seek to optimize performance across a set of rewards.

This problem remains relatively unexplored in the current RL literature. Closely related settings are reward-free RL [21, 22, 23], unsupervised RL [24, 25, 26, 27, 28, 29], and multi-objective RL [30, 31]. In reward-free RL, the goal of the agent is to estimate the optimal policy for any possible

---

Code repository: https://github.com/rssalessio/Multi-Reward-Best-Policy-Identification

38th Conference on Neural Information Processing Systems (NeurIPS 2024).

reward, using methods such as maximum entropy exploration [32, 33]. This reward-free phase is then followed by a downstream task adaptation in unsupervised RL. Lastly, in multi-objective RL, the agent optimizes over multiple (potentially conflicting) rewards, with the goal of identifying a Pareto frontier of optimal policies. While reward-free RL is the closest to our setting, exploration for any possible reward is unlikely to occur in practical problems. On the other hand, unsupervised RL and multi-objective RL do not directly address the multi-reward pure exploration problem — a setting we deem to be of practical use in a wide range of applications. For instance, in intent-based radio network optimization, rewards are designed to reflect network operator intents, such as maximizing throughput or coverage in a given network area. MR-BPI enables the agent to identify policies that perform well across different reward functions, enhancing robustness and adaptability to varying conditions and goals.

Our contributions are as follows. For MR-BPI with a finite set of user-specified rewards, we establish an asymptotic instance-specific sample complexity lower bound for Markov Decision Processes (MDPs) with a unique optimal policy, which generalizes the lower bound in the single-reward BPI setting [17]. This bound provides insights into the fundamental difficulty of the problem and guides the design of an optimal exploration policy. As in previous works [16, 17, 19, 34], determining such an exploration policy requires to solve a non-convex optimization problem. To address this issue, we propose a convex formulation of the lower bound based on a finite set of rewards. We also propose extensions for two cases: (1) for random exogenous reward signals provided by the environment (app. B.4), and (2) for a set of continuous rewards, which can be addressed by considering a carefully chosen finite set of rewards (app. B.3).

We then devise `MR-NaS` (Multi-Reward Navigate and Stop), a PC algorithm with asymptotic sample complexity guarantees, adaptable to the aforementioned extensions. We evaluate the performance of `MR-NaS` on different hard-exploration tabular environments, comparing to `RF-UCRL` [22] (a reward-free exploration method), `ID3AL` [33] (a maximum entropy exploration approach) and `MR-PSRL`, a multi-reward adaptation of PSRL [35]. Results demonstrate the efficiency of `MR-NaS` in identifying optimal policies across various rewards and in generalizing to unseen rewards when the reward set is sufficiently diverse. Lastly, we further extend our multi-reward exploration approach to Deep RL (DRL) by devising `DBMR-BPI` (Deep Bootstrapped Multi-Reward BPI), a model-free exploration algorithm for continuous-state MDPs. We assess its performance, along with `RND` [26] (Random Network Distillation), `Disagreement` [27], and `APT` [29], on hard-exploration problems from the DeepMind BSuite [36] and on link adaptation (a radio network control problem [37]).

## 2 Problem setting

In this section, we introduce the notation used in the manuscript and briefly introduce the Best Policy Identification (BPI) framework. Lastly, we describe the MR-BPI setting.

### 2.1 Markov Decision Processes and the Set of Rewards

**Markov Decision Processes (MDPs).** We consider discounted MDPs of the type $M = (\mathcal{S}, \mathcal{A}, P, \gamma)$, where $\mathcal{S}$ is the finite state space (of size $S = |\mathcal{S}|$), $\mathcal{A}$ is the finite action space (of size $A = |\mathcal{A}|$), $P : \mathcal{S} \times \mathcal{A} \to \Delta(\mathcal{S})$ is the transition function, which maps state-action pairs to a distribution over states and $\gamma \in (0, 1)$ is the discount factor. We indicate by by $M_r = (\mathcal{S}, \mathcal{A}, P, r, \gamma)$ the MDP when considering a reward $r : \mathcal{S} \times \mathcal{A} \to [0, 1]$. For a given reward $r$, we define the discounted value of a Markov policy $\pi : S \to \Delta(\mathcal{A})$ to be $V_r^\pi(s) = \mathbb{E}^\pi[\sum_{t \geq 0} \gamma^t r(s_t, a_t)|s_0 = s]$, where $s_{t+1} \sim P(\cdot|s_t, a_t)$ and $a_t \sim \pi(\cdot|s_t)$. We also let $V_r^\star(s) = \max_\pi V_r^\pi(s)$ be the optimal value in $s$ and, for MDPs with a unique optimal policy, we indicate it by $\pi_r^\star(s) = \arg\max_\pi V_r^\pi(s)$. Similarly, we also define the action-value function $Q_r^\pi(s, a) = r(s, a) + \gamma\mathbb{E}_{s' \sim P(\cdot|s,a)}[V_r^\pi(s')]$. For an MDP $M_r$, we indicate by $\Pi^\star(M_r) = \{\pi \in \Pi^d : \|V_r^\star - V_r^\pi\|_\infty = 0\}$ the set of optimal policies in $M_r$ and $\Pi^d$ is the set of deterministic policies.

**Set of rewards.** We denote the set of rewards by $\mathcal{R}$, and assume that it is known to the agent beforehand. In subsequent sections, for the sake of simplicity, we consider this set to be discrete and finite, while we present results for continuous sets of rewards in app. B.4. For finite MDPs, rewards are represented as vectors within $[0, 1]^{SA}$, with each coordinate $i$ corresponding to the reward for the $i$-th state-action pair (thus the reward in each state-action pair is bounded in $[0, 1]$). We also denote

the canonical basis of rewards as $\mathcal{R}_{\text{canonical}} = \{e_1, \ldots, e_{SA}\}$ in $\mathbb{R}^{SA}$ (with $e_i$ being the $i$-th vector of the canonical basis).

## 2.2 Single-Reward Best Policy Identification with Fixed Confidence

Best Policy Identification consists of finding an optimal policy in an MDP using the least number of samples with a given confidence level for a specific reward function [16, 17], and the proof techniques are inspired by the Best Arm Identification literature in the bandit setting [20].

**Optimal exploration strategy.** The main idea is to derive an *instance-specific* lower bound on the expected number of samples needed to identify an optimal policy. Interestingly, the lower bound is often formulated as an optimization problem, whose solution $\omega^\star$ allows us to derive an optimal exploration algorithm. In other words, the solution $\omega^\star$ provides the best proportion of static draws. That is, a policy following $\omega^\star$ matches the lower bound and is therefore optimal [20, 17].

**Change of measure.** The recipe to derive instance-specific sample complexity lower bounds is based on a *change of measure* argument. In general, the reasoning adopted is to think in an adversarial way: *Does there exist an MDP similar to the one considered for which the optimal policy is different?*

The analysis of such a problem delves into the required minimum amount of *evidence* needed to discern between different hypotheses. The evidence is quantified by the log-likelihood ratio of the observations under the true model and a *confusing model*. This confusing model is usually the one that is *statistically* closest to the true model while admitting a different optimal policy.

This argument allows us to derive an instance-dependent quantity $T^\star$, also known as the *characteristic time*, that lower bounds the sample complexity of an algorithm. Change of measure arguments have a long history [38, 39, 40, 41], and have been applied to derive lower bounds for regret minimization [42, 43, 44], best-arm identification [20, 45, 46, 47] and change detection problems [48, 49, 50].

## 2.3 Multi-Reward Best Policy Identification with Fixed Confidence

The MR-BPI setting follows closely the single-reward BPI framework [17]. In MR-BPI, the goal is to learn the best policy for all possible rewards $r \in \mathcal{R}$ as quickly as possible. This objective can be formalized in the $\delta$-Probably-Correct ($\delta$-PC) framework [51, 22], where the learner interacts with the MDP until she can output an optimal policy with confidence $1 - \delta$. Contrarily to the classical BPI framework, that solely considers a single reward, here we are interested in algorithms that identify a best policy for all rewards in $\mathcal{R}$.

**Algorithm.** Formally, an algorithm consists of (i) an exploration strategy (a.k.a. sampling rule), (ii) a stopping rule $\tau$ and (iii) an estimated optimal policy $\hat{\pi}_{\tau,r}$, for any reward $r \in \mathcal{R}$. At each time step $t$, the algorithm observes a state $s_t$, it selects an action $a_t$, and then observes the next state $s_{t+1} \sim P(\cdot|s_t, a_t)$. We denote by $\mathbb{P}$ (resp. $\mathbb{E}$) the probability law (resp. expectation) of the process $(Z_t)_t$, where $Z_t = (s_t, a_t)$. We indicate by $\mathcal{F}_t = \sigma(\{Z_0, \ldots, Z_t\})$ the $\sigma$-algebra generated by the random observations made under the algorithm up to time $t$ in the *exploration* phase. For such algorithm, we define $\tau$ to be a stopping rule w.r.t. the filtration $(\mathcal{F}_t)_{t \geq 1}$. This stopping rule $\tau$ simply states when to stop the exploration phase. At the stopping time $\tau$ the algorithm outputs an estimate $\hat{\pi}_{\tau,r}$ of the best policy for any given reward $r \in \mathcal{R}$ using the collected data. We focus on $\delta$-PC algorithms, according to the following definition, and we impose a few mild assumptions.

**Definition 2.1** ($\delta$-PC Algorithm). We say that an algorithm is $\delta$-PC if, for any MDP $M$, it outputs (in finite time) an optimal policy for any reward $r \in \mathcal{R}$ with probability at-least $1 - \delta$, i.e., $\mathbb{P}[\tau < \infty, (\forall r \in \mathcal{R}, \hat{\pi}_{\tau,r} \in \Pi^\star(M_r))] \geq 1 - \delta$.

**Assumption 2.1.** (I) $M \in \mathcal{M}$, where $\mathcal{M}$ is the set of communicating MDPs [52] with a unique optimal policy for every $r \in \mathcal{R}$ ; (II) $M$ is aperiodic under a policy that assigns a positive probability to all actions in every state; (III) for all $r \in \mathcal{R}$ there exists a model $P'$ such that $P(\cdot|s, a) \ll P'(\cdot|s, a)$ for all $(s, a)$, and $\Pi^\star(M_r) \cap \Pi^\star(M'_r) = \emptyset$, where $M'_r = (\mathcal{S}, \mathcal{A}, P', r, \gamma)$ and for any two measures $(\mu, \lambda)$, $\mu \ll \lambda$ denotes absolute continuity of $\mu$ w.r.t. $\lambda$.

Assumptions (I)-(II) are standard in the BPI literature [51, 53]. Assumption (III) is made to avoid degenerate cases in which there are no confusing model for rewards $r \in \mathcal{R}$. For example, this occurs when the reward is identical in all state-action pairs (therefore all actions are optimal).

# 3 Best Policy Identification with a Set of Rewards

In this section, we investigate the MR-BPI problem, *i.e.*, how to optimally explore an MDP to derive the best policy for any reward in $\mathcal{R}$. In the first subsection, we derive a sample complexity lower bound for any $\delta$-PC algorithm. Later, in the second subsection, we present `MR-NaS`, a $\delta$-PC algorithm inspired by `NaS` [17] with asymptotic optimality guarantees.

## 3.1 Sample Complexity Lower Bound

In the next theorem, we state an instance-specific lower bound on the sample complexity of any $\delta$-PC algorithm. This bound provides a fundamental limit on the performance of any $\delta$-PC algorithm and guides the design of optimal exploration methods for MR-BPI.

To state the bound, we indicate by $\omega \in \mathbb{R}^{SA}$ the *allocation* (or exploration policy), and by $\omega(s,a)$ the corresponding frequency of visit of a state-action pair $(s,a)$; we introduce the set of *confusing* models $\mathrm{Alt}_r(M)$ for $M_r$ as the set of MDPs that are statistically similar to the original model $M_r$, but admit a different set of optimal policies under $r$. Specifically, we let $\mathrm{Alt}_r(M) := \{M' : M \ll M', \Pi^\star(M_r) \cap \Pi^\star(M'_r) = \emptyset\}$, where $M \ll M'$ if for all $(s,a)$ we have $P(\cdot|s,a) \ll P'(\cdot|s,a)$.

**Theorem 3.1.** *Let $\delta \in (0, 1/2)$. For any $\delta$-PC algorithm we have $\liminf_{\delta \to 0} \frac{\mathbb{E}[\tau]}{\log(1/\delta)} \geq T^\star(M)$, where*

$$T^\star(M)^{-1} = \sup_{\omega \in \Omega(M)} \inf_{r \in \mathcal{R}} \inf_{M' \in \mathrm{Alt}_r(M)} \sum_{s,a} \omega(s,a) \mathrm{KL}_{M|M'}(s,a), \tag{1}$$

*with* $\mathrm{KL}_{M|M'}(s,a) := \mathrm{KL}(P(\cdot|s,a), P'(\cdot|s,a))$ *and* $\Omega(M) := \{\omega \in \Delta(\mathcal{S} \times \mathcal{A}) : \sum_a \omega(s,a) = \sum_{s',a'} P(s|s',a')\omega(s',a'), \forall s \in \mathcal{S}\}$.

**Discussion.** The proof of the theorem is presented in app. B.1. The result, obtained by classical change-of-measure arguments [20], extends the corresponding lower bound in the case of a single reward [17].

The quantity $T^\star(M)$ is referred to as the *characteristic rate* and is a non-convex optimization problem. It can be interpreted as a zero-sum game between an agent choosing a distribution (or allocation) $\omega$, and an opponent choosing a confusing model $M'$ [45] (in the multi-reward setting, the opponent needs also to consider the set of reward when choosing the confusing model, which makes the problem harder to solve). The distribution $\omega$ is a stationary distribution over states and actions that takes into account the transition dynamics, and, interestingly, the optimal solution $\omega_{\mathrm{opt}}$ to the optimization problem in eq. (1) also defines the optimal exploration strategy (as $\pi_{\mathrm{exp}}(a|s) = \omega_{\mathrm{opt}}(s,a)/\sum_b \omega_{\mathrm{opt}}(s,b)$).

In principle, with access to an optimization oracle that solves (1), we could apply classical Track-and-Stop (TaS) [20] algorithms and achieve asymptotically optimal sample complexity by tracking the optimal allocation. Unfortunately, even in classical BPI, computing the characteristic rate $T^\star(M)$ is in general a non-convex problem [16]. Instead, as in previous works [16, 17, 53], we focus on deriving a convex upper bound $U^\star(M)$ of $T^\star(M)$, which we call *relaxed characteristic rate*, that guarantees that we are still identifying the optimal policy at the cost of an increased sample complexity. We focus on the case of a finite set of user-specified rewards. The extension to externally provided reward signals and the consideration of a continuous set of rewards are detailed in the appendix (app. B.3 and app. B.4, respectively).

## 3.2 Relaxed Characteristic Rate

To define the relaxed characteristic rate, we state a few definitions of problem-dependent quantities for MDPs. We let $\Delta_r(s,a) := V_r^\star(s) - Q_r^\star(s,a)$ be the sub-optimality gap in $(s,a)$, $\mathrm{Var}_r(s,a) := \mathbb{E}_{s' \sim P(\cdot|s,a)}[(V_r^\star(s') - \mathbb{E}_{\hat{s} \sim P(\cdot|s,a)}[V_r^\star(\hat{s})])]$ be the variance of the optimal value function in the next state, and $\mathrm{MD}_r(s,a) := \|V_r^\star - \mathbb{E}_{s' \sim P(\cdot|s,a)}[V_r^\star(s')]\|_\infty$ be the maximum deviation starting from $(s,a)$. Additionally, for a given reward $r \in \mathcal{R}$ we also let $\Delta_r := \min_{s,a \neq \pi_r^\star(s)} \Delta_r(s,a)$, $\mathrm{Var}_r := \max_{s,a \in \mathcal{A}(s;M_r)} \mathrm{Var}_r(s,a)$ and $\mathrm{MD}_r = \max_{s,a \in \mathcal{A}(s;M_r)} \mathrm{MD}_r(s,a)$ be the minimum gap, maximum variance and deviation across sub-optimal state-action pairs, respectively.

We define the relaxed rate in terms of an upper bound on $T(\omega; M)$ that holds for all possible allocations $\omega \in \Delta(\mathcal{S} \times \mathcal{A})$, where

$$T(\omega; M)^{-1} := \inf_{r \in \mathcal{R}} \inf_{M' \in \mathrm{Alt}(M_r)} \sum_{s,a} \omega(s,a) \mathrm{KL}_{M|M'}(s,a). \tag{2}$$

**Theorem 3.2.** *For any allocation $\omega \in \Delta(\mathcal{S} \times \mathcal{A})$ we have that $T(\omega; M) \leq U(\omega; M)$, where*

$$U(\omega; M) := \max_{r \in \mathcal{R}} \max_{s, a \neq \pi_r^\star(s)} \frac{2\gamma^2 \mathrm{MD}_r(s,a)^2}{\Delta_r(s,a)^2 \omega(s,a)} + \max_{s'} \frac{H_r}{\Delta_r^2 \omega(s', \pi_r^\star(s'))}, \tag{3}$$

*where $H_r = \min\left\{ \frac{139(1+\gamma)^2}{(1-\gamma)^3}, \max\left( \frac{16\gamma^2 \mathrm{Var}_r(1+\gamma)^2}{(1-\gamma)^2}, \frac{6\gamma^{4/3}\mathrm{MD}_r^{4/3}(1+\gamma)^{4/3}}{(1-\gamma)^{4/3}} \right) \right\}$. We define the optimal value $U^\star(M) = U(\omega^\star; M)$, where $\omega^\star = \arg\inf_{\omega \in \Omega(M)} U(\omega; M)$ is the optimal allocation w.r.t. $U^\star(M)$.*

**Discussion.** In the above theorem (the proof can be in found in app. B.2), the various instance-specific quantities $\mathrm{Var}_r(s,a), \Delta_r(s,a), H_r$ and $\Delta_r$ are computed with respect to $M$. The result extends the corresponding upper bound in [17, 53].

We make some observations to clarify the key aspects of this result. First, $U(\omega; M)$ is convex in $\omega$ for a finite set of rewards. Hence $U(\omega; M)$ can be computed efficiently by using convex optimization methods (we report the runtime of our simulation in app. E.1).

For a continuous set of rewards there are some challenges to guarantee convexity, due to the fact that the minimum gap can be non-convex and discontinuous. However, we find that, *in practice*, it is sufficient to optimize over a set of rewards whose *convex hull* includes the original set of rewards (e.g., by using the canonical basis of rewards; refer also to app. B.4 for more details).

Second, using the allocation $\omega^\star$ guarantees that we identify the optimal policy, at the cost of an increased over-exploration [53], since $U^\star(M) \geq T^\star(M)$. The cost of this over-exploration is limited, since by plugging $\omega(s,a) = 1/(SA)$ one can see that $U^\star(M) = O\left( \frac{SA}{(1-\gamma)^3 \min_{r \in \mathcal{R}} \Delta_r^2} \right)$; note that the dependency in $(1-\gamma)^{-1}$ can be improved whenever the maximum deviation is smaller than $(1-\gamma)^{-1}$. Lastly, observe that the dependency $\Delta_r^2 \omega(s', \pi_r^\star(s'))$ in the second term in eq. (3) can be improved to $\Delta_r(s,a)^2 \omega(s', \pi_r^\star(s'))$ as in [53].

### 3.3 MR-NaS Algorithm

In this section we devise `MR-NaS`, an adaptation of MDP-NaS [17] to MR-BPI for a finite set of rewards. The algorithm consists of (1) an exploration policy (a.k.a. sampling rule) and (2) a stopping rule. We detail these components in the remainder of this section (and in app. C) and present the pseudo-code of `MR-NaS` in Alg. 1.

**Sampling rule.** The main idea, as in [16, 17, 53, 20], is that the exploratory policy should follow the allocation $\omega^\star = \arg\inf_{\omega \in \Omega(M)} U(\omega; M)$, as this will yield a sample-efficient exploration policy. However, as the model $M$ is unknown, we cannot directly compute $\omega^\star$. Alternatively, we employ the *certainty-equivalence* principle, and use the current estimate of the model $M_t$ in place of the true model $M$ to compute the allocation $\omega_t^\star = \arg\inf_{\omega \in \Omega(M_t)} U(\omega; M_t)$, where

$$U(\omega; M_t) = \max_{r \in \mathcal{R}} \max_{s, a \in \mathcal{O}_{t,r}^c(s)} \frac{2\gamma^2 \mathrm{MD}_{t,r}(s,a)^2}{\Delta_{t,r}(s,a)^2 \omega(s,a)} + \frac{H_{t,r}}{\Delta_{t,r}^2 \min_{s',a' \in \mathcal{O}_{t,r}(s)} \omega(s',a')}. \tag{4}$$

In the above expression, for any $t \geq 1$, the estimate $M_t$ is completely defined by the empirical transition function $P_t$:

$$P_t(s'|s,a) = \frac{N_t(s,a,s')}{N_t(s,a)} \text{ if } N_t(s,a) > 0, \text{ and } P_t(s'|s,a) = \frac{1}{S} \text{ otherwise}, \tag{5}$$

where $N_t(s,a)$ is the number of visits for each state-action pair up to time $t$ (sim. $N_t(s,a,s')$)). Letting $V_{t,r}^\star$ denote the optimal value in $M_{t,r} := (M_t, r)$ (sim. $Q_{t,r}^\star$) then $\mathcal{O}_{t,r}(s) = \{a : Q_{t,r}^\star(s,a) = V_{t,r}^\star(s)\}$ is the set of optimal actions for reward $r$ in state $s$ (and $\mathcal{O}_{t,r}^c(s)$ is the complement), $\mathrm{MD}_{t,r}(s,a) = \|V_{t,r}^\star(s) - \mathbb{E}_{s' \sim P_t(\cdot|s,a)}[V_{t,r}^\star(s')]\|_\infty$ is the maximum deviation starting from $(s,a)$

**Algorithm 1** `MR-NaS` (Multiple Rewards Navigate and Stop)
___
**Require:** Confidence $\delta$; exploration terms $(\alpha, \beta)$; reward vectors $\mathcal{R}$.
1: Initialize counter $t \leftarrow 1$, $N_t(s, a) \leftarrow 0$ for all $(s, a) \in \mathcal{S} \times \mathcal{A}$.
2: Set exploration term $\epsilon_t = 1/N_t(s_t)^\alpha$ and observe $s_1$.
3: **while** $tU(N_t/t; M_t)^{-1} < \beta(N_t, \delta)$ **do**
4:     Compute $\omega_t^\star = \inf_{\omega \in \Omega(M_t)} U(\omega; M_t)$ and let $\bar{\omega}_t^\star = (1/t) \sum_{n=1}^{t} \omega_n^\star$.
5:     Set $\pi_t(a|s_t) = (1 - \epsilon_t)\pi_t^\star(a|s_t) + \epsilon_t \pi_{f,t}(a|s_t)$, where $\pi_t^\star(a|s) = \bar{\omega}_t^\star(s, a)/\sum_{a'} \bar{\omega}_t^\star(s, a')$.
6:     Play $a_t \sim \pi_t(\cdot|s_t)$ and observe $s_{t+1} \sim P(\cdot|s_t, a_t)$.
7:     Update $N_t(s_t, a_t), M_t$ and set $t \leftarrow t + 1$.
8: **end while**
**Output:** an optimal policy $\hat{\pi}_{r,\tau}^\star$ for a given reward $r$.
___

for $M_{t,r}$, $\text{Var}_{t,r}(s, a) = \mathbb{E}_{s' \sim P_t(\cdot|s,a)}[(V_{t,r}^\star(s') - \mathbb{E}_{\hat{s} \sim P_t(\cdot|s,a)}[V_{t,r}^\star(\hat{s})])^2]$ is the corresponding next-state variance of the optimal value for $M_{t,r}$, and $\Delta_{t,r}(s, a) = V_{t,r}^\star(s) - Q_{t,r}^\star(s, a)$. These quantities can also be used similarly to compute $H_{t,r}$ (i.e., $H_r$ computed w.r.t. $M_{t,r}$).

At the same time, the sampling strategy must ensure that the estimated model $M_t$ converges to $M$. To that aim, we mix the computed allocation with a *forcing policy* $\pi_f$, ensuring that each state-action pair is sampled infinitely often. We use the forcing policy $\pi_{f,t}(\cdot|s) = \text{softmax}(-\beta_t(s)N_t(s, \cdot))$ with $\beta_t(s) = \frac{\beta \log(N_t(s))}{\max_a |N_t(s,a) - \min_b N_t(s,b)|}, \beta \in [0, 1]$ and $(\text{softmax}(x))_i = e^{x_i}/\sum_j e^{x_j}$ for a vector $x$. This choice encourages to select under-sampled actions for $\beta > 0$, while for $\beta = 0$ we obtain a uniform forcing policy $\pi_{f,t}(a|s) = 1/A$. In [16], the authors use an exploration factor $\epsilon_t = t^{-1/(m+1)}$, where $m$ is an problem-specific constant measuring to the "connectedness" of the MDP. As estimating this constant is typically hard for most MDPs, we instead use $\epsilon_t = 1/\max(1, N_t(s_t))^\alpha$, with $N_t(s) = \sum_a N_t(s, a)$ and $\alpha \in (0, 1]$, which only depends on the number of states visits. This change requires a modification of the proofs in [17], since now $\epsilon_t$ depends also on the number of visits of $s_t$. Finally, to ensure convergence of $\omega_t^\star \to \omega^\star$, we simply require that $\alpha + \beta \leq 1$.

**Stopping rule.** The stopping rule (line 3) is defined through the relaxed generalized likelihood ratio term $tU(N_t/t; M_t)$ and a threshold function $\beta(N_t, \delta)$. The term $N_t/t$ simply refers to the rates of visit of each state-action pair, while $M_t$ is the current estimate of the MDP (based on (5)). The threshold function is selected to guarantee that the algorithm is $\delta$-PC (the proof is in app. C.4).

**Proposition 3.1.** *Define the threshold* $\beta(N_t, \delta) = \log(1/\delta) + (S - 1)\sum_{s,a} \log\left(e\left[1 + \frac{N_t(s,a)}{S-1}\right]\right)$. *Then, the stopping rule* $\tau := \inf\{t \geq 1 : tU(N_t/t; M_t)^{-1} \geq \beta(N_t, \delta)\}$ *is $\delta$-PC, i.e.,*

$$\mathbb{P}[\tau < \infty, (\exists r \in \mathcal{R} : \hat{\pi}_{\tau,r} \notin \Pi^\star(M_r))] \leq \delta. \tag{6}$$

**Putting everything together.** Following our discussion, `MR-NaS` uses the estimate $M_t$ to compute the allocation $\omega_t^\star = \arg\inf_{\omega \in \Omega(M_t)} U(\omega; M_t)$ (line 4). To stabilize the exploration, we use an averaged allocation to obtain $\pi_t^\star = (1/t)\sum_{n=1}^{t} \omega_n^\star$ (similarly to the C-navigation rule in [20, 17]). Lastly, we mix $\pi_t^\star$ with a forcing policy $\pi_{f,t}$ (line 5). The samples observed from the MDP are then used to update the number of state action visits and the estimate of $M_t$. Finally, the stopping rule (line 3) determines when the algorithms has reached the desired confidence level.

**Sample complexity guarantees.** Lastly, we establish asymptotic optimality guarantees on the sample complexity of `MR-NaS` (see proofs in app. C.2.3 and app. C.3.5, respectively).

**Theorem 3.3.** *`MR-NaS` with the stopping rule in prop. 3.1 satisfies: (i) it stops almost surely and* $\mathbb{P}\left(\limsup_{\delta \to 0} \frac{\tau}{\log(1/\delta)} \leq U^\star(M)\right) = 1$; *(ii)* $\mathbb{E}[\tau] < \infty$ *and* $\limsup_{\delta \to 0} \frac{\mathbb{E}[\tau]}{\log(1/\delta)} \leq U^\star(M)$.

### 3.4 Numerical Results on Tabular MDPs

In this section we show results on different hard-exploration tabular environments: `Riverswim` [54], `Forked Riverswim` [53], `DoubleChain` [22] and `NArms` [54] (an adaptation of `SixArms` to $N$ arms). We compare `MR-NaS` against `RF-UCRL` [22] (a reward-free exploration method), `ID3AL` [33] (a maximum entropy exploration approach) and `MR-PSRL`, a multi-reward adaptation of `PSRL` [35] (posterior sampling for RL) that is outlined in app. D.1.2. A full description of the environments

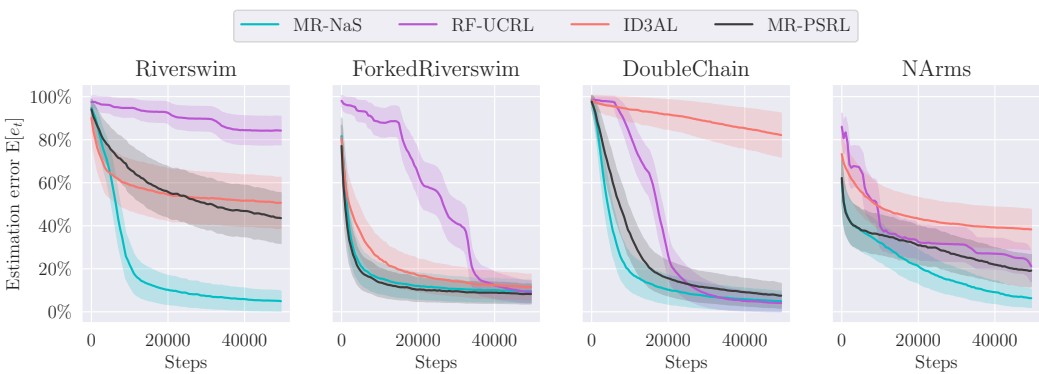

Figure 1: Average estimation error over $t = 1, \ldots, T$ steps (note that $\hat{r}$ is a random variable). Shaded area indicates $95\%$ confidence intervals over 100 seeds.

can be found in app. D.1. As the baselines are not specific to our setting, we consider different sets of experiments. In this section, we show results for `MR-NaS` on the more challenging setting with continuous sets of rewards, while in app. D.1, we also investigate the problem of identifying best policies with a finite set of rewards.

More precisely, we consider the problem of correctly estimating the optimal policies in a convex set of rewards $[0, 1]^{SA}$ as well as the optimal policies of the canonical basis, with discount factor $\gamma = 0.9$. We consider $\mathcal{R} = \mathcal{R}_{\text{canonical}} \cup \mathcal{R}_{\text{rnd}}$, where $\mathcal{R}_{\text{rnd}} = \{R_1, \ldots, R_{30}\}$ consists of 30 reward vectors uniformly sampled from $[0, 1]^{SA}$, different for each seed. For `MR-NaS`, we use the insight from the analysis of a continuous set of rewards (see app. B.4.2) which shows that exploring according to the canonical basis $\mathcal{R}_{\text{canonical}}$ is sufficient to identify the optimal policies in a continuous set.

We run each algorithm for $T = 50 \cdot 10^3$ steps, and we report the fraction of misidentified policies $e_{t,r}$ at time step $t$ and for reward $r$. Fig. 1 shows the results for this setting over 50 seeds. We see how depending on the environment, certain algorithms may be more efficient. Nonetheless, `MR-NaS` is able to reach low estimation error relatively quickly on all of the environments. We refer the reader to app. D.1 for further details.

## 4 Exploration in Deep Reinforcement Learning with a Set of Rewards

Extending `MR-NaS` to Deep-RL is not straightforward. Firstly `MR-NaS` is a model-based method, secondly there is no closed-form solution to $\arg\inf_{\omega \in \Omega} U^\star(\omega; M)$. To circumvent these issues, we adapt the `DBMF-BPI` algorithm from [53], where the authors propose `DBMF-BPI`, a deep-RL exploration approach for single-reward BPI.

**The `DBMF-BPI` algorithm.** `DBMF-BPI` [53] is an algorithm for model-free exploration. In [53] they suggest that a model-free solution can be achieved by (i) using the *generative solution* $\arg\inf_{\omega \in \Delta(S \times A)} U^\star(\omega; M)$, which can be computed in closed form (see app. B.2.4), and (ii) by estimating the parametric uncertainty of the MDP-specific quantities (sub-optimality gaps, etc.). In other words, the generative solution may be close enough to the model-based one. Nonetheless, this solution may be biased because the true sub-optimality gaps are unknown, and therefore we need account for the parametric uncertainty in the estimates.

`DBMF-BPI` estimates the parametric uncertainty using *an ensemble of Q-networks*. This ensemble of size $B$, when trained on different data, can be used to mimic the bootstrap procedure [55]: the samples $(x_b)_{b \in [B]}$ from this ensemble are used to build an empirical CDF $\hat{P}(X \le x) = \frac{1}{B} \sum_{b=1}^{B} \mathbf{1}_{\{X \le x_b\}}$ around some quantity of interest $X$ (in our case the quantity of interest is the $Q$-function). A randomized prior value mechanism [56] addresses the restricted support of $\hat{P}$ by adjusting the sampling of $(x_b)_b$. This empirical CDF is then used to sample the $Q$-values at each time step, which are then used to compute the generative solution $\omega_t$.

---
**Algorithm 2** `DBMR-BPI`(Deep Bootstrapped Multiple-Rewards BPI) - Exploration phase
---
1: Initialize replay buffer $\mathcal{D}$, ensemble network parameter $\theta$.
2: **for** $t = 0, 1, 2, \ldots,$ **do**
3:     Choose reward $r$ with probability $\propto \frac{1}{\Delta_{t,r}^2}$ and compute allocation $\omega_t$ using `DBMF-BPI` with reward $r$.
4:     Sample $a_t \sim \omega_t(s_t, \cdot)$ and observe $s_{t+1} \sim P(\cdot|s_t, a_t)$.
5:     Add transition $(s_t, a_t, r_{1,t}, \ldots, r_{R,t}, s_{t+1})$ to the buffer $\mathcal{D}$, with $r_{i,t} = r_i(s_t, a_t), i = 1, \ldots, |\mathcal{R}|$.
6:     Train the ensemble $\theta$ according to `DBMF-BPI` and update estimates $\Delta_{t,r}$ for each reward $r \in \mathcal{R}$.
7: **end for**
---

**Multi-reward adaptation.** We propose `DBMR-BPI`, an extension of `DBMF-BPI` to the multi-reward setting with a finite set of rewards. `DBMR-BPI` at each time step observes a vector of rewards, and drives its exploration according to these observed values. More precisely, the extension mainly consists of the following modifications:

1. The $Q$-values are learned for all rewards. In the finite action setting, this implies that there are $A|\mathcal{R}|$ values to estimate in every state. Alternatively, one can choose to feed the networks with a one-hot encoding of the chosen reward (or use embeddings), but we experienced this change to be less effective.

2. We explore according to the difficulty of the reward function. Specifically, we explore a reward $r$ with probability $\propto 1/\Delta_{t,r}^2$, as this term is typically the leading factor in the sample complexity. For the sake of stability, at the beginning of each episode (or every $\approx 1/(1-\gamma)$ steps) we sample a reward $r \sim \mathcal{R}$ to explore, and apply `DBMF-BPI` on this reward.

The main steps of the algorithm are outlined in Alg. 2, and all the details can be found in app. E.1.

### 4.1 Numerical Results

We evaluate the performance of `DBMR-BPI` and compare it with state-of-the-art method in unsupervised RL [24] (note that this is the first algorithm for multi-reward exploration in Deep-RL). More precisely, we compare our algorithm to `RND` [26] (Random Network Distillation), `Disagreement`[27] and `APT` [29]. The exploration procedure of these methods is guided by an intrinsic reward function that is learnt during exploration (e.g., by quantifying the parametric uncertainty of the model, or some prediction uncertainty; these methods are detailed in app. E.2). As in [57, 53], we evaluate on different challenging environments from the DeepMind behavior suite [36]: the Cartpole swing-up problem and a stochastic version of DeepSea [36, 53] with varying level of difficulties. We also evaluate the methods on the link adaptation problem, a radio network control task where the agent needs to choose a coding and modulation scheme in wireless networks [37], and report the results in app. D.3.1. Lastly, we also evaluate the exploration quality of `DBMR-BPI` both on a known set of pre-defined rewards and on unseen rewards.

**The Cartpole swing-up [36].** In the classical version of this environment, the goal is to balance a pole initially pointing downward, and the agent incurs a cost $-c$ whenever the cart moves. In the multi-reward variant of Cartpole, the goal is to stabilize the pole vertically around a specific position $x_0$. The agent earns a positive reward only if the pole's angle, its angular velocity, and the cart's position, respectively, satisfy the conditions (1) $\cos(\theta) > k/20$, (2) $|\dot{\theta}| \leq \theta_0$, and (3) $|x - x_0| \leq 1 - k/20$, where $\theta_0$ is a fixed constant and $k$ is an integer representing the difficulty. We considered 5 values of $x_0$ linearly spaced in $[-1.5, 1.5]$, so that $|\mathcal{R}| = 5$. To assess `DBMR-BPI`'s capacity to generalize on unseen rewards, we uniformly sample 5 additional values of $x_0$ in the same interval that are not used during training, and we denote them by $\mathcal{R}_{\mathrm{rnd}}$.

In tab. 1 we present results for two levels of difficulty $k \in \{3, 5\}$ and different training steps $T$. The table presents statistics for the random variable $X_i = \frac{1}{T}\sum_{t=1}^T \mathbf{1}_{r_i(s_t, a_t) > 0}$, where $i$ is the $i$-th reward in the set. This metric measures the average amount of information collected by the agent useful to learn how to stabilize the pole upright. $\mathrm{Avg}_{\mathcal{R}}(\{X_i\})$ (sim. the others metrics) indicates the arithmetic mean of $\{X_i\}_i$ over the rewards. We average results across 30 seeds.

In general, we see that `DBMR-BPI` outperforms all of the unsupervised learning baselines. Interestingly, even though `DBMR-BPI` uses the base set of rewards $\mathcal{R}$ to explore, we see better performance when collecting rewards from $\mathcal{R}_{\mathrm{rnd}}$ (which are not used by `DBMR-BPI` during exploration). The reason

| | | **Base rewards** $\mathcal{R}$ | | | **Random rewards** $\mathcal{R}_{\mathrm{rnd}}$ | | |
|---|---|---|---|---|---|---|---|
| | | $\mathbb{E}[\mathrm{Avg}_{\mathcal{R}}(\{X_i\})]$ | $\mathbb{E}[\mathrm{std}_{\mathcal{R}}(\{X_i\})]$ | $\mathbb{E}[\min_{\mathcal{R}}(\{X_i\})]$ | $\mathbb{E}[\mathrm{Avg}_{\mathcal{R}_{\mathrm{rnd}}}(\{X_i\})]$ | $\mathbb{E}[\mathrm{std}_{\mathcal{R}_{\mathrm{rnd}}}(\{X_i\})]$ | $\mathbb{E}[\min_{\mathcal{R}_{\mathrm{rnd}}}(\{X_i\})]$ |
| $k=3$ | DBMR-BPI | **21.3(20.6, 21.9)** | **6.1(5.5, 6.6)** | **12.8(12.0, 13.6)** | **23.1(22.2, 24.2)** | **4.1(3.6, 4.6)** | **17.1(16.0, 18.2)** |
| $T=150000$ | RND | 1.3(1.0, 1.6) | 0.3(0.2, 0.3) | 0.9(0.7, 1.2) | 1.4(1.1, 1.7) | 0.2(0.1, 0.2) | 1.1(0.9, 1.4) |
| | APT | 0.3(0.3, 0.4) | 0.1(0.1, 0.2) | 0.2(0.1, 0.2) | 0.3(0.3, 0.4) | 0.1(0.1, 0.1) | 0.2(0.2, 0.2) |
| | Disagreement | 0.9(0.7, 1.2) | 0.3(0.2, 0.6) | 0.5(0.4, 0.6) | 1.0(0.7, 1.6) | 0.3(0.1, 0.5) | 0.6(0.5, 0.7) |
| $k=5$ | DBMR-BPI | **20.2(19.6, 20.6)** | **6.4(6.0, 6.9)** | **11.4(10.7, 12.0)** | **22.3(21.5, 23.1)** | **4.5(4.0, 4.9)** | **15.5(14.4, 16.7)** |
| $T=200000$ | RND | 1.2(0.9, 1.5) | 0.2(0.2, 0.3) | 0.8(0.6, 1.1) | 1.1(0.9, 1.4) | 0.2(0.1, 0.2) | 0.9(0.7, 1.1) |
| | APT | 0.3(0.3, 0.4) | 0.1(0.1, 0.1) | 0.2(0.1, 0.2) | 0.3(0.3, 0.4) | 0.1(0.1, 0.1) | 0.2(0.2, 0.3) |
| | Disagreement | 0.7(0.5, 1.0) | 0.3(0.2, 0.4) | 0.4(0.3, 0.6) | 0.8(0.5, 1.0) | 0.2(0.1, 0.3) | 0.5(0.4, 0.7) |

Table 1: **Cartpole swing-up**. Statistics for the random variable $X_i = \frac{1}{T}\sum_{t=1}^{T}\mathbf{1}_{r_i(s_t,a_t)>0}$ (with $r_i \in \mathcal{R}$ or $r_i \in \mathcal{R}_{\mathrm{rnd}}$). Values are multiplied by 100 and rounded to the first digit. In bold statistically significant results (in parentheses we indicate the 95% confidence interval over 30 seeds).

may be that some rewards in $\mathcal{R}$, (e.g., those with $|x_0| = 1.5$), are harder to learn, while it is unlikely for the random rewards to have such values of $x_0$. This result seems to suggest that data collected by DBMR-BPI could also be used to optimize similar, but unseen, rewards. We refer the reader to app. D.2.1 for more results and details on the environment.

**The DeepSea [36].** In the DeepSea environment the agent moves in a riverswim-like environment, and the actions are randomized for each different seed. The agent starts in the top-left corner of a grid of size $N \times N$, and can only move diagonally, towards the bottom of the grid. In the classical setup, the agent observes a $N^2$-dimensional one-hot encoding of its position, and the goal is to reach the bottom-right corner. We adapt it to a multi-reward setting by considering $N$ different rewards $\mathcal{R} = \{r_1, \ldots, r_N\}$, so that $r_i$ assigns a positive reward whenever the agent reaches column $i$ in the last row of the grid. As in [53], we include a small probability of the agent taking the wrong action.

To assess the capacity of DBMR-BPI to collect unseen rewards, for each seed, we sampled 20 reward functions $\mathcal{R}_{\mathrm{rnd}} = (r'_i)_{i=1}^{20}$, each of dimension $N(N+1)/2$ (where the $i$-th value corresponds to a possible position in the grid), from a Dirichlet distribution with parameter $\alpha = (\alpha_i)_i$, with $\alpha_i = 1/(10N)$ for all $i = 1, \ldots, N(N+1)/2$, and computed the reward for these additional reward functions. The reader can find more details regarding the environment in app. D.2.2.

| | | **Base rewards** $\mathcal{R}$ | | | **Random rewards** $\mathcal{R}_{\mathrm{rnd}}$ | |
|---|---|---|---|---|---|---|
| | | $\mathbb{E}[\mathrm{GM}_{\mathcal{R}}(\{X_i\})]$ | $\mathbb{E}[\mathrm{std}_{\mathcal{R}}(\{X_i\})]$ | $\mathbb{E}[\sum_{i=1}^{N}\mathbf{1}_{X_i=0}]$ | $\mathbb{E}[\mathrm{Avg}_{\mathcal{R}_{\mathrm{rnd}}}(\{X_i\})]$ | $\mathbb{E}[\mathrm{std}_{\mathcal{R}_{\mathrm{rnd}}}(\{X_i\})]$ |
| $N=20$ | DBMR-BPI | **4.6(4.6, 4.6)** | **1.8(1.7, 1.8)** | **0.0(0.0, 0.0)** | **8.9(8.5, 9.2)** | **3.6(3.2, 4.0)** |
| $T=50000$ | RND | **4.6(4.2, 4.7)** | **1.7(1.6, 1.8)** | **0.0(0.0, 0.1)** | **8.7(8.4, 9.1)** | **3.5(3.1, 4.0)** |
| | APT | 1.4(0.8, 1.9) | 5.3(4.8, 5.8) | 0.9(0.6, 1.3) | **8.7(8.2, 9.2)** | 6.6(5.9, 7.4) |
| | Disagreement | 0.0(0.0, 0.0) | 6.2(5.9, 6.5) | 4.0(3.4, 4.5) | **8.9(8.3, 9.5)** | 6.4(5.8, 7.1) |
| $N=30$ | DBMR-BPI | **3.1(2.9, 3.2)** | **0.7(0.7, 0.7)** | **0.0(0.0, 0.1)** | **6.0(5.8, 6.3)** | **3.0(2.5, 3.6)** |
| $T=100000$ | RND | 1.8(1.3, 2.2) | 1.9(1.9, 2.0) | 0.3(0.2, 0.5) | **6.1(5.9, 6.4)** | **3.0(2.4, 3.5)** |
| | APT | 0.0(0.0, 0.0) | 3.9(3.6, 4.2) | 3.3(2.8, 3.8) | 5.9(5.6, 6.2) | 4.3(3.8, 4.7) |
| | Disagreement | 0.0(0.0, 0.0) | 7.0(6.6, 7.3) | 11.5(10.8, 12.2) | **6.2(5.6, 6.7)** | 6.1(5.6, 6.6) |

Table 2: **DeepSea**. Statistics for the random variable $X_i = \frac{1}{T}\sum_{t=1}^{T} r_i(s_t, a_t)$ (with $r_i \in \mathcal{R}$ or $r_i \in \mathcal{R}_{\mathrm{rnd}}$). Values are multiplied by 100 (except the third metric for the base rewards) and rounded to the first digit. In bold statistically significant results (in parentheses we indicate the 95% confidence interval over 30 seeds).

In tab. 2 we show some statistics for the random variable $X_i = \frac{1}{T}\sum_{t=1}^{T} r_i(s_t, a_t)$, that quantifies the average reward collected for the $i$-th reward in a specific instance. In the table $\mathrm{GM}_{\mathcal{R}}(\{X_i\})$ indicates the geometric mean over the rewards (the arithmetic mean would be a constant for these base rewards in this setting, and the GM is more appropriate [58]), and $\sum_{i=1}^{N}\mathbf{1}_{X_i=0}$ indicates the number of cells in the last row of the grid that have not yet been visited at time step $T$.

When considering the set of rewards $\mathcal{R}$ used by DBMR-BPI , we see that APT and Disagreement do not match the performance of DBMR-BPI and RND, focusing most of the time on a few rewards. On the other hand, while DBMR-BPI and RND show similar performance for $N = 20$, the difference increases for $N = 30$ on the base set of rewards. When considering the total reward collected from the set $\mathcal{R}_{\mathrm{rnd}}$, we see that DBMR-BPI achieves performance similar to RND, which confirms the capability of DBMR-BPI to collect reward from unseen reward functions (refer to app. D.2.2 for further details).

## 5 Discussion and Conclusions

**Exploration in RL and Best Policy Identification.** There exists a variety of exploration methods in RL [59] and DRL (see [60, 53] and references therein). Our setting is a generalization of classical single-reward BPI with fixed confidence [16, 17, 61, 62, 18, 19]. This line of work typically leverages instance-specific lower bounds on the sample complexity (or approximations of it) to devise efficient exploration strategies, but do not directly address the MR-BPI problem. While we primarily draw inspiration from [17, 53], we enhance their work by introducing a multi-reward generalization applicable to both the tabular case and DRL, along with some technical improvements (see app. C and app. E.1). Lastly, while not directly related, in [63, 64, 65] the authors study the identification of the Pareto optimal frontier in a multi-objective bandit model, a framework yet to be explored in MDPs.

**Multi-Reward.** There are several works considering a setting where the agent deals with multiple diverse reward signals [10, 66, 11, 31]. Shelton [10] investigates the case where an agent observes a vector of rewards from multiple sources at each time step. The author proposes an algorithm attaining a Nash equilibrium on the average return of each source. In [66] they investigate a multi-reward setting with continuous states and devise an algorithm yielding optimal policies for any convex combination of reward functions. Arguably, the closest work to ours is Dann et al. [11], where they investigate multi-reward tabular MDPs, aiming at optimizing a regret measure that combines the various rewards components through an operator and provide regret bounds for action elimination algorithms. Despite the similarities, these works focus on the classical regret minimization problem and do not address the pure exploration case.

A related framework is multi-objective RL [30, 67, 31], a technique with various applications in areas such as molecular design [68] and pricing strategy optimization [69]. In this framework, the agent seeks to optimize multiple rewards simultaneously. To the best of our knowledge there are no work investigating the exploration problem in Multi-Objective RL, a setting that could potentially benefit from more efficient exploration strategies, such as the ones that we propose.

**Reward-Free Exploration.** A relevant body of literature considers Reward-Free Exploration (RFE) [21, 22, 70, 71, 23, 72], where the agent learns behaviours without a specific reward. For example, Jin et al. [21] study reward-free RL in the tabular episodic case to identify $\varepsilon$-optimal policies for any possible reward. They provide a lower bound on the length of exploration phase scaling as $\Omega(H^2 S^2 A/\varepsilon^2)$ and devise an algorithm nearly matching it of order $\tilde{O}(H^5 S^2 A/\varepsilon^2)$. This bound was improved in [22] and [70], where the authors propose the RF-UCRL and RF-Express algorithms for episodic reward-free RL. On a similar line, the authors of [23] focus on an RFE setting where the minimal sub-optimality gap is a known quantity, and provide an algorithm with reduced sample complexity. Another relevant RFE framework is maximum entropy exploration [32, 33, 24, 73], which studies the problem of state entropy maximization. This type of exploration is particularly useful for offline RL [74] since it is well know that the coverage of the state space characterizes the sample complexity of the problem [75, 76]. Lastly, the reward-free exploration phase can be followed by a downstream task adaptation, as in unsupervised RL [24, 25, 26, 27, 28, 29]. While these settings are highly relevant and provide valuable insights for other RL domains (such as facilitating representation learning in a self-supervised manner [77, 78, 79]), some RFE techniques can be impractical in realistic scenarios (e.g., exploring for any reward) or do not address the MR-BPI problem directly. Furthermore, unlike our setup, most of the works in RFE focus on the minimax setting only.

**Conclusions.** In this work, we investigated the MR-BPI problem, focusing on efficiently determining the best policies for a diverse set of rewards $\mathcal{R}$ with a specified confidence level. We established a fundamental sample complexity result for MR-BPI and proposed two efficient methods for multi-reward exploration: `MR-NaS` for tabular MDPs and `DBMR-BPI` for DRL. Numerical results on challenging exploration environments suggests that both algorithms exhibit strong exploration capabilities and generalization properties. We believe that our study benefits various areas of RL by enabling more principled exploration in multi-reward scenarios. This includes applications in multi-objective RL, which naturally lends itself to optimization over multiple objectives, and offline RL, where it is crucial to collect a dataset that performs well across different rewards.

## Acknowledgments

This research was conducted while the authors were employed at Ericsson in Stockholm, Sweden. We would like to express our gratitude to Burak Demirel, Pablo Soldati, Yu Wang and Peter Björkén for their support during the preparation of this work. We would also like to thank Maxime Bouton and Martin Isaksson for their critical feedback that enhanced the quality of this paper.

On a personal note, Alessio Russo wishes to personally thank Letizia Orsini for her assistance in reviewing the manuscript. I also dedicate this work to the memory of my beloved cat, Caterina, whose companionship and gentle spirit brought immeasurable joy to my life and was a comforting presence throughout my academic journey. Her memory will forever be a source of comfort and inspiration.

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

# Appendix

## Contents

# A Limitations and Broader Impact

**Limitations.** We discuss some limitations of the current work and highlight major directions of improvement.

- **Tabular MDP.** The current theoretical analysis only applies to tabular MDPs, and may not necessarily extend to continuous domains in a similar way.

- **Tracking of the allocation.** In `MR-NaS` we employ a probabilistic policy, which may lead to worse tracking performance compared to deterministic approaches (see e.g., the classical tracking procedures for best arm identification in multi-armed bandits [20]). These approaches are, however, harder to use in an MDP setting, since it becomes harder to guarantee correct tracking and convergence of the allocation $\omega_t^\star$ to $\omega^\star$.

- **Lazy updates.** `MR-NaS` computes an allocation $\omega_t^\star = \arg\inf_{\omega \in \Omega(M_t)} U(\omega_t; M_t)$ at each time step, which could be computationally prohibitive for large-dimensional problems. A possible way to address this issue is to use "lazyness" [80], i.e., to compute $\omega_t^\star$ only at time steps $t \in \mathcal{T}$, where $\mathcal{T} \subset \mathbb{N}$ prescribes the allocation update schedule (see [80]).

- **Scaling of the rewards for `DBMR-BPI`.** The method that we introduce for exploration in deep-RL, `DBMR-BPI`, explores according to a set of rewards. These rewards, if not correctly scaled, may impact the learning performance since the method relies on estimating the sub-optimality gaps.

- **Number of rewards for `DBMR-BPI`.** The usage of `DBMR-BPI` could be limited by the number of rewards if it is too high, as the output size of the $Q$-network in `DBMF-BPI` is $A \cdot |\mathcal{R}|$. However, our experiments with an ensemble size of 20 and $A \cdot |\mathcal{R}| \approx 90$ did not encounter any issues. Furthermore, the training can be efficiently done as shown in the code, without looping over the rewards. Nevertheless, exploring the scalability to a larger number of rewards remains an interesting area for future research.

- **Finite action space for `DBMR-BPI`.** In this paper, we analyze `DBMR-BPI` within a finite action space. Although continuous action spaces are equally important, extending the algorithm to such spaces requires additional work. It is currently unclear how to adapt the approach presented in [53] to continuous action spaces.

- **Assumption on the uniqueness of the optimal policy.** In the paper we assume that there is a unique optimal policy. This assumption is common in the BPI literature [18, 16, 17, 53, 20]. Addressing MDPs with multiple optimal policies or identifying $\varepsilon$-optimal policies necessitates the use of more complex overlapping hypothesis tests [81]. This approach is already complex when applied to multi-armed bandits (i.e., MDPs with a single state), and extending it to full MDPs presents an even greater challenge. In that sense, we can provide some insights on the main problems and potential solutions:

  - In presence of multiple optimal policies, one may be interested in ensuring that at the stopping time we have $\Pi^\star(M_{\tau,r}) \subseteq \Pi^\star(M_r)$ for all rewards $r \in \mathcal{R}$. This guarantees that we identify a subset of policies that are optimal. Unfortunately, ensuring this condition causes an asymmetry between the definition of the set of confusing models, and the event $M \in \mathrm{Alt}(M_t)$ in proof of the stopping rule, that cannot be easily reconciled.

  - A positive result is that if the set of confusing models can be decomposed as a union, or intersection, over the optimal policies in $M$, then the decomposition proofs presented in apps. B.2 and B.4.3, can be easily extended to the case of multiple optimal policies by considering the sub-optimal actions $a \notin \mathcal{O}_r(s)$ in the decomposition, where $\mathcal{O}_r(s) = \{a : Q^\star_{M_r}(s,a) = V^\star_{M_r}(s)\}$.

**Broader impact.** This paper primarily focuses on foundational research in reinforcement learning, particularly the exploration problem. Although we do not directly address societal impacts, we recognize their importance. The methods proposed here improve the sample efficiency of RL algorithms and could be applied in various contexts with societal implications. For instance, RL is used in decision-making systems in healthcare, finance, and autonomous vehicles, where biases or errors could have significant consequences. Therefore, while the immediate societal impact of our work may not be evident, we urge future researchers and practitioners to carefully consider the ethical implications and potential negative impacts in their specific applications.

# B    Sample Complexity Bounds

In this appendix we first prove thm. 3.1. In the subsequent subsection we study how to obtain the relaxed characteristic rate presented in thm. 3.2. After that, we present extensions of these results to the case of random rewards (app. B.3) and continuous rewards (app. B.4). Finally, in the last subsection, we present some useful lemmas.

## B.1    Lower Bound Proof

The proof leverage the classical change of measure argument [41, 20] and proceeds similarly to [16]. For simplicity, we consider the generative setting (i.e., we consider allocations $\omega \in \Delta(\mathcal{S} \times \mathcal{A})$), while the extension to the forward model follows similarly to [17].

*Proof of thm. 3.1.* The proof begins by analysing the likelihood ratio of the observations, and compare against the worst confusing model. Then, by considering the $\delta$-PC event and a data processing inequality yields the lower-bound.

*Log-likelihood ratio.* Consider the MDP $M = (\mathcal{S}, \mathcal{A}, P, \gamma)$ and an alternative model $M' = (\mathcal{S}, \mathcal{A}, P', \gamma)$ such that $P \ll P'$. The log-likelihood ratio between $M$ and $M'$ of a sequence of observations $(z_0, z_1, \dots)$, where $z_i = (s_i, a_i)$, is defined as

$$L_t = \sum_{n=1}^{t} \log \frac{P(s_n|z_{n-1})}{P'(s_n|z_{n-1})},$$

To simplify the analysis, let

$$L_t^{s,a} := \sum_{n=1}^{t} \mathbf{1}_{\{Z_{n-1}=(s,a)\}} \log \left( \frac{P(s_n|s,a)}{P'(s_n|s,a)} \right) = \sum_{n=1}^{N_t(s,a)} \log \left( \frac{P(y_n|s,a)}{P'(y_n|s,a)} \right),$$

where $y_n$ denotes the next state after the pair $(s,a)$ has been visited for the $n$-th time $(s,a)$ and $N_t(s,a)$ is the number of times the state-action pair $(s,a)$ has been visited by the algorithm up to time $t$. Then we can write $L_t = \sum_{s,a} L_t^{s,a}$. Taking the expectation over the measure induced by $M$, we have

$$\mathbb{E}_M[L_t] = \mathbb{E}_M \left[ \sum_{n=1}^{t} \log \frac{P(s_n|z_{n-1})}{P'(s_n|z_{n-1})} \right],$$

where $\mathbb{E}_M[\cdot]$ is the expectation under the probability measure induced by a model $M$ (sim. $\mathbb{P}_M[\cdot]$). Then, at the stopping time $\tau$ we have

$$\mathbb{E}_M[L_\tau] = \sum_{s,a} \mathbb{E}_M \left[ L_\tau^{s,a} \right],$$

$$= \sum_{s,a} \mathbb{E}_M \left[ \sum_{n=1}^{N_\tau(s,a)} \log \left( \frac{P(y_n|s,a)}{P'(y_n|s,a)} \right) \right],$$

$$= \sum_{s,a} \mathbb{E}_M \left[ \sum_{t=0}^{\infty} \mathbf{1}_{\{N_\tau(s,a) \geq t\}} \log \left( \frac{P(y_t|s,a)}{P'(y_t|s,a)} \right) \right].$$

Since $X_n^{s,a} := \log \left( \frac{P(y_n|s,a)}{P'(y_n|s,a)} \right)$ is independent of $\mathcal{F}_{n-1}$, and since $\{N_\tau(s,a) \leq n-1\}^c = \{N_\tau(s,a) \geq n\} \in \mathcal{F}_{n-1}$ we have that

$$\mathbb{E}_M[L_\tau] = \sum_{s,a} \sum_{t=0}^{\infty} \mathbb{P}_M[N_\tau(s,a) \geq t] \mathrm{KL}_{M|M'}(s,a),$$

$$= \sum_{s,a} \mathbb{E}_M[N_\tau(s,a)] \mathrm{KL}_{M|M'}(s,a),$$

where $\mathrm{KL}_{M|M'}(s,a) = \mathrm{KL}(P(s,a), P'(s,a))$.

*δ-PC argument.* Consider a $\delta$-PC algorithm, and define the event $\mathcal{E} = \{\forall r \in \mathcal{R}, \hat{\pi}_{\tau,r} \in \Pi^\star(M_r)\}$ and observe that $\mathcal{E}^c = \{(\exists r \in \mathcal{R}, \hat{\pi}_{\tau,r} \notin \Pi^\star(M_r))\}$.

Let $\mathrm{Alt}(M) = \{M' \in \mathcal{M} | \exists r : \Pi^\star(M'_r) \cap \Pi^\star(M_r) = \emptyset\}$, which can be rewritten as $\mathrm{Alt}(M) = \cup_{r \in \mathcal{R}} \mathrm{Alt}_r(M)$, where $\mathrm{Alt}_r(M) = \{M' \in \mathcal{M} | \Pi^\star(M'_r) \cap \Pi^\star(M_r) = \emptyset\}$.

Before proceeding by applying the transportation lemma in [82], we first note that for any $\delta$-PC algorithm we have $\mathbb{P}_M[\mathcal{E}^c] \leq \delta$. Denote by $\Pi$ the set of all Markovian policies. Then, under $M'$ we have

$$
\begin{aligned}
\mathbb{P}_{M'}[\mathcal{E}^c] &= \mathbb{P}_{M'}[\exists r : \hat{\pi}_{\tau,r} \notin \Pi^\star(M_r)], \\
&\geq \max_{r'} \mathbb{P}_{M'}[\hat{\pi}_{\tau,r'} \notin \Pi^\star(M_{r'})], \\
&\geq \mathbb{P}_{M'}[\hat{\pi}_{\tau,r} \notin \Pi^\star(M_r)], \\
&= \mathbb{P}_{M'}[\hat{\pi}_{\tau,r} \in \Pi \setminus \Pi^\star(M_r)], \\
&\geq \mathbb{P}_{M'}[\hat{\pi}_{\tau,r} \in \Pi^\star(M'_r)], \\
&\geq \min_r \mathbb{P}_{M'}[\hat{\pi}_{\tau,r} \in \Pi^\star(M'_r)], \\
&\geq \mathbb{P}_{M'}[\forall r, \hat{\pi}_{\tau,r} \in \Pi^\star(M'_r)], \\
&= 1 - \mathbb{P}_{M'}[\exists r, \hat{\pi}_{\tau,r} \notin \Pi^\star(M'_r)] \geq 1 - \delta.
\end{aligned}
$$

Hence, in view of the data processing inequality in [82], for any $M' \in \mathrm{Alt}(M)$ we can lower bound the expected log-likelihood at the stopping time $\tau$ as

$$
\mathbb{E}_M[L_\tau] = \sum_{s,a} \mathbb{E}_M[N_\tau(s,a)]\mathrm{KL}_{M|M'}(s,a) \geq \mathrm{kl}(\mathbb{P}_M[\mathcal{E}], \mathbb{P}_{M'}[\mathcal{E}]) \geq \mathrm{kl}(\delta, 1 - \delta).
$$

In view of the transportation lemma in [82], for any $M' \in \mathrm{Alt}(M)$ we can lower bound the expected log-likelihood at the stopping time $\tau$ as

$$
\mathbb{E}_M[L_\tau] = \sum_{s,a} \mathbb{E}_M[N_\tau(s,a)]\mathrm{KL}_{M|M'}(s,a) \geq \mathrm{kl}(\mathbb{P}_M[\mathcal{E}], \mathbb{P}_{M'}[\mathcal{E}]) \geq \mathrm{kl}(1 - \delta, \delta).
$$

As the inequality above holds for all $M' \in \mathrm{Alt}(M)$, and by optimizing over the set of confusing model, we have

$$
\inf_{M' \in \mathrm{Alt}(M)} \sum_{s,a} \mathbb{E}_M[N_\tau(s,a)]\mathrm{KL}_{M|M'}(s,a) = \min_{r \in \mathcal{R}} \inf_{M' \in \mathrm{Alt}_r(M)} \sum_{s,a} \mathbb{E}_M[N_\tau(s,a)]\mathrm{KL}_{M|M'}(s,a)
$$
$$
\geq \mathrm{kl}(1 - \delta, \delta).
$$

*Sample complexity.* By letting $\omega(s,a) = \mathbb{E}_M[N_\tau(s,a)]/\mathbb{E}_M[\tau]$, and optimizing over all possible allocations $\omega \in \Omega$, we have

$$
\mathbb{E}_M[\tau] \max_{\omega \in \Delta(\mathcal{S} \times \mathcal{A})} \min_{r \in \mathcal{R}} \inf_{M' \in \mathrm{Alt}_r(M)} \sum_{s,a} \omega(s,a)\mathrm{KL}_{M|M'}(s,a) \geq \mathrm{kl}(1 - \delta, \delta).
$$

Finally, taking the limit as $\delta \to 0$ yields the result. $\qquad\square$

The extension from the generative model to the forward model can be done as in [17, Proposition 2].

## B.2 Relaxed Characteristic Rate Proof

Since the characteristic time $T^\star(M)$ is a non-convex optimization problem [16], we focus on deriving a convex upper bound of $T^\star(M)$ that is convex for a finite set of rewards (we discuss the continuous case in the next section). This property guarantees that we are still identifying the optimal policy at the cost of an increased over-exploration. In particular, we find a convex upper bound of $T(\omega; M)$ that holds for all possible allocations $\omega \in \Delta(\mathcal{S} \times \mathcal{A})$. Recall the definition of $T(\omega; M)$

$$
T(\omega; M)^{-1} := \inf_{r \in \mathcal{R}} \inf_{M' \in \mathrm{Alt}(M_r)} \left( \sum_{s,a} \omega(s,a)\mathrm{KL}_{M|M'}(s,a) \right).
$$

To find the upper bound, we proceed in 2 steps: (1) we simplify the set of confusing models; (2) we relate the KL-divergences in $T(\omega; M)$ to the sub-optimality gaps of the MDP, and formulate the problem of computing the relaxed characteristic rate as a convex problem.

### B.2.1 Step 1: simplifying the set of confusing models

We use the following lemma from [16, Lemma 2] to decompose the set of confusing models.

**Lemma B.1** (Decomposition lemma.). *We have that the set of confusing models for $r \in \mathcal{R}$ satisfies*

$$\mathrm{Alt}_r^{\pi_r^\star}(M) = \cup_s \cup_{a \neq \pi_r^\star(s)} \mathrm{Alt}_{sar}^{\pi_r^\star}(M_r). \tag{7}$$

*where $\mathrm{Alt}_{sar}(M) := \{M' \in \mathrm{Alt}_r^\pi(M) : Q_{M_r'}^\pi(s,a) > V_{M_r'}^\pi(s)\}$ is the set of confusing models for $(s, a, r, \pi)$.*

### B.2.2 Step 2: relating the KL divergence to the sub-optimality gaps

Using lem. B.1, we can convexify the characteristic rate and related the KL-divergence terms with the sub-optimality gaps $\Delta_r(s, a)$. The goal is to obtain an optimization problem where only the sub-optimality gaps depend on the reward function. We have the following result.

*Proof of thm. 3.2.* The proof follows similar steps to [16]. We begin by decomposing the set of confusing models. After that, we relate the sub-optimality gaps to the inner product between the transitions and the value functions. Using this relation, we can lower bound the KL terms in $T(\omega; M)$. Finally, through an optimization step, we establish the upper bound. In the proof, we indicate by $V_{M_r}^\pi$ the value of a policy $\pi$ in MDP $M_r$, and similarly $Q_{M_r}^\pi$.

**Decomposition.** Fix $r, s, a \neq \pi_r^\star(s)$. As in [16], we combine the condition $Q_{M_r'}^{\pi_r^\star}(s,a) > V_{M_r'}^{\pi_r^\star}(s)$ with the fact that $Q_{M_r}^\star(s,a) + \Delta_r(s,a) = V_{M_r}^\star(s)$ to obtain that

$$\Delta_r(s,a) < V_{M_r}^\star(s) - Q_{M_r}^\star(s,a) + Q_{M_r'}^{\pi_r^\star}(s,a) - V_{M_r'}^{\pi_r^\star}(s).$$

This is similar to Eq. (5) in [16]. Next, let $\Delta P(s,a) = P_{M'}(s,a) - P_M(s,a)$, where $P_M(s,a)$ (sim. $P_{M'}(s,a)$) is a vector of dimension $S$ representing the distribution of the next state given $(s,a)$. Further define the vector difference between the value in $M_r$ and $M_r'$: $\Delta V^{\pi_r^\star} = \left[ V_{M_r'}^{\pi_r^\star}(s_1) - V_{M_r}^\star(s_1) \quad \ldots \quad V_{M_r'}^{\pi_r^\star}(s_S) - V_{M_r}^\star(s_S) \right]^\top$. Then, letting $\mathbf{1}(s) = e_s$ be the unit vector with 1 in position $s$, we find

$$\Delta_r(s,a) < \gamma(P_{M'}(s,a)^\top V_{M_r'}^{\pi_r^\star} - P_M(s,a)^\top V_{M_r}^\star) - \mathbf{1}(s)^\top \Delta V^{\pi_r^\star},$$

$$= \gamma \Delta P(s,a)^\top V_{M_r}^\star + (\gamma P_{M'}(s,a) - \mathbf{1}(s))^\top \Delta V^{\pi_r^\star}. \tag{8}$$

Hence, the condition $Q_{M_r'}^{\pi_r^\star}(s,a) > V_{M_r'}^{\pi_r^\star}(s)$ is equivalent to the inequality in (8). This inequality only involves the pairs $\{(s,a), \cup_{s'}(s', \pi_r^\star(s'))\}$, and therefore the infimum over $\mathrm{Alt}_{sa}(M_r)$ of $\sum_{s,a} \omega(s,a) \mathrm{KL}_{M|M'}(s,a)$ is a model $M'$ satisfying $\mathrm{KL}_{M|M'}(s,a) = 0$ over all the other state-action pairs.

Hence, using lem. B.1, and the reasoning above, we can write

$$T(\omega; M)^{-1} = \min_{r \in \mathcal{R}} \min_{s, a \neq \pi_r^\star(s)} \inf_{M' \in \mathrm{Alt}_{sar}^{\pi_r^\star}(M)} \left[ \omega(s,a) \mathrm{KL}_{M|M'}(s,a) \right.$$

$$\left. + \sum_{s'} \omega(s', \pi_r^\star(s')) \mathrm{KL}_{M|M'}(s', \pi_r^\star(s')) \right].$$

**Sub-optimality gaps.** We now relate the KL-terms to the sub-optimality gaps, similarly as in [53, 16]. For any state $s$ and action $a \neq \pi_r^\star(s)$, we write each of the terms (8) as a fraction of $\Delta_r(s,a)$ using $\{\alpha_i\}_{i=1}^2$, which are real values satisfying $\sum_{i=1}^2 \alpha_i > 1$:

$$\begin{cases} \alpha_1 \Delta_r(s,a) = \gamma |\Delta P(s,a)^\top V_{M_r}^{\pi_r^\star}|, \\ \alpha_2 \Delta_r(s,a) = (\gamma P_{M'}(s,a) - \mathbf{1}(s))^\top \Delta V^{\pi_r^\star}. \end{cases}$$

For the first term, we obtain

$$(\alpha_1 \Delta_r(s,a))^2 = \gamma^2 |\Delta P(s,a)^\top (V_{M_r}^{\pi_r^\star} - \mathbb{E}_{s' \sim P(\cdot|s,a)}[V_{M_r}^{\pi_r^\star}(s')])|^2,$$
$$\leq \gamma^2 \|\Delta P(s,a)\|_1^2 \mathrm{MD}_r(s,a)^2,$$
$$\leq 2\gamma^2 \mathrm{KL}_{M|M'}(s,a)\mathrm{MD}_r(s,a)^2,$$

where we applied Pinsker inequality. Therefore, we can write

$$\mathrm{KL}_{M|M'}(s,a) \geq \alpha_1^2 \underbrace{\frac{\Delta_r(s,a)^2}{2\gamma^2 \mathrm{MD}_r(s,a)^2}}_{=:B_{1,r}(s,a)}$$

For the second term in (8) we have $\|\gamma P_{M'}(s,a) - \mathbf{1}(s)\|_1 = |\gamma P_{M'}(s|s,a) - 1| + \gamma(1 - P_{M'}(s|s,a)) \leq 1 + \gamma$ and $|\alpha_2 \Delta_r(s,a)| \leq (1+\gamma)\|\Delta V^\pi\|_\infty$. Following the same steps as in lem. B.3, we derive that

$$K_r(s,a,\alpha_2) \leq K_r, \tag{9}$$

where $K_r := \max_s \mathrm{KL}_{M|M'}(s,\pi_r^\star(s))$, $K_r(s,a,\alpha_2) = \max(K_{r,1}(s,a,\alpha_2), K_{r,2}(s,a,\alpha_2))$ and[1]

$$K_{r,1}(s,a,\alpha_2) := \min\left(\frac{(\alpha_2 \Delta_r(s,a))^2(1-\gamma)^2}{16\gamma^2 \mathrm{Var}_r(1+\gamma)^2}, \frac{|\alpha_2 \Delta_r(s,a)|^{4/3}(1-\gamma)^{4/3}}{6\gamma^{4/3}\mathrm{MD}_r^{4/3}(1+\gamma)^{4/3}}\right), \tag{10}$$

$$K_{r,2}(s,a,\alpha_2) := \min\left(\frac{(\alpha_2 \Delta_r(s,a))^2(1-\gamma)^3}{139(1+\gamma)^2}, \frac{|\alpha_2 \Delta_r(s,a)|(1-\gamma)^{5/2}}{17\gamma(1+\gamma)}, \frac{(|\alpha_2 \Delta_r(s,a)|(1-\gamma))^{4/3}}{14(1+\gamma)^{4/3}\mathrm{MD}_r^{4/3}}\right). \tag{11}$$

**Optimization.** Putting all the inequalities together, we find

$$T(\omega; M)^{-1} \geq \min_{r \in \mathcal{R}} \min_{s,a \neq \pi_r^\star(s)} \inf_{\alpha: \sum_i \alpha_i > 1} \alpha_1^2 \omega(s,a)B_{r,1}(s,a) + \min_{s'} \omega(s',\pi_r^\star(s'))K_r(s,a,\alpha_2).$$

As in [16], the above inequalities are satisfied also when $\alpha$ belongs to the simplex, and in particular we also have $\alpha_i^2 \leq \alpha_i^{4/3} \leq |\alpha_i|$, for $i \in \{1,2\}$. Therefore we can write

$$T(\omega; M)^{-1} \geq \min_{r \in \mathcal{R}} \min_{s,a \neq \pi_r^\star(s)} \inf_{\alpha \in \Sigma_2} \alpha_1^2 \omega(s,a)B_{r,1}(s,a)$$
$$+ \min_{s'} \alpha_2^2 \omega(s',\pi_r^\star(s')) \max(B_{r,2}(s,a), B_{r,3}(s,a)),$$

with

$$B_{r,2}(s,a) := \min\left(\frac{\Delta_r(s,a)^2(1-\gamma)^2}{16\gamma^2 \mathrm{Var}_r(1+\gamma)^2}, \frac{\Delta_r(s,a)^{4/3}(1-\gamma)^{4/3}}{6\gamma^{4/3}\mathrm{MD}_r^{4/3}(1+\gamma)^{4/3}}\right), \tag{12}$$

$$B_{r,3}(s,a) := \min\left(\frac{\Delta_r(s,a)^2(1-\gamma)^3}{139(1+\gamma)^2}, \frac{\Delta_r(s,a)(1-\gamma)^{5/2}}{17\gamma(1+\gamma)}, \frac{\Delta_r(s,a)^{4/3}(1-\gamma)^{4/3}}{14\mathrm{MD}_r^{4/3}(1+\gamma)^{4/3}}\right). \tag{13}$$

Optimizing for $\alpha$, we get

$$T(\omega; M)^{-1} \geq \min_{r \in \mathcal{R}} \min_{s,a \neq \pi_r^\star(s)} \left(\frac{1}{\omega(s,a)B_{r,1}(s,a)} + \frac{1}{\max_{s'} \omega(s',\pi_r^\star(s')) \max(B_{r,2}, B_{r,3})(s,a)}\right)^{-1}.$$

Then, using that $\Delta_r \leq \Delta_r(s,a)$ we have

$$T(\omega; M) \leq \max_{r \in \mathcal{R}} \max_{s,a \neq \pi_r^\star(s)} \frac{H_r(s,a)}{\omega(s,a)} + \frac{\min(H_{r,1}, H_{r,2})}{\min_{s'} \omega(s',\pi_r^\star(s'))},$$

---

[1]Here we have different constants compared to [16]. In that work, we believe the authors may have miscomputed the constants by using the logarithm in base 10 instead of the natural logarithm.

where

$$H_r(s,a) = \frac{2\gamma^2 \mathrm{MD}(s,a)^2}{\Delta_r(s,a)^2},$$

$$H_{r,1} = \max\left( \frac{16\gamma^2 \mathrm{Var}_r (1+\gamma)^2}{\Delta_r^2 (1-\gamma)^2}, \frac{6\gamma^{4/3} \mathrm{MD}_r^{4/3} (1+\gamma)^{4/3}}{\Delta_r^{4/3} (1-\gamma)^{4/3}} \right),$$

$$H_{r,2} = \max\left( \frac{139(1+\gamma)^2}{\Delta_r^2 (1-\gamma)^3}, \frac{17\gamma(1+\gamma)}{\Delta_r (1-\gamma)^{5/2}}, \frac{14\mathrm{MD}_r^{4/3} (1+\gamma)^{4/3}}{\Delta_r^{4/3} (1-\gamma)^{4/3}} \right).$$

Finally, as $\Delta_r < 1$ (see [16]) and $\mathrm{MD}_{\max} \le 1/(1-\gamma)$, we have that $H_{r,2} = \frac{139(1+\gamma)^2}{\Delta_r^2 (1-\gamma)^3}$. Henceforth, we can derive the following simplified expression for the upper bound

$$T(\omega; M) \le \max_{r \in \mathcal{R}} \max_{s, a \ne \pi_r^\star(s)} \frac{2\gamma^2 \mathrm{MD}_r(s,a)^2}{\Delta_r(s,a)^2 \omega(s,a)} + \frac{H_r}{\Delta_r^2 \min_{s'} \omega(s', \pi_r^\star(s'))},$$

with $H_r = \min\left( \frac{139(1+\gamma)^2}{(1-\gamma)^3}, \max\left( \frac{16\gamma^2 \mathrm{Var}_r (1+\gamma)^2}{(1-\gamma)^2}, \frac{6\gamma^{4/3} \mathrm{MD}_r^{4/3} (1+\gamma)^{4/3}}{(1-\gamma)^{4/3}} \right) \right).$

$\square$

**Remark B.1.** *Note that one can improve the dependency in $\Delta_r^2$ in the second term of $U(\omega; M)$ by considering $\Delta_r(s,a)^2$ instead, as done in [53] (we did not for the sake of readability).*

### B.2.3 Technical lemmas

In this subsection, we present and prove some lemmas useful for the proofs of previous results.

**Lemma B.2** (Lemma 4 [16]). *Consider two transition functions $P, P'$ and let $\Delta P(s,a) = P(\cdot|s,a) - P(\cdot|s,a)$. Then, $\forall r \in \mathcal{R}$, we have for the optimal value function $V_r^\star \in \mathbb{R}^S$ we have*

$$|\Delta P(s,a)^\top V_r^\star|^2 \le 8\mathrm{Var}_r(s,a)\mathrm{KL}(P(\cdot|s,a), P'(\cdot|s,a)) + 4\sqrt{2}\mathrm{MD}_r(s,a)^2 \mathrm{KL}(P(\cdot|s,a), P'(\cdot|s,a))^{3/2}.$$

*Therefore*

$$\min\left( \frac{|\Delta P(s,a)^\top V_r^\star|^2}{16\mathrm{Var}_r(s,a)}, \frac{|\Delta P(s,a)^\top V_r^\star|^{4/3}}{2^{7/3}\mathrm{MD}_r(s,a)^{4/3}} \right) \le \mathrm{KL}(P(\cdot|s,a), P'(\cdot|s,a)).$$

*Proof.* The first inequality follows from Lemma 4 in [16], and by noting that the variance/span terms are both convex functions in the reward for a fixed state-action pair. The second inequality comes from a simple application of the inequality $a + b \le 2\max(a,b)$ to the first inequality. $\square$

**Lemma B.3.** *Consider two MDPs $M, M'$ with transitions $P, P'$ and a reward vector $r \in \mathcal{R}$. Let $\Delta V_r^\pi = V_{M_r'}^\pi - V_{M_r}^\star$, where $\pi \in \Pi^\star(M_r)$ and $V_{M_r}^\star$ is the optimal value in $(M, r)$. Then*

$$\max(K_1, K_2) \le K_r, \tag{14}$$

*where $K_r := \max_{s, a \in \mathcal{O}_r(s)} \mathrm{KL}_{M|M'}(s,a)$, with $\mathcal{O}_r(s) = \{a : Q_{M_r}^\star(s,a) = V_{M_r}^\star(s)\}$ and*

$$K_1 := \min\left( \frac{\|\Delta V_r^\pi\|_\infty^2 (1-\gamma)^2}{16\gamma^2 \mathrm{Var}_r}, \frac{\|\Delta V_r^\pi\|_\infty^{4/3} (1-\gamma)^{4/3}}{6\gamma^{4/3}\mathrm{MD}_r^{4/3}} \right), \tag{15}$$

$$K_2 := \min\left( \frac{\|\Delta V_r^\pi\|_\infty^2 (1-\gamma)^3}{139}, \frac{\|\Delta V_r^\pi\|_\infty (1-\gamma)^{5/2}}{17\gamma}, \frac{\|\Delta V_r^\pi\|_\infty^{4/3} (1-\gamma)^{4/3}}{14\mathrm{MD}_r^{4/3}} \right). \tag{16}$$

*Proof.* First, note that $V_{M_r}^\star = V_{M_r}^\pi$, and hence, $\Delta V_r^\pi = V_{M_r'}^\pi - V_{M_r}^\pi$. Since the reward is the same, we have that

$$\Delta V_r^\pi = \gamma(P')^\pi \Delta V_r^\pi + \gamma\Delta P^\pi V_{M_r}^\pi,$$

where $\Delta P^\pi = (P')^\pi - P^\pi$. Hence, $\|\Delta V_r^\pi\|_\infty \le \frac{\gamma}{1-\gamma}\|\Delta P^\pi V_{M_r}^\pi\|_\infty$. Following the same logic as in lem. B.2, and upper-bounding over all states and optimal actions we obtain

$$\|\Delta V_r^\pi\|_\infty^2 \le \frac{4\gamma^2}{(1-\gamma)^2}(2\mathrm{Var}_r K_r + \sqrt{2}\mathrm{MD}_r^2 K_r^{3/2}),$$

where

$$K_r := \max_{s,a \in \mathcal{O}_r(s)} \mathrm{KL}_{M|M'}(s,a),$$

$$\mathrm{Var}_r := \max_{s,a \in \mathcal{O}_r(s)} \mathrm{Var}_r(s,a),$$

$$\mathrm{MD}_r := \max_{s,a \in \mathcal{O}_r(s)} \mathrm{MD}(s,a).$$

Therefore

$$K_1 := \min\left(\frac{\|\Delta V_r^\pi\|_\infty^2(1-\gamma)^2}{16\gamma^2\mathrm{Var}_r}, \frac{\|\Delta V_r^\pi\|_\infty^{4/3}(1-\gamma)^{4/3}}{2^{7/3}\gamma^{4/3}\mathrm{MD}_r^{4/3}}\right) \leq K_r,$$

and note that $2^{7/3} < 6$. Secondly, we can also write $\Delta V_r^\pi = [(I-\gamma(P')^\pi)^{-1} - (I-\gamma P^\pi)^{-1}]r$. Using Lem. 5 in [16], we derive

$$\|\Delta V_r^\pi\|_\infty \leq \sqrt{\frac{32\log(2)^2 K_r}{(1-\gamma)^3}} + \frac{8\log(2)\gamma K_r}{(1-\gamma)^{5/2}} + \frac{2^{5/4}K_r^{3/4}\mathrm{MD}_r}{(1-\gamma)}.$$

Using the fact that $a + b + c \leq 3\max(a,b,c)$, we find the following lower bound on $K_r$

$$K_2 := \min\left(\frac{\|\Delta V_r^\pi\|_\infty^2(1-\gamma)^3}{288\log(2)^2}, \frac{\|\Delta V_r^\pi\|_\infty(1-\gamma)^{5/2}}{24\gamma\log(2)}, \frac{\|\Delta V_r^\pi\|_\infty^{4/3}(1-\gamma)^{4/3}}{3^{4/3}\cdot 2^{5/3}\mathrm{MD}_r^{4/3}}\right) \leq K_r,$$

and note that $288\log(2)^2 < 139$ (natural logarithm), $24\log(2) < 17$ (natural logarithm) and $3^{4/3} \cdot 2^{5/3}\mathrm{MD}_r^{4/3} < 14$. Combining the two inequalities $K_1 \leq K_r$ and $K_2 \leq K_r$, we conclude that

$$\max(K_1, K_2) \leq K_r.$$

$\square$

### B.2.4    A Closed Form Solution for the Generative Case

As a corollary, we find an upper bound on $U(\omega; M)$ that admits a closed form solution in the generative case ($\omega \in \Omega(\mathcal{S} \times \mathcal{A})$). Although this solution is not optimal (i.e., it does not attain the optima of $U(\omega; M)$ under the constraint $\omega \in \Omega(M)$), we can consider it as a computational-efficient approximation of the optimal solution.

**Corollary B.1.** *Consider $\xi \in \Delta(\mathcal{S} \times \mathcal{A} \times \mathcal{R})$. We have that an approximate solution to $\arg\inf_{\omega\in\Delta(\mathcal{S}\times\mathcal{A})} U(\omega; M)$ is given by*

$$\xi(s,a,r) \propto \begin{cases} H_r(s,a) & a \neq \pi_r^\star(s), \\ \sqrt{H^\star \sum_{s\in\mathcal{S}, r\in B, a\neq\pi_r^\star(s)} H_r(s,a)/(S\cdot|\mathcal{R}|)} & \text{otherwise,} \end{cases} \tag{17}$$

*with $\omega(s,a) = \sum_{r\in\mathcal{R}} \xi(s,a,r)$, $H_r(s,a) = 2\gamma^2\mathrm{MD}_r(s,a)^2/\Delta_r(s,a)^2$, $H^\star = \max_{r\in\mathcal{R}} H_r/\Delta_r^2$.*

*Proof.* We first derive an upper bound of $U(\omega; M)$ and then show the it can be solved in closed form.

**Upper bound on $U(\omega; M)$.**    Introduce the variable $\xi(s,a,r)$ such that $\omega(s,a) = \sum_{r\in\mathcal{R}} \xi(s,a,r)$, for all $(s,a) \in \mathcal{S} \times \mathcal{A}$. By rewriting $U(\omega; M)$ as a function of $\xi$, we have

$$
\begin{aligned}
U(\xi; M) &= \max_{r\in\mathcal{R}} \max_{s,a\neq\pi_r^\star(s)} \frac{2\gamma^2\mathrm{MD}_r(s,a)^2}{\Delta_r(s,a)^2\sum_{r'}\xi(s,a,r')} + \frac{H_r}{\Delta_r^2\sum_{r'}\min_{s'}\xi(s',\pi_r^\star(s'),r')}, \\
&\leq \max_{r\in\mathcal{R}} \max_{s,a\neq\pi_r^\star(s)} \frac{2\gamma^2\mathrm{MD}_r(s,a)^2}{\Delta_r(s,a)^2\xi(s,a,r)} + \frac{H_r}{\Delta_r^2\min_{s'}\xi(s',\pi_r^\star(s'),r)}, \\
&\leq \max_{r\in\mathcal{R}} \max_{s,a\neq\pi_r^\star(s)} \frac{2\gamma^2\mathrm{MD}_r(s,a)^2}{\Delta_r(s,a)^2\xi(s,a,r)} + \frac{H^\star}{\min_{s'}\xi(s',\pi_r^\star(s'),r)}, \\
&= \max_{r\in\mathcal{R}} \max_{s,a\neq\pi_r^\star(s)} \frac{H_r(s,a)}{\xi(s,a,r)} + \frac{H^\star}{\min_{s',r'}\xi(s',\pi_{r'}^\star(s'),r')},
\end{aligned}
$$

where we used that $1/(x+y) \leq 1/x$ for $x, y > 0$ in the first inequality and we bounded the second term using $H^\star = \max_{r \in \mathcal{R}} H_r / \Delta_r^2$ in the second inequality. Finally, to lighten the notation, we let $H_r(s,a) = 2\gamma^2 \mathrm{MD}_r(s,a)^2 / \Delta_r(s,a)^2$.

**Closed-form solution.** We now show that the problem at the last line of the previous equation can be solved in closed form. For the sake of notation, in the proof we denote by $\mathcal{O}_r(s) = \{a : Q^\star_{M_r}(s,a) = V^\star_{M_r}(s)\}$ the set of optimal actions in state $s$ for the MDP $M_r$.

To distinguish $\xi$ when the action is optimal, or not, we write $\xi_{sra}$ if $a \in \mathcal{O}_r(s)$ and $\xi(s,a,r)$ otherwise. After introducing the auxiliary variables $t, p \geq 0$ the optimization problem can be equivalently rewritten as

$$\min_{\xi,t,p} \quad t + p$$

$$\text{s.t.} \quad t \geq \frac{H_r(s,a)}{\xi(s,a,r)} \quad s,r,a \notin \mathcal{O}_r(s),$$

$$p \geq \frac{H^\star}{\xi_{sra}} \quad s,r,a \in \mathcal{O}_r(s), \tag{18}$$

$$\sum_{s,a,r} \xi(s,a,r) = 1,$$

$$\xi(s,a,r) > 0 \quad \forall s,a,r.$$

The Lagrangian associated to (18) is then defined as

$$\mathcal{L} = t + p + \sum_{s,r,a \in \mathcal{O}_r(s)} \theta_{sra}\left(\frac{H^\star}{\xi_{sra}} - t\right) + \sum_{s,r,a \notin \mathcal{O}_r(s)} \mu_{sra}\left(\frac{H_r(s,a)}{\xi(s,a,r)} - p\right)$$

$$+ \lambda\left(\sum_{s,a,r} \xi(s,a,r) - 1\right) + \sum_{s,a,r} \nu_{sra}\xi(s,a,r),$$

where $\theta, \mu, \nu$ are the Lagrange multipliers. Through the KKT conditions one derives $\nu_{sra} = 0$ for every $(s,a,r)$, and

$$\frac{\partial \mathcal{L}}{\partial t} = 0 \Rightarrow \sum_{s,r,a \notin \mathcal{O}_r(s)} \mu_{sra} = 1,$$

$$\frac{\partial \mathcal{L}}{\partial p} = 0 \Rightarrow \sum_{s,r,a \in \mathcal{O}_r(s)} \theta_{sra} = 1,$$

$$\frac{\partial \mathcal{L}}{\partial \xi(s,a,r)} = 0 \Rightarrow \lambda - \mu_{sra}\frac{H_r(s,a)}{\xi(s,a,r)^2} = 0, \quad s,r,a \notin \mathcal{O}_r(s),$$

$$\frac{\partial \mathcal{L}}{\partial \xi_{sra}} = 0 \Rightarrow \lambda - \theta_{sra}\frac{H^\star}{\xi_{sra}^2} = 0, \quad s,r,a \in \mathcal{O}_r(s).$$

We further find that $\sum_{s,a,r} \xi(s,a,r) = 1, t = H_r(s,a)/\xi(s,a,r)$ for every $s,r,a \notin \mathcal{O}_r(s)$ and $p = \frac{H^\star}{\xi_{sra}}$ for every $s,r,a \in \mathcal{O}_r(s)$. Hence it follows that all the $\xi_{sra}$ have the same value, that we denote by $\xi_0$.

From $\xi(s,a,r) = H_r(s,a)/t$, to find $t$, observe that $H_r(s,a) = t\xi(s,a,r) \Rightarrow \sum_{s,r,a \notin \mathcal{O}_r(s)} H_r(s,a) = t(1 - \xi_{sum})$, where $\xi_{sum} = \sum_{s,r,a \in \mathcal{O}_r(s)} \xi_0 = \xi_0|\mathcal{G}|$, with $\mathcal{G} = \{(s,r,a) : s \in \mathcal{S}, r \in \mathcal{R}, a \in \mathcal{O}_r(s)\}$. Thus

$$\xi(s,a,r)\lambda - \mu_{sra}t = 0 \Rightarrow t = (1 - \xi_{sum})\lambda.$$

Use the fact that $\sum_{s,r,a \notin \mathcal{O}_r(s)} H_r(s,a) = t(1 - \xi_{sum})$ to find

$$\lambda \sum_{s,r,a \notin \mathcal{O}_r(s)} H_r(s,a,r) = t^2.$$

From $\frac{\partial \mathcal{L}}{\partial \xi_{sra}} = 0$ find $\lambda = \theta_{sra}H^\star/\xi_{sra}^2$ and

$$\xi_{sra}\lambda = \theta_{sra}p \Rightarrow p = \xi_0|\mathcal{G}|\lambda.$$

Thus $\lambda = p/\xi_0 = H^\star/(\xi_0^2|\mathcal{G}|)$ and

$$\frac{H^\star}{\xi_0^2|\mathcal{G}|} \sum_{s,r,a \notin \mathcal{O}_r(s)} H_r(s,a) = t^2.$$

Therefore

$$\xi_0 = \frac{1}{t}\sqrt{\frac{H^\star}{|\mathcal{G}|} \sum_{s,r,a \notin \mathcal{O}_r(s)} H_r(s,a)}$$

Finally, to determine $t$, note that

$$\sum_{s,r,a} \xi(s,a,r) = 1 \Rightarrow \sum_{s,r,a \notin \mathcal{O}_r(s)} \frac{H_r(s,a)}{t} + |\mathcal{G}|\xi_0 = 1$$

Hence

$$t = \sum_{s,r,a \notin \mathcal{O}_r(s)} H_r(s,a) + \sqrt{|\mathcal{G}|H_\varepsilon \sum_{s,a \neq \pi^\star(s)} H_\varepsilon(s,a)}.$$

Therefore $\xi(s,a,r) \propto H_r(s,a)$ for all $s \in \mathcal{S}, r \in \mathcal{R}, a \notin \mathcal{O}_r(s)$ and $\xi(s,a,r) \propto \sqrt{H^\star \sum_{s,r,a \notin \mathcal{O}_r(s)} H_r(s,a)/|\mathcal{G}|}$ otherwise. The result in $\omega$ follows automatically from the definition $\omega(s,a) = \sum_{r \in \mathcal{R}} \xi(s,a,r)$. $\qquad\square$

## B.3 Extension to Random Rewards

In this appendix, we consider an extension of MR-BPI in which the observed rewards are random. We begin by motivating the setting, and comparing it to the setting used in the main part of the paper. Then, we introduce the model, and its assumptions. Finally, we extend the lower bound to this setting (app. B.3.1) and derive the corresponding convex upper bound (app. B.3.2).

**Setting motivation.** So far, we have discussed the setting where the rewards are (endogenous signals) specified by the user, rather than being provided by the environment (exogenous signals). This distinction is crucial because it influences how we model and interpret the reward signals. Endogenous rewards are typically deterministic, as they directly reflect the user's specific objectives (e.g., a fixed reward for achieving a particular goal, ensuring a consistent reinforcement signal). Conversely, rewards provided by the environment (exogenous) are typically variable and uncertain. These rewards may be modeled as random quantities with distribution conditioned on observed state-action pair, as described in the next paragraph.

**Model.** We shortly describe the MR-BPI model with random rewards, which uses a slightly different notation (w.r.t. sec. 2) to accommodate the random reward setting. In this section, we assume that the number of rewards is finite. Let $\mathcal{Q}$ be a set of distributions over real numbers conditioned on the state-action space. In the following, we use $i \in \{1, \ldots, |\mathcal{Q}|\}$ as an index for the elements of $\mathcal{Q}$, so that $q_i$ is the $i$-th reward distribution. The related problem specific-quantities and definitions are hence indexed by $i$, (instead of $r$ as in sec. 2). We also denote by $r_{q_i}(s,a)$ the expected reward for state $s$ and action $a$ (w.r.t. to the corresponding reward measure $q_i$). Similarly to the deterministic-reward setting, we assume that for all rewards $i \in \{1, \ldots, |\mathcal{Q}|\}$ the optimal policy $\pi_i^\star$ is unique.

**Agent interaction.** At each round $t \geq 1$, the agent in state $s_t$ selects an action $a_t$, and observes a set of rewards samples $(r_{i,t}(s_t, a_t))_{i \in |\mathcal{Q}|}$, where $r_{i,t}(s_t, a_t) \sim q_i(\cdot|s_t, a_t)$ and the rewards conditioned on the state-action are independent. It then transition to a new state $s_{t+1} \sim P(\cdot|s_t, a_t)$. and the interaction repeats. In this section, we denote by $\mathbb{P}$ the probability law (resp. expectation) of the process $(\zeta_t)_t$, where $\zeta_t = (s_t, a_t, r_{1,t}, \ldots, r_{|\mathcal{Q}|,t})$. We also denote by $\mathcal{F}_t = \sigma(\{\zeta_0, \ldots, \zeta_t\})$ the $\sigma$-algebra generated by the random observations made up to time $t$. For such algorithm, we define $\tau$ to be a stopping rule w.r.t. the filtration $(\mathcal{F}_t)_{t \geq 1}$. This stopping rule $\tau$ simply states when to stop the exploration phase. At the stopping time $\tau$ the algorithm outputs an estimate $\hat{\pi}_{\tau,r}$ of a best policy for any reward $r \in \mathcal{R}$ using the estimated MDP $M_\tau$. We focus on $\delta$-PC algorithms, according to the following definition, and we impose a few mild assumptions.

Similarly to the deterministic reward setting, the goal is to devise a $\delta$-PC algorithm with minimal sample complexity, according to the following definition

**Definition B.1** ($\delta$-PC Algorithm). We say that an algorithm is $\delta$-PC if, for any MDP $M \in \mathcal{M}$, $\mathbb{P}[\tau < \infty, (\forall i \in \{1, \ldots, |\mathcal{Q}|\}, \hat{\pi}_{\tau,i} = \pi_i^\star)] \geq 1 - \delta$.

### B.3.1 Lower bound with random rewards

**Theorem B.1.** *Let $\delta \in (0, 1/2)$. For any $\delta$-PC algorithm and under assumption* (3)*, we have* $\liminf_{\delta \to 0} \frac{\mathbb{E}[\tau]}{\log(1/\delta)} \geq T_{\mathcal{Q}}^\star(M)$*, where*

$$T_{\mathcal{Q}}^\star(M)^{-1} = \sup_{\omega \in \Omega(M)} \inf_{M' \in \mathrm{Alt}(M)} \sum_{s,a} \omega(s,a) \mathrm{KL}_{M|M'}^{\mathcal{Q}}(s,a), \tag{19}$$

*with* $\mathrm{KL}_{M,M'}^{\mathcal{Q}} = \mathrm{KL}_{P|P'}(s,a) + \sum_{i=1}^{|\mathcal{Q}|} \mathrm{KL}_{q_i|q_i'}(s,a)$ *where* $\mathrm{KL}_{P|P'}(s,a) = \mathrm{KL}(P(\cdot|s,a), P'(\cdot|s,a))$ *and* $\mathrm{KL}_{q_i|q_i'}(s,a) = \mathrm{KL}(q_i(\cdot|s,a), q_i'(\cdot|s,a))$.

*Proof.* The proof proceeds similarly to the one of the lower bound with deterministic rewards app. B.1. The main difference is that the log-likelihood ratio $L_t$ also includes the reward observations sampled from distributions in $\mathcal{Q}$.

*Log-likelihood ratio.* Let $Q \in \mathcal{Q}$, and consider the MDP $M_q = (\mathcal{S}, \mathcal{A}, P, q, \gamma)$ and an alternative model $M_q' = (S, A, P', q', \gamma)$ such that $P \ll P'$ and $q \ll q'$. The log-likelihood ratio between $M$ and $M'$ of a sequence of observations $(z_0, z_1, \ldots)$, where $z_i = (s_i, a_i, r_{i,1}, \ldots, r_{i,s})$, is defined as

$$L_t := \sum_{n=1}^{t} \left( \log \frac{P(s_n|z_{n-1})}{P'(s_n|z_{n-1})} + \sum_{i \in |\mathcal{Q}|} \log \frac{q_i(r_{i,n}|z_{n-1})}{q_i'(r_{i,n}|z_{n-1})} \right)$$

To simplify the analysis, let

$$\begin{aligned}
L_t^{s,a} &:= \sum_{n=1}^{t} \mathbf{1}_{\{s_n=s, a_n=a\}} \left( \log \frac{P(s_n|s,a)}{P'(s_n|s,a)} + \sum_{i=1}^{|\mathcal{Q}|} \log \frac{q_i(r_{i,n}|s,a)}{q_i'(r_{i,n}|s,a)} \right) \\
&= \sum_{n=1}^{N_t(s,a)} \left( \log \frac{P(y_n|s,a)}{P'(y_n|s,a)} + \sum_{i=1}^{|\mathcal{Q}|} \log \frac{q_i(x_{i,n}|s,a)}{q_i'(x_{i,n}|s,a)} \right),
\end{aligned}$$

where $y_n$ and $x_{i,n}$ denotes the next state and $i^{th}$ reward component collected after the pair $(s,a)$ has been visited for the $n$-th time $(s,a)$ and $N_t(s,a)$ is be the number of times the state-action pair $(s,a)$ has been visited by the algorithm up to time $t$. Then we can write $L_t = \sum_{s,a} L_t^{s,a}$.

Hence, at the stopping time $\tau$ we have

$$\begin{aligned}
\mathbb{E}_M[L_\tau] &= \sum_{s,a} \mathbb{E}_M \left[ L_\tau^{s,a} \right], \\
&= \sum_{s,a} \mathbb{E}_M \left[ \sum_{n=1}^{N_\tau(s,a)} \left( \log \frac{P(y_n|s,a)}{P'(y_n|s,a)} + \sum_{i=1}^{|\mathcal{Q}|} \log \frac{q_i(x_{i,n}|s,a)}{q_i'(x_{i,n}|s,a)} \right) \right], \\
&= \sum_{s,a} \mathbb{E}_M \left[ \sum_{t=0}^{\infty} \mathbf{1}_{\{N_\tau(s,a) \geq t\}} \left( \log \frac{P(y_t|s,a)}{P'(y_t|s,a)} + \sum_{i=1}^{|\mathcal{Q}|} \log \frac{q_i(x_{i,n}|s,a)}{q_i'(x_{i,n}|s,a)} \right) \right].
\end{aligned}$$

Since $W_n^{s,a} := \log \frac{P(y_n|s,a)}{P'(y_n|s,a)} + \sum_{i=1}^{|\mathcal{Q}|} \log \frac{q_i(x_{i,n}|s,a)}{q_i'(x_{i,n}|s,a)}$ is independent of $\mathcal{F}_{n-1}$, and since $\{N_\tau(s,a) \leq n-1\}^c = \{N_\tau(s,a) \geq n\} \in \mathcal{F}_{n-1}$ we have that

$$\begin{aligned}
\mathbb{E}_M[L_\tau] &= \sum_{s,a} \sum_{t=0}^{\infty} \mathbb{P}_M[N_\tau(s,a) \geq t] \mathrm{KL}_{M|M'}^{\mathcal{Q}}(s,a), \\
&= \sum_{s,a} \mathbb{E}_M[N_\tau(s,a)] \mathrm{KL}_{M|M'}^{\mathcal{Q}}(s,a),
\end{aligned}$$

where $\mathrm{KL}^{\mathcal{Q}}_{M,M'} = \mathrm{KL}(P(\cdot|s,a)|P'(\cdot|s,a)) + \sum_{i=1}^{|\mathcal{Q}|} \mathrm{KL}(q_i(\cdot|s,a), q'_i(\cdot|s,a))$.

*δ-PC argument.* Consider a δ-PC algorithm Alg, and define the event $\mathcal{E} = \cup_{i \in \{1,\dots|\mathcal{Q}|\}} \mathcal{E}_i$, where

$$\mathcal{E}_i = \{\hat{\pi}_{i,\tau} \neq \pi_i^\star\}.$$

Note that, since we are considering a δ-PC algorithm, we have

$$\mathbb{P}_M(\mathcal{E}) \leq \delta,$$

Let then $M' \in \mathrm{Alt}(M)$. Let $\mathrm{Alt}(M) = \{M' : \exists i \in 1, \dots, |\mathcal{Q}|, \Pi^\star(M_{q_i}) \cap \Pi^\star(M'_{q_i}) = \emptyset\}$, and note that we can write $\mathrm{Alt}(M) = \cup_{i \in \{1,\dots,|\mathcal{Q}|\}} \mathrm{Alt}_i(M_{q_i})$, where $\mathrm{Alt}_i(M_{q_i}) = \{M' : \Pi^\star(M_{q_i}) \cap \Pi^\star(M'_{q_i}) = \emptyset\}$. Furthermore, we have that

$$\mathbb{P}_{M'}(\mathcal{E}) \geq 1 - \delta.$$

In view of the transportation lemma in [82], we can lower bound the previous quantity at the stopping time τ as

$$\sum_{s,a} \mathbb{E}_M[N_\tau(s,a)]\mathrm{KL}^{\mathcal{Q}}_{M|M'}(s,a) \geq \mathrm{kl}(\mathbb{P}_M[\mathcal{E}], \mathbb{P}_{M'}[\mathcal{E}]) \geq \mathrm{kl}(\delta, 1-\delta).$$

As the inequality above holds for all $M' \in \mathrm{Alt}(M)$, by optimizing over the set of confusing model, we have

$$\inf_{M' \in \mathrm{Alt}(M)} \sum_{s,a} \mathbb{E}_M[N_\tau(s,a)]\mathrm{KL}^{\mathcal{Q}}_{M|M'}(s,a) \geq \mathrm{kl}(\delta, 1-\delta).$$

*Sample complexity.* By taking the minimum over the set of rewards, letting $\omega(s,a) = \mathbb{E}_M[N_\tau(s,a)]/\mathbb{E}_M[\tau]$, and optimizing over all possible allocations $\omega \in \Omega$, we have

$$\mathbb{E}_M[\tau] \sup_{\omega \in \Delta(\mathcal{S} \times \mathcal{A})} \inf_{M' \in \mathrm{Alt}(M)} \sum_{s,a} \omega(s,a)\mathrm{KL}^{\mathcal{Q}}_{M|M'}(s,a) \geq \mathrm{kl}(\delta, 1-\delta).$$

Finally, taking the limit as $\delta \to 0$ yields the result. $\square$

### B.3.2 Relaxed characteristic rate with random rewards

**Theorem B.2.** *For any allocation $\omega \in \Delta(\mathcal{S} \times \mathcal{A})$ we have that $T_{\mathcal{Q}}(\omega; M) \leq U_{\mathcal{Q}}(\omega; M)$, where*

$$U_{\mathcal{Q}}(\omega; M) := \max_{i,s,a \neq \pi_i^\star(s)} \frac{2(\gamma^2 \mathrm{MD}_i(s,a)^2 + 1)}{\Delta_i(s,a)^2 \omega(s,a)} + \frac{H_i + 2/(1-\gamma)^2}{\Delta_i^2 \min_{s'} \omega(s', \pi_i^\star(s'))}, \tag{20}$$

*and $H_i = \min\left(\frac{139}{\Delta_i^2(1-\gamma)^3}, \max\left(\frac{16\mathrm{Var}_i}{\Delta_i^2(1-\gamma)^2}, \frac{6\mathrm{MD}_i^{4/3}}{\Delta_i^{4/3}(1-\gamma)^{4/3}}\right)\right)$.*

*Proof.* The proof follows similarly to the one of thm. 3.2 but it uses a different decomposition of the analyic form ([16, Thm. 2]). We sketch its main steps below.

*Decomposition.* We first state a decomposition result on the set of confusing parameters similar to the one presented in lem. B.1 for deterministic rewards.

**Lemma B.4** (Decomposition lemma random rewards.). *We have that the set of confusing models, for all $i = 1, \dots, |\mathcal{Q}|$, satisfies*

$$\mathrm{Alt}_i(M_{q_i}) \subseteq \cup_{s,a \neq \pi_i^\star} \mathrm{Alt}^{\pi_i^\star}_{sai}(M_{q_i}).$$

*where $\mathrm{Alt}^{\pi}_{sai}(M_{q_i}) := \{M' \in \mathrm{Alt}_i(M_{q_i}) : Q^\pi_{M'_i}(s,a) > V^\pi_{M'_i}(s)\}$ is the set of confusing models for $(s,a,i,\pi)$.*

**Remark B.2.** *Note that $T^\star_{\mathcal{Q}}$ can be equivalently rewritten using thm. B.2 as*

$$(T^\star_{\mathcal{Q}})^{-1} = \sup_{\omega \in \Omega(M)} \min_{i,s,a \neq \pi_i^\star(s)} \inf_{M' \in \mathrm{Alt}_{sai}(M_{q_i})} \sum_{s',a'} \omega(s',a') \left( \mathrm{KL}_{P|P'}(s',a') + \sum_j \mathrm{KL}_{q_j|q'_j}(s',a') \right),$$

*Lower bound analytic form.* We now rewrite the lower bound optimization problem analytically in a similar form to (5) in [16]. For all $s, a, i$ let $dr_i(s,a) = r_{q_i}(s,a) - r_{q_i'}(s,a)$ and $dP(s,a) = P_{M'}(s,a) - P_M(s,a)$. Further define the vector difference between the value in $M_i$ and $M_i'$:
$dV_i^\pi = \left[V_{M_i'}^{\pi_i^\star}(s_1) - V_{M_i}^\star(s_1) \quad \dots \quad V_{M_i'}^{\pi_i^\star}(s_S) - V_{M_i}^\star(s_S)\right]^\top$. Finally, we let $\mathbf{1}(s) = e_s$ be the unit vector with 1 in position $s$.

Fix $i, s, a \neq \pi_i^\star(s)$. Then, we find that $M' \in \text{Alt}_{s,a,i}^{\pi_i^\star}(M_i)$ if and only if

$$dr_i(s,a) + \Delta_i(s,a) < \gamma dP(s,a)^\top V_{M_i}^\star + (\gamma P_{M_i'}(s,a) - \mathbf{1}(s))^\top \left(I - \gamma P_{\pi_i^\star}'\right)^{-1} (r_{\pi_i^\star}' - r_{\pi_i^\star})$$
$$+ (\gamma P_{M_i'}(s,a) - \mathbf{1}(s))^\top \left((I - \gamma P_{\pi_i^\star}')^{-1} - (I - \gamma P_{\pi_i^\star})^{-1}\right)^{-1} r_{\pi_i^\star}).$$

Hence, the condition $Q_{M_{q_i}'}^\pi(s,a) > V_{M_{q_i}'}^\pi(s)$ is equivalent to the above inequality. This inequality only involves the $\{(s,a,i), \cup_{s'}(s', \pi_i^\star(s'))\}$, and therefore the infimum over $\text{Alt}_{sai}(M_i)$ of $\sum_{s,a} \omega(s,a)\text{KL}_{M|M'}^\mathcal{Q}(s,a)$ is a model $M'$ satisfying $\text{KL}_{M|M'}^\mathcal{Q}(s,a) = 0$ over all the other state-action pairs.

Hence, using lem. B.1, and the reasoning above, we have

$$T_\mathcal{Q}(\omega; M)^{-1} = \min_{i,s,a \neq \pi_i^\star(s)} \inf_{M' \in \text{Alt}_{sai}^\pi(M)} \omega(s,a)\text{KL}_{M|M'}^i(s,a)$$
$$+ \sum_{s'} \omega(s', \pi_i^\star(s')) \left[\text{KL}_{P|P'}(s', \pi_i^\star(s'))) + \text{KL}_{q_i|q_i'}(s', \pi_i^\star(s')))\right]$$

*Sub-optimality gaps.* We now relate the KL-terms to the sub-optimality gaps, similarly as in [53, 16]. By following a similar approach to [16, Thm. 2] and the proof of the relaxed characteristic rate in app. B.2 one can show that, for any $\alpha \in \Sigma_4$, we have

$$\begin{cases} \text{KL}_{q_i|q_i'}(s,a) \geq B_{i,1}(\alpha_1) := \alpha_1^2 \frac{\Delta_i^2(s,a)}{2} \\ \text{KL}_{P,P'}(s,a) \geq B_{i,2}(\alpha_2) := \alpha_2^2 \frac{\Delta_i(s,a)^2}{2\gamma^2 \text{MD}_i(s,a)^2} \\ \max_{s'} \text{KL}_{q_i|q_i'}(s', \pi_i^\star(s')) \geq B_{i,3}(\alpha_3) := \alpha_3^2 \frac{(\Delta_i(1-\gamma))^2}{2} \\ \max_{s'} \text{KL}_{P|P}(s', \pi_i^\star(s')) \geq B_{i,4}(\alpha_4) := \min\left(\frac{[\alpha_4 \Delta_i(1-\gamma)]^2}{16\text{Var}_i}, \frac{[\alpha_4\Delta_i(1-\gamma)]^{4/3}}{2^{7/3}\text{MD}_i^{4/3}}\right) \\ \max_{s'} \text{KL}_{P|P}(s', \pi_i^\star(s')) \geq B_{i,5}(\alpha_4) := \min\left(\frac{\alpha_4^2\Delta_i^2(1-\gamma)^3}{288 \log(2)^2}, \frac{\alpha_4\Delta_i^2(1-\gamma)^{5/2}}{24\log(2)}, \frac{\alpha_4^{4/3}\Delta_i^{4/3}(1-\gamma)^{4/3}}{2^{5/3}3^{4/3}\text{MD}_i}\right) \end{cases}$$

*Optimization.* Putting all the inequalities together, we get

$$T_\mathcal{Q}(\omega; M)^{-1} \geq \min_{i,s,a \neq \pi_i^\star(s)} \inf_{\alpha \in \Sigma_4} \omega(s,a)(B_{i,1}(\alpha_1) + B_{i,2}(\alpha_2))$$
$$+ \min_s \omega(s, \pi_i^\star(s))(B_{i,3} + \max(B_{i,4}(\alpha_4), B_{i,5}(\alpha_4))).$$

Then, proceeding as in thm. 3.2, and optimizing for $\alpha \in \mathbb{R}_{\geq 0}^4$ such that $\sum_i \alpha_i = 1$ yields the result. $\qquad \square$

## B.4 Extension to a Continuous Set of Rewards

In this section we consider the extension to a continuous set of rewards [2]. We briefly motivate the setting, and then explain what are the challenges.

**Setting motivation.** So far, we have discussed the case where the set of rewards $\mathcal{R}$ is finite. However, it is also useful to compare with classical reward-free techniques, which do not rely on

---

[2] In this context, by a continuous set, we are not necessarily referring to a connected set in $\mathbb{R}^n$, but rather to a set with a non-countable number of elements.

a pre-specified reward signal. Therefore, we extend our discussion to include a continuous set of rewards. This extension presents several challenges. First, it may not be feasible to assume that the optimal policy is unique for each reward. Second, the minimum gap can be non-convex and discontinuous, as we will discuss in the following section. However, we find that, *in practice*, it is sufficient to consider a finite set of rewards. This observation provides a basis for the implementation of a practical algorithm.

First, we address the challenge of optimizing the inverse of the sub-optimality gap over a continuous set of rewards. Later, we propose a practical solution by considering a finite set of rewards.

### B.4.1 Non-convexity of the minimum sub-optimality gap

The relaxed characteristic rate is mainly characterized by the minimum sub-optimality gap $\Delta_r = \min_{s,a \in \mathcal{O}_r^c(s)} \Delta_r$, where $\mathcal{O}_r(s) = \{a : Q_{M_r}^\star(s,a) = V_{M_r}^\star(s)\}$ is the set of optimal actions in $s$ for the MDP $M_r$. Unfortunately, even with a *convex set* of rewards, the minimum sub-optimality gap may be non-convex and possibly discontinuous. The discontinuity arises from the fact that in a certain state(s) all actions are optimal for specific rewards.

We illustrate this intuition with an example. Consider the MDP depicted in fig. 2 with states $\mathcal{S} = \{s_0, s_1\}$ and actions $\mathcal{A} = \{a_0, a_1\}$. For this MDP, we have $P(s_0|\{s_0, s_1\}, a_0) = 1, P(s_1|s_0, a_1) = 0.3, P(s_0|s_0, a_1) = 0.7, P(s_1|s_1, a_1) = 0.3$ and $P(s_0|s_1, a_1) = 0.7$.

We use the following reward function, which parametrized by $\theta \in [0, 1]$, in vector form

$$
r = \begin{matrix} (s_0,a_0) \\ (s_0,a_1) \\ (s_1,a_0) \\ (s_1,a_1) \end{matrix} \begin{bmatrix} 0 \\ \theta \\ 0 \\ 1 - \theta \end{bmatrix} = \theta e_2 + (1 - \theta) e_4.
$$

Next, we derive the sub-optimality gaps $\Delta_\theta(s, a) = V_\theta^\star(s) - Q_\theta^\star(s, a)$ as a function of $\theta$.

**Optimal Value function $V_\theta^\star$ and policy $\pi_\theta^\star$.**   For a discount factor $\gamma = 0.5$ we obtain that

$$
V_\theta^\star(s_0) = \max \left( \underbrace{\frac{1}{2} V_\theta^\star(s_0)}_{\text{action } a_0}, \underbrace{\theta + \frac{7}{20} V_\theta^\star(s_0) + \frac{3}{20} V_\theta^\star(s_1)}_{\text{action } a_1} \right),
$$

$$
V_\theta^\star(s_1) = \max \left( \underbrace{\frac{1}{2} V_\theta^\star(s_0)}_{\text{action } a_0}, \underbrace{1 - \theta + \frac{7}{20} V_\theta^\star(s_0) + \frac{3}{20} V_\theta^\star(s_1)}_{\text{action } a_1} \right).
$$

One can immediately see that $a_1$ is optimal in $s_0$ for all $\theta$, while in $s_1$ we need to distinguish between the two cases.

If $a_0$ is optimal in $s_1$, then we find

$$
V_\theta^\star(s_0) = \frac{40}{23} \theta, \quad V_\theta^\star(s_1) = \frac{20}{23} \theta.
$$

Alternatively, if $a_1$ is optimal in $s_1$, then

$$
V_\theta^\star(s_0) = \frac{7}{5} \theta + \frac{3}{10}, \quad V_\theta^\star(s_1) = -\frac{3}{5} \theta + \frac{13}{10}.
$$

It is possible then to see that for $\theta < 23/26$ the optimal policy $\pi_\theta^\star = [a_1 \quad a_1]$ is to choose action $a_1$ in both states, while for $\theta > 23/26$ the optimal policy $\pi_\theta^\star = [a_1 \quad a_0]$ changes so that action $a_0$ is now optimal in state $s_1$. For $\theta = 23/26$ both actions $\{a_0, a_1\}$ are optimal in state $s_1$.

**Optimal Action-value function $Q_\theta^\star$.**   We distinguish two cases. For $\theta \geq 23/26$ we have

$$
Q_\theta^\star(s_0, a) = \begin{cases} \frac{20}{23} \theta & a = a_0 \\ \frac{40}{23} \theta & a = a_1 \end{cases}, \quad Q_\theta^\star(s_1, a) = \begin{cases} \frac{20}{23} \theta & a = a_0 \\ 1 - \frac{6}{23} \theta & a = a_1 \end{cases}.
$$

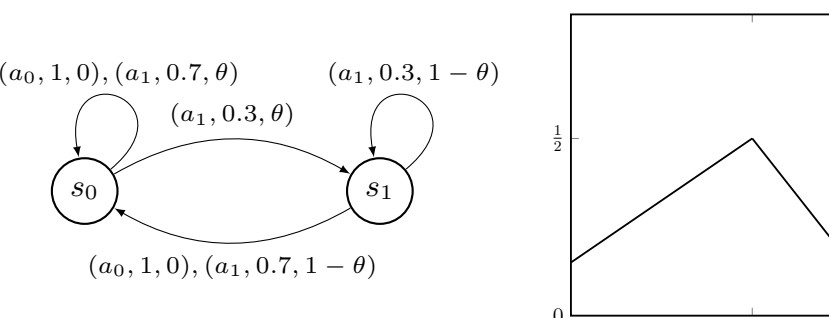

Figure 2: **On the left**: an MDP with two states and two actions; each tuple $(a, p, r)$ consists of (1) the action $a$ that triggers the transition, (2) the transition probability $p$ and (3) the reward $r$. **On the right**: the minimum sub-optimality gap $\Delta_\theta$ as a function of $\theta \in [0, 1]$ (note the discontinuity in $\theta = 23/26$).

While for $\theta < 23/26$ we have

$$Q_\theta^\star(s_0, a) = \begin{cases} \frac{7}{10}\theta + \frac{3}{20} & a = a_0 \\ \frac{7}{5}\theta + \frac{3}{10} & a = a_1 \end{cases}, \quad Q_\theta^\star(s_1, a) = \begin{cases} \frac{7}{10}\theta + \frac{3}{20} & a = a_0 \\ \frac{-3}{5}\theta + \frac{13}{10} & a = a_1 \end{cases}.$$

**Minimum sub-optimality gap.** Hence, we conclude that the minimum sub-optimality gap is

$$\Delta_\theta = \min_s \min_{a \notin \mathcal{O}_\theta(s)} \Delta_\theta(s, a) = \begin{cases} \frac{7}{10}\theta + \frac{3}{20} & \theta \leq \frac{1}{2}, \\ -\frac{13}{10}\theta + \frac{23}{20} & \frac{1}{2} \leq \theta < \frac{23}{26}, \\ \frac{20}{26} & \theta = \frac{23}{26}, \\ \frac{26}{23}\theta - 1 & \frac{23}{26} < \theta \leq 1. \end{cases}$$

Note that the minimum gap is non-convex on $\theta$ and it has a discontinuity at $\theta = 23/26$, as shown in fig. 2.

### B.4.2 A finite set of rewards can be sufficient

In this subsection, we derive a convex upper bounds on $T^\star(M)$ when $\mathcal{R}$ is a continuous set of rewards. The idea is to find a finite set of rewards $\mathcal{R}'$ so that we can still identify the optimal policies for all rewards in $\mathcal{R}$. The proof follows similar step to the one in the discrete reward setting in app. B.2 but relies on the following assumption.

**Convex hull (CH) assumption.** We assume that the convex hull[3] of $\mathcal{R}'$ contains $\mathcal{R}$. This last assumption is easy to satisfy, since one can just consider the canonical basis $\mathcal{R}_{\mathrm{canonical}}$ in $\mathcal{R}'$. Using this property, we derive a novel decomposition that leads to an alternative upper bound on $T(\omega; M)$.

**Theorem B.3.** *Consider a continuous set of rewards $\mathcal{R}$ satisfying the non-degeneracy assumption, and a finite set of rewards $\mathcal{R}'$. Assume that $\mathcal{R}'$ satisfies assumption (CH) with respect to $\mathcal{R}$. Then, for all $\omega \in \Omega(\mathcal{S} \times \mathcal{A})$ we have that*

$$T(\omega; M) \leq \min_{r \in \mathcal{R}'} \min_{s,a} \frac{8\gamma^2 \mathrm{MD}_r(s, a)^2}{\Delta_r^2(1 - \gamma)^2 \omega(s, a)}.$$

*Proof.* Using the decomposition in lem. B.5 in conjunction with the results in prop. B.1 and lem. B.6 leads to the desired result. Using also that for any $\omega \in \mathbb{R}^{SA}_{>0}$

$$\Delta_e^2 \leq 8 \frac{\gamma^2}{(1 - \gamma)^2} \max_{s,a} \mathrm{KL}_{M|M'}(s, a)\omega(s, a) \frac{\mathrm{MD}_e(s, a)^2}{\omega(s, a)},$$

$$\leq 8 \frac{\gamma^2}{(1 - \gamma)^2} \max_{s,a} \mathrm{KL}_{M|M'}(s, a)\omega(s, a) \max_{s',a'} \frac{\mathrm{MD}_e(s', a')^2}{\omega(s', a')}.$$

---

[3] Alternatively one may consider the conic hull of $\mathcal{R}'$, yielding a similar result

Then

$$T(\omega; M)^{-1} \geq \min_{r \in \mathcal{R}} \inf_{M' \in \mathrm{Alt}_r(M)} \left( \sum_{s,a} \omega(s,a) \mathrm{KL}_{M|M'}(s,a) \right),$$

$$\geq \min_{r \in \mathcal{R}} \min_{e \in \mathcal{R}'} \min_{s'} \inf_{M' \in \mathrm{Alt}^{\pi_r^\star}_{s', \pi_r^\star(s'), r, e}(M)} \left( \sum_{s,a} \omega(s,a) \mathrm{KL}_{M|M'}(s,a) \right),$$

$$\geq \min_{r \in \mathcal{R}} \min_{e \in \mathcal{R}'} \min_{s,a} \omega(s,a) \frac{\Delta_e^2 (1-\gamma)^2}{8\gamma^2 \mathrm{MD}_e(s,a)^2},$$

$$= \min_{e \in \mathcal{R}'} \min_{s,a} \omega(s,a) \frac{\Delta_e^2 (1-\gamma)^2}{8\gamma^2 \mathrm{MD}_e(s,a)^2}.$$

$\square$

While the previous bound in the worst case scenario scales as $(1-\gamma)^{-4}$, one can use techniques as in the proof of thm. 3.2 to find a scaling of $(1-\gamma)^{-3}$ (we refer the reader to the proof of prop. B.1 and note that we could use lem. B.3 to find an alternative scaling).

### B.4.3 Technical lemmas

**Confusing set of models decomposition: continuous set of rewards.** From the example in fig. 2 we know that an MDP may have a non-convex sub-optimality gap over a continuous set of rewards. Instead of considering the entire set of rewards, an alternative could be to just deal with a finite set of rewards $\mathcal{R}' \subset \mathcal{R}$.

We obtain a decomposition that depends on $\mathcal{R}'$ as long as every reward in $r \in \mathcal{R}$ can also be written as a convex combination (or conic combination) of the rewards $\mathcal{R}'$ (which can be guaranteed simply by including $\mathcal{R}_{\mathrm{canonical}}$ to $\mathcal{R}'$).

**Lemma B.5** (Decomposition lemma for a continuous set of rewards.)*. Consider a finite set of rewards $\mathcal{R}'$ such that $\mathcal{R} \subset \mathrm{Hull}(\mathcal{R}')$ and each $r \in \mathcal{R}'$ admits at-least a confusing model with respect to $M_r$ (as in assumption (III)). Then, we have that the set of confusing models for any $r \in \mathcal{R}$ satisfies*

$$\mathrm{Alt}_r(M) \subseteq \cup_{e \in \mathcal{R}'} \cup_{s, a \neq \pi_r^\star(s)} \mathrm{Alt}^{\pi_r^\star}_{sare}(M), \tag{21}$$

*where* $\mathrm{Alt}^{\pi}_{sare}(M) := \{M' : Q^\pi_{M_e'}(s,a) > V^\pi_{M_e'}(s)\}$.

*Proof.* Consider a reward $r \in \mathcal{R}$. From the fact that $\mathcal{R}$ is contained in a convex hull of $\mathcal{R}'$ we can always find $\theta \in [0,1]^{|\mathcal{R}'|}$ such that $r = \sum_{e \in \mathcal{R}'} \theta_e e$, where $\theta_e$ is the non-negative coefficient for the corresponding reward $e$ (satisfying $\sum_e \theta_e = 1$).

We start by showing that $\mathrm{Alt}_r(M) \subseteq \{M' | \exists e \in \mathcal{R}', s \in \mathcal{S}, a \neq \pi_r^\star(s) : Q^\pi_{M_e'}(s,a) > V^\pi_{M_e'}(s)\}$ with $\pi = \pi_r^\star$.

By contradiction, assume there exists $M' \in \mathrm{Alt}(M_r)$ so that $\forall e \in \mathcal{R}', s \in \mathcal{S}, a \neq \pi(s)$ the inequality $Q^\pi_{M_e'}(s,a) \leq V^\pi_{M_e'}(s)$ holds. Then, since $Q^\pi_{M_e}(s, \pi(s)) = V^\pi_{M_e}(s)$ the inequality $Q^\pi_{M_e'}(s,a) \leq V^\pi_{M_e'}(s)$ holds for every $e, s, a$.

By linearity, the inequality holds also if we multiply by $x \geq 0$ both sides. Choosing $x = \theta_e$, and summing over $e$, we obtain that

$$\sum_{e \in \mathcal{R}'} \theta_e Q^\pi_{M_e'}(s,a) \leq \sum_e \theta_e V^\pi_{M_e'}(s).$$

Letting $j_{s,a} \in \{1, \ldots, SA\}$ be the index corresponding to the state-action pair $(s,a)$ and $(P')^\pi$ be the transition function associated to policy $\pi$, i.e., $(P')^\pi(s', a'|s,a) = \pi(a'|s')P'(s'|s,a)$, we obtain that [4]

$$\sum_e \theta_i Q^\pi_{M_e'}(s,a) = \sum_e \theta_e e^\top_{j_{s,a}} (I - \gamma(P')^\pi)^{-1} e = e^\top_{j_{s,a}} (I - \gamma(P')^\pi)^{-1} r = Q^\pi_{M_r}(s,a).$$

---

[4]We indicate by $e_i$ the $i$-th element of the canonical basis.

Therefore, we conclude that $Q_{M'_r}^\pi(s, a) \le V_{M'_r}^\pi(s)$ is valid for all state-action pairs.

Let now $\hat\pi$ be an optimal policy in $M'_r$. Then, for all states we have $Q_{M'_r}^\pi(s, \hat\pi(s)) \le V_{M'_r}^\pi(s)$. Hence, as in [16, Lemma 2], by repeated applications of the Bellman operator $\Gamma^{\hat\pi}$ with respect to the MDP $M'_r$, one can conclude that $V_{M'_r}^\star(s) \le V_{M'_r}^\pi(s)$ in all states.

However, this implies that $\pi$ is optimal in $M'_r$, which contradicts that $M' \in \mathrm{Alt}(M_r)$. $\qquad\square$

**Remark B.3.** *While the previous lemma is stated for a convex hull of the vectors in $\mathcal{R}'$, it can be equivalently stated for a conic hull of $\mathcal{R}'$. This latter version can be used if, for example, one considers $\mathcal{R}_{\mathrm{canonical}}$ (note that the conic hull of the canonical basis contains all rewards in $[0, 1]^{SA}$).*

**Upper bound on the minimum sub-optimality gap.** In the following proposition, we derive a bound on the minimum sub-optimality gap.

**Proposition B.1.** *Consider two rewards $r, e$ and two MDPs $M, M'$ such that $\Pi^\star(M_r) \cap \Pi^\star(M'_e) = \emptyset$. Then*

$$\Delta_e \le \frac{2\gamma}{1 - \gamma} \|\Delta P(s, a)^\top V_{M_e}^\star\|_\infty, \qquad (22)$$

*where $\Delta P(s, a) = P_M(s, a) - P_{M'}(s, a)$, and $P_M(s, a)$ (sim. $P_{M'}(s, a)$) is a vector of size $S$ representing the distribution of the next state given $(s, a)$.*

*In particular we also have*

$$\Delta_e^2 \le 8 \frac{\gamma^2}{(1 - \gamma)^2} \max_{s,a} \mathrm{KL}_{M|M'}(s, a) \mathrm{MD}_e(s, a)^2.$$

*Proof.* The fact that $\Pi^\star(M_r) \cap \Pi^\star(M'_e) = \emptyset$ implies that in all states we have $0 \le V_{M_e}^\star(s) - V_{M_e}^\pi(s)$, for all $\pi \in \Pi^\star(M_r)$. Let $s_0$ be a state such that in $M_e$ there is a sub-optimal action. Henceforth, we have that $\Delta_e \le V_{M_e}^\star(s_0) - Q_{M_e}^\star(s_0, a)$ for some $a$. Hence

$$\Delta_e \le V_{M_e}^\star(s_0) - Q_{M_e}^\star(s_0, a) + V_{M'_e}^\star(s_0) - V_{M'_e}^\star(s_0),$$
$$\le \|V_{M_e}^\star - V_{M'_e}^\pi\|_\infty + \|Q_{M'_e}^\star - Q_{M_e}^\star\|_\infty.$$

Now we bound the two terms separately.

1. For the first term, let $\pi' \in \Pi^\star(M_e)$, and since $Q_{M_e}^\star(s, \pi'(s)) \ge Q_{M_e}^\star(s, \pi(s))$ write
$$[V_{M'_e}^\pi - V_{M_e}^\star](s) = Q_{M'_e}^\pi(s, \pi(s)) - Q_{M_e}^\star(s, \pi'(s)),$$
$$\le Q_{M'_e}^\pi(s, \pi(s)) - Q_{M_e}^\star(s, \pi(s)),$$
$$\le \|Q_{M'_e}^\pi - Q_{M_e}^\star\|_\infty.$$
Therefore $\|V_{M_e}^\star - V_{M'_e}^\pi\|_\infty \le \|Q_{M'_e}^\pi - Q_{M_e}^\star\|_\infty = \|Q_{M_e}^\star - Q_{M'_e}^\pi\|_\infty$. Hence, since
$$[Q_{M_e}^\star - Q_{M'_e}^\pi](s, a) = \gamma \left[ \Delta P(s, a)^\top V_{M_e}^\star + P_{M'}(s, a)^\top (V_{M_e}^\star - V_{M'_e}^\pi) \right],$$
we obtain
$$\|Q_{M_e}^\star - Q_{M'_e}^\pi\|_\infty \le \frac{\gamma}{1 - \gamma} \|\Delta P(s, a)^\top V_{M_e}^\star\|_\infty,$$
leading to $\|V_{M_e}^\star - V_{M'_e}^\pi\|_\infty \le \frac{\gamma}{1-\gamma} \|\Delta P(s, a)^\top V_{M_e}^\star\|_\infty$.

2. For the second term we have
$$[Q_{M_e}^\star - Q_{M'_e}^\star](s, a) = \gamma \left[ \Delta P(s, a)^\top V_{M_e}^\star + P_{M'}(s, a)^\top (V_{M_e}^\star - V_{M'_e}^\star) \right],$$
hence, as for the first term, we have $\|Q_{M'_e}^\star - Q_{M_e}^\star\|_\infty \le \frac{\gamma}{1-\gamma} \|\Delta P(s, a)^\top V_{M_e}^\star\|_\infty$.

Therefore we have that $\Delta_e \le \frac{2\gamma}{1-\gamma} \|\Delta P(s, a)^\top V_{M_e}^\star\|_\infty$.

Finally, an application of Pinsker inequality leads to

$$\Delta_e^2 \le 4 \frac{\gamma^2}{(1 - \gamma)^2} \max_{s,a} |\Delta P(s, a)^\top V_{M_e}^\pi|^2 \le 8 \frac{\gamma^2}{(1 - \gamma)^2} \max_{s,a} \mathrm{KL}_{M|M'}(s, a) \mathrm{MD}_e(s, a)^2.$$

$\qquad\square$

**Lemma B.6.** *Consider two MDPs $M, M'$ and two rewards $r, e$. If for all policies $\pi \in \Pi^\star(M_r)$ there exists a state-action pair $(s, a)$ such that $Q^\pi_{M'_e}(s, a) > V^\pi_{M'_e}(s)$, then $\Pi^\star(M_r) \cap \Pi^\star(M'_e) = \emptyset$.*

*Proof.* The proof simply follows by the fact that for all policies $\pi \in \Pi^\star(M_r)$ there exists a state $s$ where the policy is improvable. Therefore, none of the policies in $\Pi^\star(M_r)$ is optimal in $M'_e$. $\qquad \square$

# C  `MR-NaS` Algorithm

In this section we describe the `MR-NaS` algorithm in detail. We prove its $\delta$-PC property and its guarantees on the sample complexity.

**Notation.**  In addition to the notation introduction in sec. 2, in tab. 3 we provide a summary of the notation used in this section.

Table 3: Table of Notation

| Symbol | Description |
|---|---|
| $\mathcal{Z}$ | State-action space, $\mathcal{Z} = \mathcal{S} \times \mathcal{A}$. |
| $P_\pi$ | Transition induced by $\pi$, that is $P_\pi((s',a')\|(s,a)) = P(s'\|s,a)\pi(a'\|s')$. |
| $P_u$ | Transition induced by uniform policy. |
| $P_t$ | Transition induced by the policy $\pi_t$ of `MR-NaS`. |
| $M_t$ | Estimated MDP at time step $t$. |
| $\omega_t$ | Stationary distribution induced by the policy $\pi_t$. |
| $\omega_t^\star$ | Stationary distribution induced by the policy $\pi_t^\star$. |
| $\bar{\omega}_t^\star$ | Averaged stationary distribution $\bar{\omega}_t^\star := (1/t)\sum_{n=1}^{t} \omega_n^\star$. |
| $\omega_u$ | Stationary distribution induced by the uniform policy. |
| $\omega^\star$ | Optimal allocation $\omega^\star := \arg\inf_{\omega \in \Omega(M)} U(\omega; M)$ w.r.t. $U(\omega; M)$. |
| $\pi^\star(a\|s)$ | Oracle exploration policy, defined as $\pi^\star(a\|s) := \omega^\star(a\|s)/\sum_b \omega^\star(b\|s)$. |
| $\hat{\pi}_t(a\|s)$ | Exploration policy at time step $t$, defined as $\hat{\pi}_t^\star(a\|s) := \omega_t^\star(a\|s)/\sum_b \omega_t^\star(b\|s)$. |
| $\mu_t$ | Defined as $\mu_t := 1/N_t(s)^{\alpha+\beta}$. |
| $m_0$ | Maximum number of steps to move between any pair of states $(s, s')$. |
| $m$ | Maximum number of steps to move between any pair $(s, a), (s', a')$. Equal to $m = m_0 + 1$. |
| $\eta_k$ | Minimal probability of reaching $z'$ from $z$ in $n \leq k$ transitions using a uniform random policy, defined as $\eta_k := \min\{P_u^n(z'\|z)\|z, z' \in \mathcal{Z}, n \in \{1,\ldots,k\}, P_u^n(z'\|z) > 0\}$. |
| $\eta$ | Defined as $\eta := \eta_1 \eta_m$. |
| $B_\rho(P)$ or $(B_\rho(M))$ | Ball of radius $\rho$ around a transition function $P$, defined as $\{P' : \max_{s,a} \|P(\cdot\|s,a) - P'(\cdot\|s,a)\|_1 \leq \rho\}$. |
| $W_t$ | Rank-one matrix whose rows are equal to $\omega_t$. |
| $r_0$ | Ergodicity constant such that $P^r(z'\|z) > 0$ for all $r \geq r_0$. |
| $\sigma_u$ | Constant, defined as $\sigma_u := \min_{(z',z) \in \mathcal{Z}^2} P_u^{r_0}(z'\|z)/\omega_u(z')$. |
| $\sigma(\epsilon, \pi)$ | Defined as $\sigma(\epsilon, \pi) := (1-\epsilon)A\min_{s,a}\pi(\alpha\|s)^\tau + \epsilon_t^{\frac{r_0(\alpha+\beta)}{\alpha}}\sigma_u$. |
| $\theta(\epsilon, \pi)$ | $\theta(\epsilon, \pi) := 1 - \sigma(\epsilon, \pi)$ |
| $C(\epsilon, \pi)$ | $C(\epsilon, \pi) := 2/\theta(\epsilon, \pi)$ |
| $\rho(\epsilon, \pi)$ | $\rho(\epsilon, \pi) := \theta(\epsilon, \pi)^{1/r_0}$ |
| $L(\epsilon, \pi)$ | $L(\epsilon, \pi) := (1 - \rho(\epsilon, \pi))^{-1}C(\epsilon, \pi)$ |
| $\sigma_t, \theta_t, C_t\rho_t, L_t$ | Respectively defined as $\sigma_t := \sigma(\epsilon_t, \pi_t), \theta_t := \theta(\epsilon_t, \pi_t)$, etc. |
| $\mathcal{E}$ | Convergence of the MDP estimate and state-action visitation frequency event $\mathcal{E} :=$ $(\forall(s,a) \in \mathcal{Z}, \lim_{t\to\infty} N_t(s,a)/t = \omega^\star(s,a), \lim_{t\to\infty} M_t = M)$. |
| $\mathcal{E}_{\text{cnt}}(\lambda)$ | Count event, defined as $\mathcal{E}_{\text{cnt}}(\lambda) :=$ $\left\{\forall t \geq 1, \forall(s,a), N_t(s,a) \geq \frac{1}{2}\sum_{k=1}^{\lfloor \frac{t}{m} \rfloor} \frac{\eta_m}{(km)^{\alpha+\beta}} - \log\left(\frac{\|S\|\|A\|}{\lambda}\right)\right\}$. |
| $\mathcal{E}_{p,T}^1(\xi)$ | Concentration of the empirical MDP event $\mathcal{E}_{p,T}^1(\xi) = \cap_{t=h_p(T)}^{T}(M_t \in B_{\rho(\xi)}(M))$. |
| $\mathcal{E}_T^2(\xi)$ | Concentration of the state-action visitation frequency event $\mathcal{E}_T^2(\xi) := \cap_{t=T^{1-p}}^{T}\left(\|\frac{N_t}{t} - \omega^\star\| \leq K_\xi \xi\right)$. |

**Remark C.1** (Extension to the setting of continuous/stochastic rewards.)**.** *In this setting we consider the case with finite rewards. For the other two settings described in app. B (continuous and stochastic rewards),* `MR-NaS` *can be directly adapted by using the relaxed characteristic rate specified in app. B.3.2 for stochastic rewards and in app. B.4.2 for continuous rewards. The primary modification involves the computation of $\omega^\star$, where the relaxed characteristic rate appropriate for the respective setting should be applied. Furthermor, for stochastic rewards, the algorithm also needs to account for an additional exploration threshold in the stopping rule similarly to [17, Prop. 5].*

**Remark C.2** (Novelties with respect to previous works.)**.** *Our work is mostly inspired by [17]. With respect to previous works, we introduce a novel forcing policy that does not use an exploration factor $\epsilon_t$ that depends MDP-specific quantities, such as the "connectedness" of the MDP, that we denote by $m$. In [17] they use $\epsilon_t = t^{-\frac{1}{2(m+1)}}$, while we use $\epsilon_t = 1/\max(1, N_t(s_t))^\alpha$. This change requires a slight modification of the proofs in [17], since now $\epsilon_t$ depends also on the number of visits of $s_t$. Additionally, we improve on the high probability forced exploration [17, Lemma 11], where the authors have a scaling of $N_t(s,a) \overset{\sim}{\geq} \left(\frac{t\eta_m^2}{m^2\log^2(1+SA/\lambda)}\right)^{1/4}$ while we obtain a scaling $N_t(s,a) \overset{\sim}{\geq} \frac{\eta_m}{m^{\alpha+\beta}}H_{\lfloor\frac{t}{m}\rfloor,\alpha+\beta} - \log(SA/\lambda)$, where $H_{x,y}$ is the $x$-th generalized harmonic number of order $y$ and $\eta_m$ is the minimal probability of reaching a state-action pair in $m$ steps under a uniform random policy.*

## C.1 Forcing policy

We begin by providing some results on the forcing policy, that serve as a basis to obtain sample complexity results.

Consider the forcing policy $\pi_{f,t}(\cdot|s) = \texttt{softmax}\left(-\beta_t(s)N_t(s,\cdot)\right)$, which is inspired by the classical Boltzmann distribution. Intuitively this forcing policy will choose actions that have been under-sampled. We find the following lower bound on the probability of picking an action $a$ at time $t \geq 1$.

**Lemma C.1.** *Consider the forcing policy $\pi_{f,t}(\cdot|s) = \texttt{softmax}\left(-\beta_t(s)N_t(s,\cdot)\right)$ with $\beta_t(s) = \frac{\beta \log(N_t(s))}{\max_a |N_t(s,a)-\min_b N_t(s,b)|}$ and $\beta \in [0,1]$. Then, for all $t \geq 1$ we have*

$$\pi_{f,t}(a|s) \geq \frac{1}{AN_t(s)^\beta}. \tag{23}$$

*Proof.* Note that proving (23) is equivalent to showing that

$$AN_t(s)^\beta e^{-\beta_t(s)N_t(s,a)} \geq \sum_b e^{-\beta_t(s)N_t(s,b)}.$$

We actually show the following stronger result:

$$AN_t(s)^\beta e^{-\beta_t(s)N_t(s,a)} \geq Ae^{-\beta_t(s)\min_b N_t(s,b)} \geq \sum_b e^{-\beta_t(s)N_t(s,b)}.$$

Therefore, taking the logarithm on both hand-sides, we obtain

$$\beta \log(N_t(s)) \geq \beta_t(s)(N_t(s,a) - \min_b N_t(s)) = \frac{\beta \log(N_t(s))(N_t(s,a) - \min_b N_t(s)}{\max_a |N_t(s,a) - \min_b N_t(s,b)|}.$$

Thus, we obtain the condition

$$1 \geq \frac{N_t(s,a) - \min_b N_t(s)}{\max_a |N_t(s,a) - \min_b N_t(s,b)|},$$

which is always true for all $a \in \mathcal{A}$, proving the lemma. $\qquad\square$

## C.2 Almost Sure Sample Complexity Upper Bound of `MR-NaS`

We now describe the forced exploration property of `MR-NaS`. We conclude by showing that $N_t(s,a)/t \to \omega^\star(s,a)$ a.s., and the almost sure sample complexity bound of `MR-NaS`.

### C.2.1 Forced exploration property

The following proposition is a modification to [17, Lem. 4] which combines a result from [83].

**Proposition C.1** (Forced Exploration). *The sampling rule of `MR-NaS` with $\alpha \in (0,1], \beta \in [0,1]$ and $\alpha + \beta \leq 1$ satisfies $\mathbb{P}\left(\forall (s,a) \in \mathcal{S} \times \mathcal{A}, \lim_{t\to\infty} N_t(s,a) = \infty\right) = 1$.*

*Proof.* Recall that that $\epsilon_t = 1/N_t(s_t)^\alpha$ and $\pi_{f,t}(\cdot|s) = \texttt{softmax}\left(-\beta_t(s)N_t(s,\cdot)\right)$ with $\beta_t(s) = \frac{\beta \log(N_t(s))}{\max_a |N_t(s,a) - \min_b N_t(s,b)|}$. Therefore, using lem. C.1 and the fact that $\alpha + \beta \leq 1$ we have that

$$\sum_{n=1}^{\infty} \mathbb{P}(a_t = a | s_t = s, N_t(s) = n) \geq \sum_{n=1}^{\infty} \varepsilon_t \frac{1}{n^\beta A},$$
$$= \sum_{n=1}^{\infty} \frac{1}{A n^{\alpha+\beta}},$$
$$= \infty.$$

Thus if a state $s$ is visited infinitely often, by the Borel-Cantelli lemma we have that the action $a$ is also chosen infinitely often. Using [83, Lem. 4] we conclude that in a communicating MDP we have that $\mathbb{P}\left(\forall (s,a) \in \mathcal{Z}, \lim_{t\to\infty} N_t(s,a) = \infty\right) = 1$. $\square$

### C.2.2 Convergence of the state-action visitation frequency

Next, using the previous results we establish the convergence of $N_t(s,a)/t$. For the sake of compactness, we denote by $\mathcal{Z} = \mathcal{S} \times \mathcal{A}$ the set of state-action pairs, and by $P_\pi((s',a')|(s,a)) = P(s'|s,a)\pi(a'|s')$ the transition induced by $\pi$. Similarly, we denote by $P_u$ the transition induced by a uniform policy, and by $P_t$ the transition induced by the policy $\pi_t$ of `MR-NaS`. We also indicate by $\omega^\star = \arg\inf_{\omega \in \Omega(M)} U(\omega; M)$ the optimal allocation.

**Proposition C.2** (Almost sure convergence of the allocation). *Using `MR-NaS` guarantees that $\mathbb{P}\left(\forall (s,a) \in \mathcal{Z}, \lim_{t\to\infty} N_t(s,a)/t = \omega^\star(s,a)\right) = 1$, where $\omega^\star = \arg\inf_{\omega \in \Omega(M)} U(\omega; M)$.*

*Proof.* To show the result it suffices to ensure that `MR-NaS` satisfies the 5 assumptions of [16, Proposition 12] (see also [84, Corollary 2.9]). Assumption 1 is a consequence of the fact that the MDP is ergodic under any policy that assigns positive probability to all actions. Hence, also $P_t$ is ergodic. Assumption 2 is guaranteed by prop. C.1: since $N_t(s,a) \to \infty$ a.s. for all $(s,a) \in \mathcal{Z}$, then the estimate of the MDP converges $M_t \to M$ almost surely. By continuity, and the fact that $\bar{\omega}_t$ is a Cesàro mean, we also have $\omega_t^\star \to \omega^\star$ and $\bar{\omega}_t^\star \to \omega^\star$. Hence $P_t \to P_{\pi^\star}$ almost surely, where $\pi^\star(a|s) = \omega^\star(s,a)/\sum_{a'} \omega^\star(s,a')$. Assumption 3 and 4 are satisfied as in the proof of [17, Thm. 7]. In fact, by employing lem. C.1 we have that

$$P_t = (1 - \epsilon_t)P_{\pi_t^\star} + \epsilon_t P_{\pi_{f,t}} \geq (1 - \epsilon_t)P_{\pi_t^\star} + \mu_t P_u$$

with $\mu_t = 1/N_t(s)^{\alpha+\beta}$. Hence, one can follow the same reasoning of [17, Thm. 7], by also noting that $\mu_t \geq 1/t^{\alpha+\beta}$ for every $\geq 1$. Similarly, assumption 5 is satisfied by noting that $\omega_t^\star \to \omega^\star$ a.s., and for all $(s,a) \in \mathcal{Z}, \epsilon_t, \mu_t \to 0$ almost surely. $\square$

### C.2.3 Sample complexity upper bound

Next, using the forced exploration property, we prove that the stopping time is almost surely finite and an almost sure upper bound of the sample complexity of `MR-NaS` that holds asymptotically as $\delta \to 0$.

**Proof strategy.** The idea is to prove that $N_t(s,a)/t \to \omega^\star(s,a)$ and that the estimate $M_t \to M$. If $N_t(s,a) \to \infty$ a.s. (prop. C.1 ), the latter follows automatically, while the former is given by prop. C.2 (given prop. C.1 ). At this point, one can conclude by continuity of the relaxed characteristic rate over its arguments in the stopping rule.

**Theorem C.1** (Almost sure sample complexity bound). *`MR-NaS` with the stopping rule in props. 3.1 and C.2 guarantees that `MR-NaS` stops almost surely, and that $\mathbb{P}\left(\limsup_{\delta\to 0} \frac{\tau}{\log(1/\delta)} \leq U^\star(M)\right) = 1$.*

*Proof.* Consider the event

$$\mathcal{E} = \left( \forall (s,a) \in \mathcal{Z}, \lim_{t \to \infty} N_t(s,a)/t = \omega^\star(s,a), \lim_{t \to \infty} M_t = M \right).$$

By prop. C.1 we have that $\mathcal{E}$ holds almost surely. Hence, one can prove straightforwardly prove that MR-NaS stops almost surely. Under $\mathcal{E}$, by continuity, $\forall \lambda > 0, \exists t_\lambda \in \mathbb{N}$ such that $\forall t \geq t_\lambda$ we have $U(N_t/t; M_t)^{-1} \geq (1-\lambda)U(\omega^\star; M)^{-1} = (1-\lambda)U^\star(M)$.

Now, recall the definition of stopping threshold $\beta(N_t, \delta) = \log(1/\delta) + (S - 1)\sum_{s,a} \log\left( e\left[1 + \frac{N_t(s,a)}{S-1}\right]\right)$. Note that there exists $C > 0$ such that $\beta(N_t, \delta) \leq \log(Ct/\delta)$. Thus, we have that

$$\begin{aligned}
\tau &= \inf\{t \geq t_\lambda, tU(N_t/t; M_t)^{-1} \geq \beta(N_t, \delta)\}, \\
&\leq t_\lambda \wedge \{t \geq t_\lambda, t(1-\lambda)U^\star(M)^{-1} \geq \beta(N_t, \delta)\}, \\
&\leq t_\lambda \wedge \{t \geq t_\lambda, t(1-\lambda)U^\star(M)^{-1} \geq \log(Ct/\delta)\}.
\end{aligned}$$

Applying [19, Lem. 8] we get

$$\tau \leq \max\left( t_\lambda, \frac{U^\star(M)}{1-\lambda}\left[ \log\left( \frac{CU^\star(M)}{\delta(1-\lambda)} \right) + \sqrt{2\left( \log\left( \frac{CU^\star(M)}{\delta(1-\lambda)} \right) - 1 \right)} \right] \right).$$

Hence $\limsup_{\delta \to 0} \frac{\tau}{\ln(1/\delta)} \leq U^\star(M)/(1-\lambda)$. Letting $\lambda \to 0$ concludes the proof. $\quad\square$

## C.3 Expected Sample Complexity Upper Bound of MR-NaS

To derive a sample complexity upper bound in expectation of MR-NaS we define the following quantities:

1. Let $m_0 \in \mathbb{N}$ be the (maximum) number of steps to move between any pair of states $(s, s')$. Then we can move between any pair $(z, z')$ in $m_0 + 1$ steps at-most [17]. Hence, $m_0$ can be interpreted as a sort of metric measuring the difficulty of navigating the MDP. We also define $m := m_0 + 1$.

2. Let $\eta_k := \min\{P_u^n(z'|z)|z, z' \in \mathcal{Z}, n \in \{1, \ldots, k\}, P_u^n(z'|z) > 0\}$ be the minimal probability of reaching $z'$ from $z$ in $n \leq k$ transitions using a uniform random policy, with $k \in \mathbb{N}$.

3. We define by $B_\rho(P)$ the ball of radius $\rho$ around a transition function $P$, i.e., $B_\rho(P) := \{P' : \max_{s,a} \|P(\cdot|s,a) - P'(\cdot|s,a)\|_1 \leq \rho\}$. Since in the setting of this work an MDP is identified by its transition function, for the sake of notation we also interchengeably write $B_\rho(M)$ to indicate the ball of radius $\rho$ around $M$.

4. We denote by $\omega_t$ the stationary distribution induced by the policy $\pi_t$ and by $\omega_u$ the stationary distribution induced by the uniform policy. We also indicate by $W_t$ a rank-one matrix whose rows are equal to $\omega_t$.

5. From the ergodicity of $P_u$ there exists an integer $r_0$ such that $P_u^r(z'|z) > 0$ for all $r \geq r_0$ and all $(z', z) \in \mathcal{Z}^2$ (the existence of $r_0$ is guaranteed by [85, Proposition 1.7]).

6. The oracle exploration policy $\pi^\star(a|s) := \omega^\star(a|s)/\sum_b \omega^\star(b|s)$ and the exploration policy at time step $t$: $\hat{\pi}_t^\star(a|s) := \omega_t^\star(a|s)/\sum_b \omega_t^\star(b|s)$.

We begin by proving two independent results: (1) the geometric convergence of $P_t^n$ to $W_t$ in app. C.3.1, a result that is used to prove the concentration of the state-action visitation frequency; (2) the forced exploration property of MR-NaS in app. C.3.2, which is used to prove the concentration of the MDP estimate. This latter result is novel compared to previous works.

We subsequently present the concentration of the MDP estimate in app. C.3.3 and of the state-action visitation frequency in app. C.3.4. We conclude with the expected sample complexity of MR-NaS in app. C.3.5. Finally, at the end of the section, we present some technical lemmas as well as an alternative result to the forced exploration property presented in app. C.3.2.

### C.3.1 Geometric ergodicity of the sampling rule

In the following lemma we adapt a result from [17] to our forcing policy $\pi_{f,t}$. We also define the following quantities

$$\sigma(\epsilon, \pi) := \left( [(1-\epsilon)A \min_{s,a} \pi(a|s)]^{r_0} + \epsilon_t^{\frac{r_0(\alpha+\beta)}{\alpha}} \right) \sigma_u,$$

$$\theta(\epsilon, \pi) := 1 - \sigma(\epsilon, \pi),$$

$$C(\epsilon, \pi) := 2/\theta(\epsilon, \pi),$$

$$\rho(\epsilon, \pi) := \theta(\epsilon, \pi)^{1/r_0},$$

$$L(\epsilon, \pi) := (1 - \rho(\epsilon, \pi))^{-1} C(\epsilon, \pi).$$

where $\sigma_u := \min_{z,z'} P_u^{r_0}(z'|z)/\omega_u(z')$ and $r_0, \alpha, \beta$ are as before. For the sake of brevity, we let $\sigma_t = \sigma(\epsilon_t, \pi_t^\star), \theta_t = \theta(\epsilon_t, \pi_t^\star)$, and similarly for $C_t, \rho_t, L_t$.

**Lemma C.2** (Geometric ergodicity of the sampling rule). *For* `MR-NaS` *it holds that*

$$\forall n \geq 1, \|P_t^n - W_t\|_\infty \leq C_t \rho_t^n. \tag{24}$$

*Proof.* For two state-action pairs $z = (s, a), z' = (s', a')$ denote by $P_t^r(z'|z)$ the probability of reaching $z'$ from $z$ in $r$ steps. Then, for all $(z, z')$ we have

$$P_t^r \geq (1-\epsilon_t)^{r_0} P_{\pi_t^\star}^{r_0}(z'|z) + \epsilon_t^{r_0} P_{\pi_f,t}^{r_0}(z'|z),$$

$$\geq (1-\epsilon_t)^{r_0} P_{\pi_t^\star}^{r_0}(z'|z) + (1/N_t(s)^{\alpha+\beta})^{r_0} P_u^{r_0}(z'|z),$$

$$\geq (1-\epsilon_t)^{r_0} P_{\pi_t^\star}^{r_0}(z'|z) + \epsilon_t^{\frac{r_0(\alpha+\beta)}{\alpha}} P_u^{r_0}(z'|z).$$

Let $c_t = \min_{s,a} \pi_t^\star(a|s)$ and note that $P_{\pi_t^\star}(z'|z) \geq Ac_t P_u(z'|z)$ for all $(z, z')$.

Recall the definitions for the stationary distribution under the uniform policy by $\omega_u$, and $\sigma_u = \min_{z,z'} P_u^{r_0}(z'|z)/\omega_u(z')$. Then, we can rewrite the previous inequality as

$$P_t^r \geq \left( [(1-\epsilon_t)Ac_t]^{r_0} + \epsilon_t^{\frac{r_0(\alpha+\beta)}{\alpha}} \right) P_u^{r_0}(z'|z) \frac{\omega_u(z')}{\omega_u(z')},$$

$$\geq \left( [(1-\epsilon_t)Ac_t]^{r_0} + \epsilon_t^{\frac{r_0(\alpha+\beta)}{\alpha}} \right) \sigma_u \omega_u(z'),$$

$$\geq \left( [(1-\epsilon_t)Ac_t]^{r_0} + \epsilon_t^{\frac{r_0(\alpha+\beta)}{\alpha}} \right) \sigma_u \omega_u(z') \omega_t(z').$$

Then using [17, Lemma 21] we conclude that for all $n \geq 1$ it holds that:

$$\|P_t^n - W_t\|_\infty \leq 2\theta_t^{\frac{n}{r}-1}.$$

Hence $P_t$ also satisfies $\|P_t^n - W_t\|_\infty \leq C_t \rho_t^n$. $\qquad\square$

### C.3.2 Forced exploration with high probability

In the following lemma, we provide a novel result for the forced exploration property of `MR-NaS`, which relies on lem. C.7.

**Lemma C.3.** *For all* $\lambda \in (0, 1)$, $m := m_0 + 1$, *define the event*

$$\mathcal{E}_{\mathrm{cnt}}(\lambda) = \left\{ \forall t \geq 1, \forall (s, a) \in \mathcal{S} \times \mathcal{A}, N_t(s, a) \geq \frac{1}{2} \sum_{k=1}^{\lfloor \frac{t}{m} \rfloor} \frac{\eta_m}{(km)^{\alpha+\beta}} - \log\left(\frac{SA}{\lambda}\right) \right\}.$$

*Then, for* `MR-NaS` *with* $\alpha + \beta \leq 1$ *we have that* $\mathbb{P}\left( \overline{\mathcal{E}_{\mathrm{cnt}}(\lambda)} \right) \leq \lambda$.

*Proof.* Fix a state-action pair $z = (s, a)$ and let $Y_t = \mathbf{1}_{z_t = z}$ and $X_k = \mathbf{1}_{Y_{(k-1)m+1} + \cdots + Y_{km} > 0}$ for $k = 1, \ldots, \lfloor \frac{t}{m} \rfloor$. Define $\bar{N}_{\lfloor \frac{t}{m} \rfloor}(s, a) = \sum_{k=1}^{\lfloor \frac{t}{m} \rfloor} X_k$, and note that $N_t(s, a) \geq \bar{N}_{\lfloor \frac{t}{m} \rfloor}(s, a)$ for all

$t \geq 1$. Let $Q_k = \mathbb{P}(X_k = 1|\mathcal{F}_{k-1})$, where $\mathcal{F}_k = \sigma(Y_1, \ldots, Y_{km})$. Note that we can also write $Q_k$ as $Q_k = \mathbb{P}(\exists n \in \{1, \ldots, m\} : Y_{(k-1)m+n} = 1|\mathcal{F}_{k-1})$. Then

$$Q_k = \mathbb{E}_{(\pi_t)_t} \left[ \mathbb{P}\left( \exists n \in \{1, \ldots, m\} : Y_{(k-1)m+n} = 1|\mathcal{F}_{k-1}, (\pi_t)_t \right) \right],$$

$$\geq \mathbb{E}_{(\pi_t)_t} \left[ \max_{1 \leq n \leq m} \mathbb{P}\left( Y_{(k-1)m+n} = 1|\mathcal{F}_{k-1}, (\pi_t)_t \right) \right],$$

$$\geq \mathbb{E}_{(\pi_t)_t, z_0} \left[ \max_{1 \leq n \leq m} \mathbb{P}\left( z_{(k-1)m+n} = z|z_{(k-1)m} = z_0, \mathcal{F}_{k-1}, (\pi_t)_t \right) \right],$$

$$= \mathbb{E}_{(\pi_t)_t, z_0} \left[ \max_{1 \leq n \leq m} \mathbb{E}_{(z_i)_{i=1}^{n-1}} \left[ \prod_{j=0}^{n-1} P_{(k-1)m+j}(z_{(k-1)m+j+1} = z_{j+1}|z_{(k-1)m+j} = z_j)|z_1, \ldots, z_{n-1}, z_n = z \right] |z_0 \right],$$

$$\geq \mathbb{E}_{(\pi_t)_t, z_0} \left[ \max_{1 \leq n \leq m} \mathbb{E}_{(z_i)_{i=1}^{n-1}} \left[ \prod_{j=0}^{n-1} \frac{1}{(km)^{\alpha+\beta}} P_u(z_{(k-1)m+j+1} = z_{j+1}|z_{(k-1)m+j} = z_j)|z_1, \ldots, z_{n-1}, z_n = z \right] |z_0 \right],$$

$$= \mathbb{E}_{(\pi_t)_t, z_0} \left[ \max_{1 \leq n \leq m} \frac{1}{(km)^{\alpha+\beta}} P_u^n(z_{(k-1)m+n} = z|z_{(k-1)m} = z_0)|z_0 \right],$$

$$\geq \frac{\eta_m}{(km)^{\alpha+\beta}}.$$

Hence, for $\lambda' \geq 0$ due to lem. C.7 we also have

$$\mathbb{P}\left( \exists t : N_t(s, a) \leq \frac{1}{2} \sum_{k=1}^{\lfloor \frac{t}{m} \rfloor} \frac{\eta_m}{(km)^{\alpha+\beta}} - \lambda' \right) \leq \mathbb{P}\left( \exists t : \bar{N}_{\lfloor \frac{t}{m} \rfloor}(s, a) \leq \frac{1}{2} \sum_{k=1}^{\lfloor \frac{t}{m} \rfloor} Q_k - \lambda' \right) \leq e^{-\lambda'}.$$

Consequently, choosing $\lambda' = \log(SA/\lambda)$, we get that

$$\mathbb{P}\left( \overline{\mathcal{E}_{\mathrm{cnt}}(\lambda)} \right) \leq \sum_{s,a} \mathbb{P}\left( \exists t : N_t(s, a) \leq \frac{1}{2} \sum_{k=1}^{\lfloor \frac{t}{m} \rfloor} Q_k(s, a) - \log\left( \frac{SA}{\lambda} \right) \right) \leq \sum_{s,a} \frac{\lambda}{SA} = \lambda.$$

$\square$

The following result, automatically follows from the previous lemma.

**Corollary C.1.** *For all $t \geq 1$, under $\mathcal{E}_{\mathrm{cnt}}(\lambda)$ we have*

$$\epsilon_t \leq \min(1, \kappa_t(\lambda)) \text{ where } \kappa_t(\lambda) := \frac{2^\alpha}{A^\alpha \left( \sum_{k=1}^{\frac{t}{m}} \frac{\eta_m}{(km)^{\alpha+\beta}} - 2\log(SA/\lambda) \right)^\alpha}. \tag{25}$$

### C.3.3 Concentration of the empirical MDP

In the following lemma we bound the probability that $M_t$ is not close to $M$. In order to do so, recall that $\pi^\star(a|s) = \omega^\star(a|s)/\sum_b \omega^\star(b|s)$ and $\hat{\pi}_t^\star(a|s) = \omega_t^\star(a|s)/\sum_b \omega_t^\star(b|s)$. Then, observe that by continuity of $M_t \to \omega_t^\star$ (due to Berge's theorem) we that for every $\xi > 0$ there exists $\rho > 0$ such that for $M_t \in B_\rho(M)$ we have $\|\hat{\pi}_t^\star - \pi^\star\|_\infty \leq \xi$.

**Lemma C.4.** *For $\xi > 0, T \geq 1$ define the event $\mathcal{E}_{p,T}^1(\xi) = \cap_{t=h_p(T)}^T (M_t \in B_{\rho(\xi)}(M))$ where $h_p(T) = T^p$ with $p \in (0, 0.5)$. Then, there exists a constant $C > 0$ that only depend on $\xi$ and $M$ such that*

$$\forall T \geq 1, \mathbb{P}\left( \overline{\mathcal{E}_{p,T}^1(\xi)} \right) \leq \frac{1}{T^2} + CT \exp(-2g_{h_p(T)}(\lambda)\rho), \tag{26}$$

*where $g_{h_p(T)}(\lambda) = \frac{1}{2} \sum_{k=1}^{\lfloor \frac{h_p(T)}{m} \rfloor} \frac{\eta_m}{(km)^{\alpha+\beta}} - \log\left( \frac{SA}{\lambda} \right)$.*

*Proof.* Consider the event $\mathcal{E}_{\mathrm{cnt}}(\lambda)$ from lem. C.3 and let $\lambda = 1/T^2$. Note that

$$\mathbb{P}\left( \overline{\mathcal{E}_{p,T}^1(\xi)} \right) \leq \lambda + \mathbb{P}\left( \overline{\mathcal{E}_{p,T}^1(\xi)} \cap \mathcal{E}_{\mathrm{cnt}}(\lambda) \right).$$

For simplicity, we omit the dependence on $\lambda$ and simply write $\mathcal{E}_{\text{cnt}}$. Then, we have [5]

$$\mathbb{P}(\overline{\mathcal{E}_{p,T}^1(\xi)} \cap \mathcal{E}_{\text{cnt}}) \leq \sum_{t=h_p(T)}^{T} \mathbb{P}(M_t \notin B_\rho(M) \cap \mathcal{E}_{\text{cnt}}),$$

$$= \sum_{t=h_p(T)}^{T} \mathbb{P}(\max_{s,a} \|P_t(\cdot|s,a) - P(\cdot|s,a)\|_1 > \rho \cap \mathcal{E}_{\text{cnt}}),$$

$$\leq \sum_{t=h_p(T)}^{T} \sum_{s,a,s'} \mathbb{P}(|P_t(s'|s,a) - P(s'|s,a)| > \rho/S \cap \mathcal{E}_{\text{cnt}}).$$

Now, let $g_t(\lambda) = \frac{1}{2} \sum_{k=1}^{t} \frac{\eta_m}{(km)^{\alpha+\beta}} - \log\left(\frac{SA}{\lambda}\right)$. Then, we have

$$\mathbb{P}(|P_t(s'|s,a) - P(s'|s,a)| > \rho/S \cap \mathcal{E}_{\text{cnt}}) \leq \mathbb{P}(|P_t(s'|s,a) - P(s'|s,a)| > \rho/S \cap N_t(s,a) \geq g_t(\lambda)),$$

$$\leq \sum_{k=g_t(\lambda)}^{t} \mathbb{P}(|P_t(s'|s,a) - P(s'|s,a)| > \rho/S \cap N_t(s,a) = k),$$

$$\leq \sum_{k=g_t(\lambda)}^{t} 2\exp(-2k\rho^2),$$

$$\leq \sum_{k=0}^{t-g_t(\lambda)} 2\exp(-2(k + g_t(\lambda)\rho^2),$$

$$\leq \exp(-2g_t(\lambda)\rho) \sum_{k=0}^{t-g_t(\lambda)} 2\exp(-2k\rho^2),$$

$$\leq \frac{2\exp(-2g_t(\lambda)\rho)}{1 - \exp(-2\rho^2)}.$$

Hence, we find

$$\sum_{t=h_p(T)}^{T} \sum_{s,a,s'} \frac{2\exp(-2g_t(\lambda)\rho)}{1 - \exp(-2\rho^2)} \leq \sum_{t=h_p(T)}^{T} \frac{2S^2A\exp(-2g_t(\lambda)\rho)}{1 - \exp(-2\rho^2)},$$

$$\leq \frac{2S^2AT\exp(-2g_{h_p(T)}(\lambda)\rho)}{1 - \exp(-2\rho^2)}.$$

$\square$

### C.3.4 Concentration of the state-action visitation frequency

Recall now that $\omega_t$ is the stationary distribution induced by the exploration policy $\pi_t$ at time $t$. Further, recall that $\pi_t = (1 - \varepsilon_t)\pi_t^\star + \varepsilon_t\pi_{f,t}$, $\omega_t^\star = \inf_{\omega \in \Omega(M_t)} U(\omega; M_t)$, $\bar{\omega}_t^\star = (1/t)\sum_{n=1}^{t} \omega_n^\star$ and that $\pi_t^\star(a|s) = \bar{\omega}_t^\star(a|s)/\sum_{a'} \bar{\omega}_t^\star(s,a')$. We have the following result.

**Corollary C.2.** *Let $T \geq 1, p \in (0,1), \xi > 0, \lambda \in (0,1)$. For $t \geq T^p$ under $\mathcal{E}_{p,T}^1(\xi)$ we have that there exists $\rho > 0$ and $\kappa_t(\lambda)$ such that*

$$\|\omega_t - \omega^\star\|_1 \leq \kappa_M \left[\frac{T^p}{t} + \xi + \epsilon_t\right]. \tag{27}$$

*As a consequence, under $\mathcal{E}_{p,T}^1(\xi)$ we also have that*

$$\|P_t - P_{t-1}\|_\infty \leq 2\left(\frac{T^p}{t-1} + \xi + \epsilon_{t-1}\right).$$

---

[5]For simplicity, and w.l.o.g., we consider $h_p(T)$ as an integer in the summation.

*Proof.* Let $\pi^\star(a|s) = \omega^\star(a|s)/\sum_b \omega^\star(b|s)$ and $\hat{\pi}_t^\star(a|s) = \omega_t^\star(a|s)/\sum_b \omega_t^\star(b|s)$.

Under $\mathcal{E}_{p,T}^1(\xi)$ using [17, Lemma 23] there exists an MDP-specific constant $\kappa_M$ such that

$$
\begin{aligned}
\|\omega_t - \omega^\star\|_1 &\leq \kappa_M \|P_t - P_{\pi^\star}\|_\infty, \\
&\leq \kappa_M \|\pi_t - \pi^\star\|_\infty, \\
&\leq \kappa_M \left[\|\pi_t^\star - \pi^\star\|_\infty + \epsilon_t\right], \\
&\leq \kappa_M \left[\|\pi_t^\star - \pi^\star\|_\infty + \epsilon_t\right].
\end{aligned}
$$

Then, for $t \geq T^p$, we have

$$
\|\pi_t^\star - \pi^\star\|_\infty \leq \left\| \frac{1}{t}\sum_{k=1}^{t}(\hat{\pi}_t^\star - \pi^\star) \right\| \leq \frac{T^p}{t} + \xi,
$$

which concludes the first part of the proof. For the second part, simply note that

$$
\begin{aligned}
\|P_t - P_{t-1}\|_\infty &\leq \|P_t - P_{\pi^\star}\|_\infty + \|P_{\pi^\star} - P_{t-1}\|_\infty, \\
&\leq 2\left(\frac{T^p}{t-1} + \xi + \epsilon_{t-1}\right).
\end{aligned}
$$

$\square$

From the previous lemma we remark the following important fact: under $\mathcal{E}_{p,T}^1(\xi)$ for $k \geq T^q \geq T^p$ we have

$$
\|\pi_k^\star - \pi^\star\|_\infty \leq \frac{T^p}{k} + \xi \leq \frac{1}{T^{q-p}} + \xi.
$$

In the following proof we provide a a reformulation of [17, Proposition 19]. We require $t$ and $T$ to be larger than some constants. Specifically, we require $t \geq T^{1-p}$ (note that $p \in (0, 0.5)$) and $T \geq \max\left(T_q(\xi, \lambda), \frac{1}{\xi^{\frac{1}{q-p}}}, \frac{1}{\xi^{\frac{1}{1-p-q}}}\right)$, where $q \in (0,1)$ satisfies $p < q, p + q < 1$ and $T_q(\xi, \lambda) := \inf\{T \geq 1 : \xi \geq \kappa_{T^q}(\lambda) + \frac{1}{T^{(1-3p)/2}}\}$ with $\xi > 0, \lambda \in (0,1)$ and $\kappa_t$ defined in corollary C.1.

**Proposition C.3.** *Under* `MR-NaS` *there exists* $T(\xi, \lambda)$ *such that for all* $T \geq \max\left(T_q(\xi, \lambda), \frac{1}{\xi^{\frac{1}{q-p}}}, \frac{1}{\xi^{\frac{1}{1-p-q}}}\right)$ *and all* $t \geq T^{1-p}$. *and all functions* $f : \mathcal{Z} \to [0,1]$ *we have*

$$
\mathbb{P}\left( \left| \frac{\sum_{k=1}^{t} f(z_k)}{t} - \omega^\star(f) \right| \geq \kappa_\xi \xi \Big| \mathcal{E}_{p,T}^1(\xi) \cap \mathcal{E}_{\mathrm{cnt}}(\lambda) \right) \leq 2\exp(-t\xi^2), \tag{28}
$$

*where* $\xi \to K_\xi$ *is a mapping with values in* $(1, \infty)$ *such that* $\limsup_{\xi \to 0} K_\xi < \infty$.

*Proof.* Let $\omega^\star(f) = \mathbb{E}_{z \sim \omega^\star}[f(z)]$ and $\omega_t(f) = \mathbb{E}_{z \sim \omega_t}[f(z)]$. Then

$$
\begin{aligned}
D &= \frac{\sum_{k=1}^{t} f(z_k)}{t} - \omega^\star(f), \\
&= \frac{\sum_{k=1}^{T^q} f(z_k) - \omega^\star(f)}{t} + \frac{\sum_{k=T^q+1}^{t} f(z_k) - \omega^\star(f)}{t}, \\
&= \underbrace{\frac{\sum_{k=1}^{T^q} f(z_k) - \omega^\star(f)}{t}}_{(a)} + \underbrace{\frac{\sum_{k=T^q+1}^{t} f(z_k) - \omega_{k-1}(f)}{t}}_{(b)} + \underbrace{\frac{\sum_{k=T^q+1}^{t} \omega_{k-1}(f) - \omega^\star(f)}{t}}_{(c)}.
\end{aligned}
$$

**First term (a).** For the first term $(a)$ for $t \geq T^{1-p}$ and $\xi \geq 1/T^{1-p-q}$ we have

$$
(a) \leq \frac{T^q}{t} \leq \frac{1}{T^{1-p-q}} \leq \xi.
$$

**Last term (c).** The last term $(c)$ under $\mathcal{E}^1_{p,T}(\xi)$ for $t \geq T^{1-p}$ is bounded as [6]

$$
\begin{aligned}
(c) &\leq \frac{\sum_{k=T^q}^{t-1} \|\omega_k - \omega^\star\|_1}{t}, \\
&\leq \frac{\sum_{k=T^q}^{t-1} \kappa_M [T^p/k + \xi + \epsilon_{T_q}]}{t}, \\
&\leq \kappa_M \left( \xi + \epsilon_{T_q} + \frac{\sum_{k=T^q}^{t-1} T^p/k}{t} \right), \\
&\leq \kappa_M \left( \xi + \epsilon_{T_q} + \frac{T^p \log(t)}{t} \right), \\
&\leq \kappa_M \left( \xi + \epsilon_{T_q} + \frac{T^p}{\sqrt{t}} \right), \\
&\leq \kappa_M \left( \xi + \epsilon_{T_q} + \frac{1}{T^{(1-3p)/2}} \right).
\end{aligned}
$$

Hence, under $\mathcal{E}^1_{p,T}(\xi) \cap \mathcal{E}_{\mathrm{cnt}}(\lambda)$, since $\epsilon_k \leq \kappa_k(\lambda)$ we get that $(c) \leq \kappa_M \left( \xi + \kappa_{T_q}(\lambda) + \frac{1}{T^{(1-3p)/2}} \right)$. Let $T_q(\xi, \lambda) = \inf\{T \geq 1 : \xi \geq \kappa_{T^q}(\lambda) + \frac{1}{T^{(1-3p)/2}}\}$ where $\kappa_t$ is defined in corollary C.1. Then for $T \geq T_q(\xi, \lambda)$ we have $(c) \leq 2\kappa_M \xi$.

**Second term (b).** For the second term we use the function $\hat{f}_k$ solution to the Poisson equation $f(\cdot) - \omega_{k-1}(f) = [\hat{f}_k - P_k \hat{f}_k](\cdot)$ [17, Lemma 23]. Hence

$$
(b) = (1/t) \sum_{k=T^q+1}^{t} \hat{f}_{k-1}(z_k) - P_{k-1} \hat{f}_{k-1}(z_k) = M_t + C_t + R_t,
$$

where

$$
\begin{aligned}
M_t &:= \frac{\sum_{k=T^q+1}^{t} \hat{f}_{k-1}(z_k) - P_{k-1} \hat{f}_{k-1}(z_{k-1})}{t}, \\
C_t &:= \frac{\sum_{k=T^q+1}^{t} P_k \hat{f}_k(z_k) - P_{k-1} \hat{f}_{k-1}(z_k)}{t}, \\
R_t &:= \frac{P_{T^q} \hat{f}_{T^q}(z_{T^q}) - P_t \hat{f}_t(z_t)}{t}.
\end{aligned}
$$

Before proceeding, always under $\mathcal{E}^1_{p,T}(\xi) \cap \mathcal{E}_{\mathrm{cnt}}(\lambda)$ for $T \geq T_q(\xi, \lambda)$ we have that

$$
|\epsilon_k| \leq \kappa_{T_q}(\lambda) \leq \xi,
$$

and for $k \geq T^q, \xi \geq 1/T^{q-p}$

$$
\|\pi^\star_k - \pi^\star\|_\infty \leq \frac{T^p}{k} + \xi \leq \frac{T^p}{T^{q-p}} + \xi \leq 2\xi.
$$

Therefore $L_k \leq L_\xi$, where $L_\xi = \sup_{|\epsilon| \leq \xi, \|\pi - \pi^\star\|_\infty \leq 2\xi} L(\epsilon, \pi)$.

**Bounding $M_t$.** As in [17], $t M_t$ is a martingale satisfying $|k M_k - (k-1) M_{k-1}| \leq 2 L_{k-1}$, and hence for $t \geq T^{1-p}$ we bound using Azuma-Hoeffding inequality $\mathbb{P}(|M_t| \geq 2 L_\xi \xi) \leq 2 \exp(-t\xi^2)$.

---

[6]For simplicity, and w.l.o.g., we consider $T^q$ as an integer in the summation.

**Bounding $C_t$.** Again, for $t \geq T^{1-p}$ we have

$$C_t \leq (1/t) \sum_{k=T^q+1}^{t} L_k[\|\omega_k - \omega_{k-1}\| + L_{k-1}\|P_k - P_{k-1}\|_\infty],$$

$$\leq (1/t) \sum_{k=T^q+1}^{t} L_k \left[ \|\omega_k - \omega_{k-1}\| + 2L_{k-1} \left( \frac{T^p}{t-1} + \xi + \epsilon_{t-1} \right) \right],$$

$$\leq (1/t) \sum_{k=T^q+1}^{t} L_k \left[ \|\omega_k - \omega^\star\| + \|\omega_{k-1} - \omega^\star\| + 2L_{k-1} \left( \frac{T^p}{t-1} + \xi + \epsilon_{t-1} \right) \right],$$

$$\leq (1/t) \sum_{k=T^q+1}^{t} 2L_k \left[ \kappa_M \left( \frac{T^p}{k-1} + \xi + \epsilon_{k-1} \right) + L_{k-1} \left( \frac{T^p}{k-1} + \xi + \epsilon_{k-1} \right) \right],$$

$$\leq 2(1/t) \sum_{k=T^q+1}^{t} L_\xi(\kappa_M + L_\xi) \left[ \frac{T^p}{k-1} + \xi + \epsilon_{k-1} \right],$$

$$\leq 2L_\xi(\kappa_M + L_\xi) \left[ \frac{T^p}{\sqrt{t}} + \xi + \kappa_{T_q}(\lambda) \right],$$

$$\leq 2L_\xi(\kappa_M + L_\xi) \left[ \frac{1}{T^{(1-3p)/2}} + \xi + \kappa_{T_q}(\lambda) \right],$$

$$\leq 4L_\xi(\kappa_M + L_\xi)\xi.$$

**Bounding $R_t$.** Finally, for the last term we get for $T \geq T_q(\xi, \lambda)$ we have $|R_t| \leq 2L_\xi/t \leq 2L_\xi/T^{1-p} \leq 2L_\xi/T^{(1-3p)/2} \leq 2L_\xi\xi$.

**Last step.** Summing up all terms yields that for $t \geq T^{1-p}$, $T \geq \max\left( T_q(\xi, \lambda), \frac{1}{\xi^{\frac{1}{q-p}}}, \frac{1}{\xi^{\frac{1}{1-p-q}}} \right)$ we have

$$\mathbb{P}\left( \left| \frac{\sum_{k=1}^{t} f(z_k)}{t} - \omega^\star(f) \right| \geq \kappa_\xi \xi \Big| \mathcal{E}_{p,T}^1(\xi) \cap \mathcal{E}_{\mathrm{cnt}}(\lambda) \right) \leq 2\exp(-t\xi^2), \tag{29}$$

where $\kappa_\xi = 1 + 2\kappa_M + 4L_\xi(1 + \kappa_M + L_\xi)$. $\qquad\square$

As a corollary of the previous proposition, we find the following concentration result on $N_t(s,a)/t$ and the high probability event $\mathcal{E}_T^2(\xi) = \cap_{t=T^{1-p}}^T \left( |\frac{N_t}{t} - \omega^\star| \leq K_\xi\xi \right)$.

**Corollary C.3.** *For `MR-NaS` we have*

$$\mathbb{P}\left( \exists(s,a) \in \mathcal{Z}, \left| \frac{N_t(s,a)}{t} - \omega^\star(s,a) \right| \geq \kappa_\xi\xi \Big| \mathcal{E}_{p,T}^1(\xi) \cap \mathcal{E}_{\mathrm{cnt}}(\lambda) \right) \leq 2SA\exp(-t\xi^2), \tag{30}$$

*and*

$$\mathbb{P}\left( \overline{\mathcal{E}_{p,T}^2(\xi)} \Big| \mathcal{E}_{p,T}^1(\xi) \cap \mathcal{E}_{\mathrm{cnt}}(\lambda) \right) \leq \frac{2SA\exp(-T^{1-p}\xi^2)}{1 - \exp(-\xi^2)}. \tag{31}$$

### C.3.5 Expected sample complexity of `MR-NaS`

Finally, we have the expected sample complexity upper bound of `MR-NaS`, which closely follows the classical proofs in [17, 20].

**Proof strategy.** To summarize, the proof strategy relies on ensuring that $N_t(s,a)/t$ concentrates around $\omega^\star(s,a)$. To that aim, the concentration happens under some events $\mathcal{E}_{p,T}^1(\xi)$ (defined in lem. C.4) given that $N_t(s,a)$ is visited sufficiently often (see lem. C.3, or the alternative version in lem. C.5). As $\delta \to 0$, by the properties of $\beta(N_t, \delta)$, one can show that that $\mathbb{E}[\tau]/\ln(1/\delta)$ does not exceed $\sup_{M' \in B_{\rho(\xi)}(M), \|\omega' - \omega^\star\|_\infty \leq K_\xi\xi} U(\omega', M')\ln(1/\delta)$, where $B_{\rho(\xi)}(M)$ is a ball of a certain radius $\rho(\xi)$ around $M$ (see def. of $\rho(\xi)$ in app. C.3.3) and $K_\xi > 0$ for all $\xi > 0$. Hence, letting $\xi \to 0$ concludes the proof.

**Theorem C.2** (Expected sample complexity). *`MR-NaS` with the stopping rule in prop. 3.1 guarantees that $\mathbb{E}[\tau] < \infty$ and $\limsup_{\delta \to 0} \frac{\mathbb{E}[\tau]}{\log(1/\delta)} \leq U^\star(M)$.*

*Proof.* Define
$$U^\star(\xi) = \sup_{M' \in B_{\rho(\xi)}(M), \|\omega' - \omega^\star\|_\infty \leq K_\xi \xi} U(\omega', M').$$
Similarly to [17], by prop. C.3 under $\mathcal{E}^1_{p,T}(\xi)$ (defined in lem. C.4) the event $\mathcal{E}^2_{p,T}(\xi)$ (defined in corollary C.3) holds with high probability, and there exists $T_1(\xi, \lambda)$ such that for $T \geq T(\xi, \lambda)$ we have $U(N_t/t, M_t) \leq U^\star(\xi)$ for all $t \geq T^{1-p}$.

Further, by the properties of the threshold function $\beta(N_t, \delta)$, there exists $T_3(\xi)$ such that for $t \geq T_2(\xi)$ we have $\beta(N_t, \delta) \leq \log(1/\delta) + \xi U^\star(\xi)^{-1} t$. Then, let $T_3(\xi, \delta)$ be such that $\log(1/\delta) + \xi U^\star(\xi)^{-1} T_3(\xi, \delta) = T_3(\xi, \delta)$, that is
$$T_3(\xi, \delta) = \frac{U^\star(\xi) \log(1/\delta)}{1 - \xi}.$$
Hence, under $\mathcal{E}^1_{p,T}(\xi) \cap \mathcal{E}^2_{p,T}(\xi)$ for $T \geq \max(T_1(\xi, \lambda), T_2(\xi), T_3(\xi, \delta))$ it holds that $T U^{-1}(N_T/T, M_t) \geq \beta(N_T, \delta)$, and thus $(\tau \leq T) \supseteq \mathcal{E}^1_{p,T}(\xi) \cap \mathcal{E}^2_{p,T}(\xi)$. Therefore, using that $\mathbb{E}[\tau] = \sum_{T=1}^\infty \mathbb{P}(\tau > T)$ one can show[7]

$$\mathbb{E}[\tau] \leq \max(T_1(\xi, \lambda), T_2(\xi), T_3(\xi, \delta)) + \sum_{T=\max(T_1(\xi,\lambda), T_2(\xi), T_3(\xi,\delta))}^\infty \mathbb{P}\left(\overline{\mathcal{E}^1_{p,T}(\xi)} \cup \overline{\mathcal{E}^2_{p,T}(\xi)}\right),$$

$$\leq \max(T_1(\xi, \lambda), T_2(\xi), T_3(\xi, \delta)) + \sum_{T=\max(T_1(\xi,\lambda), T_2(\xi), T_3(\xi,\delta))}^\infty \frac{1}{T^2} + CT \exp(-2g_{h_p(T)}(\lambda)\rho)$$

$$+ \frac{2SA \exp(-T^{1-p}\xi^2)}{1 - \exp(-\xi^2)},$$

and thus $\mathbb{E}[\tau]$ is finite since $g_{h_p(T)}(\lambda)$ grows as $T^p$ for a fixed $\lambda$. Hence
$$\limsup_{\delta \to 0} \frac{\mathbb{E}[\tau]}{\log(1/\delta)} \leq \limsup_{\delta \to 0} \frac{\max(T_1(\xi, \lambda), T_2(\xi), T_3(\xi, \delta))}{\log(1/\delta)} \leq \frac{U^\star(\xi)}{1 - \xi}.$$
Letting $\xi \to 0$ concludes the proof. $\qquad\square$

### C.3.6 Alternative forced exploration with high probability

In this section we introduce an alternative result for the forced exploration property with high probability that follows the one proposed in [17]. In addition to the quantities defined in app. C.3, we also define $\eta := \eta_1 \eta_m$ and $\bar{m} := \max\left(m, \left\lceil \frac{1}{2(\alpha+\beta)} \right\rceil\right)$.

The following lemma is an alternative to lem. C.3, and it is adaptation of [17, Lemma 11]. Using this lemma, one can obtain a sample complexity upper bound in expectation as in Alg. 2. For the sake of simplicity, we only state the main steps of the proof that differ from the one in [17].

**Lemma C.5** (High probability forced exploration). *For all $\lambda \in (0, 1)$ we have*

$$\mathbb{P}\left(\forall z \in \mathcal{Z}, \forall t \geq 1, N_t(z) \geq \left(\frac{t}{g(\lambda)^2}\right)^{1/4} - 1\right) \geq 1 - \lambda, \tag{32}$$

*where $g(\lambda) = \frac{\bar{m}}{\eta} \log\left(1 + \frac{SA}{\lambda}\right)$ and $\bar{m} := \max\left(m, \left\lceil \frac{1}{2(\alpha+\beta)} \right\rceil\right)$.*

*Proof.* Denote by $\tau_k(z)$ the $k$-th time an agent visit a state-action pair $z = (s, a)$, and define the event $\mathcal{E} = (\forall z \in \mathcal{Z}, \forall k \geq 1, \tau_k(z) \leq f(k))$ for some increasing function $f : \mathbb{N} \to \mathbb{N}$, satisfying $f(0) = 0$. One can find that

$$\mathbb{P}(\overline{\mathcal{E}}) \leq SA \sum_{k=1}^\infty b_k, \quad b_k := \prod_{j=0}^{\lfloor \frac{f(k)-f(k-1)-1}{\bar{m}} \rfloor - 1} \left(1 - \eta \prod_{l=1}^{\bar{m}} \frac{1}{(f(k-1) + \bar{m}j + l)^{\alpha+\beta}}\right).$$

---

[7]For simplicity we consider $\max(T_1(\xi, \lambda), T_2(\xi), T_3(\xi, \delta))$ as an integer.

For $f(k) = \lambda k^4$ we have $\lfloor \frac{f(k)-f(k-1)-1}{\bar{m}} \rfloor \geq \frac{\lambda k^3}{\bar{m}}$ and

$$b_k = \prod_{j=0}^{\lfloor \frac{f(k)-f(k-1)-1}{\bar{m}} \rfloor - 1} \left( 1 - \eta \prod_{l=1}^{\bar{m}} \frac{1}{(f(k-1)+\bar{m}j+l)^{\alpha+\beta}} \right),$$

$$\leq \left( 1 - \eta \prod_{l=1}^{\bar{m}} \frac{1}{f(k)^{\alpha+\beta}} \right)^{\lfloor \frac{f(k)-f(k-1)-1}{\bar{m}} \rfloor},$$

$$\leq \left( 1 - \eta \frac{1}{f(k)^{\bar{m}(\alpha+\beta)}} \right)^{\lfloor \frac{f(k)-f(k-1)-1}{\bar{m}} \rfloor},$$

$$\leq \left( 1 - \eta \frac{1}{f(k)^{\bar{m}(\alpha+\beta)}} \right)^{\frac{\lambda k^3}{\bar{m}}},$$

$$\leq \exp\left( -\eta \frac{\lambda k^3}{\bar{m} f(k)^{\bar{m}(\alpha+\beta)}} \right).$$

Now, since $\frac{1}{2(\alpha+\beta)} \leq \bar{m}$, then we have that $\bar{m}(\alpha+\beta) \geq 1/2$, thus $b_k \leq \exp\left( -\eta \frac{\lambda k^3}{\bar{m}\sqrt{f(k)}} \right) = \exp\left( -\frac{\eta k \sqrt{\lambda}}{\bar{m}} \right)$. Hence

$$\sum_{k=1}^{\infty} b_k \leq \frac{1 - \exp\left( -\frac{\eta \sqrt{\lambda}}{\bar{m}} \right)}{\exp\left( -\frac{\eta \sqrt{\lambda}}{\bar{m}} \right)}.$$

Finally, the conclusion follows similarly as in [17]. $\qquad \square$

The previous result relies on the following close adaptation of [17, Lemma 20]. Again, we only show the main steps of the proof.

**Lemma C.6.** *Fix $z_0 \in \mathcal{Z}$, and let $P_t$ be the transition induced by the sampling policy $\pi_t$ of* MR-NaS. *Define the sub-stochastic matrix $Q_t$ obtained by removing from $P_t$ the row and the column corresponding to $z_0$:*

$$P_t = \left[ \begin{array}{c|c} Q_t & [P_t(z_0|z)]_{z \neq z_0} \\ \hline [P_t(z|z_0)]^\top_{z \neq z_0} & P_t(z_0|z_0) \end{array} \right]$$

*Then, we have*

$$\forall n \geq 1, \left\| \prod_{k=n+1}^{n+m} Q_k \right\|_\infty \leq 1 - \eta \prod_{k=n+1}^{n+m} \frac{1}{k^{\alpha+\beta}}. \tag{33}$$

*Proof.* Define $r_k(n_1, n_2) := \sum_{j=1}^{SA-1} \left( \prod_{l=n_1+1}^{n_2} Q_l \right)_{kj}$ to be the sum of the $k$-th row of the product of matrices $Q_l$ for $l \in \{n_1+1, \ldots, n_2\}$, and observe that $\left\| \prod_{l=n+1}^{n+m} Q_l \right\|_\infty = \max_{i \in \{1, \ldots, SA-1\}} r_i(n, n+m)$.

Since the MDP is communicating, there exists $z \in \mathcal{Z}$ s.t. $P_u(z_0|z) \geq \eta_1$ and let $k^*$ be the corresponding row of $z$ in $Q_t$. Then, using lem. C.1 and the fact that $\epsilon_t = 1/\max(1, N_t(s))^\alpha$, for $n_1 \geq 1$ we have $P_t \geq \frac{P_u}{N_t(s)^{\alpha+\beta}} \geq \eta_1/t^{\alpha+\beta}$, and thus

$$r_{k^*}(n_1, n_1+1) = \sum_{j=1}^{SA-1} (Q_{n_1+1})_{k^*j} = 1 - P_{n_1+1}(z_0|z) \leq 1 - \eta_1 \frac{1}{(n_1+1)^{\alpha+\beta}}.$$

For $n_1, n_2 \geq 1$, by recursion one can show that

$$r_{k^*}(n_1, n_1+n_2) = \sum_{j=1}^{SA} \left( \prod_{l=n_1+1}^{n_1+n_2} Q_l \right)_{k^*j} \leq r_{k^*}(n_1, n_1+1) \leq 1 - \eta_1 \frac{1}{(n_1+1)^{\alpha+\beta}}.$$

Similarly, for any $i \in \{1, \ldots, SA\} \setminus \{k^*\}$, for all $n_1 \in \{1, \ldots, m_0\}$ we have

$$r_i(n, n+m) = \sum_{j=1}^{SA-1} \left( \prod_{l=n+1}^{n+n_1} Q_l \right)_{ij} r_j(n+n_1, n+m),$$

$$\leq 1 - \eta_1 \frac{1}{(n+n_1+1)^{\alpha+\beta}} \left( \prod_{l=n+1}^{n+n_1} Q_l \right)_{ik^*}.$$

To bound the last term, observe that for $n_i \leq m$

$$\left( \prod_{l=n+1}^{n+n_i} Q_l \right)_{ik^*} \geq \left( \prod_{l=n+1}^{n+n_i} \frac{\epsilon_l}{N_l(s)^\beta} P_u \right)_{ik^*} \geq \eta_m \prod_{l=n+1}^{n+n_i} \frac{1}{l^{\alpha+\beta}}.$$

Therefore

$$r_i(n, n+m) \leq 1 - \eta_1 \eta_m \prod_{l=n+1}^{n+m} \frac{1}{l^{\alpha+\beta}},$$

which concludes the proof. $\qquad\square$

## C.4  Stopping Rule

Next, we provide the proof of the $\delta$-PC property of `MR-NaS`. Note that a necessary condition for the algorithm to stop is that the model at the stopping time $M_\tau$ needs to have a unique optimal policy for every $r \in \mathcal{R}$. Given this condition, we have the following proof of the stopping rule.

*Proof of prop. 3.1.* Let $\mathcal{C} = \{\exists r \in \mathcal{R} : \hat{\pi}_{\tau,r} \notin \Pi^\star(M_r)\}$ be the event that at the stopping time there exists a reward such that the estimated optimal policy is not optimal.

Let $\text{Alt}(M) = \cup_r \text{Alt}(M_r)$. Then, since $M_\tau \in \mathcal{M}$, admitting a single optimal policy for all $r \in \mathcal{R}$, we can say that $\mathcal{C} \subset \{\exists r \in \mathcal{R} : M \in \text{Alt}(M_{t,r})\} \subset \cup_r \{M \in \text{Alt}(M_{t,r})\} \subset \{M \in \text{Alt}(M_t)\}$, where $M_{t,r} = (M_t, r)$. Then

$$\mathbb{P}\left( \tau_\delta < \infty, \mathcal{C} \right) \leq \mathbb{P}\left( \exists t \geq 1 : tU(N_t/t; M_t)^{-1} \geq \beta(N_t, \delta), M \in \text{Alt}(M_t) \right),$$

$$\leq \mathbb{P}\left( \exists t \geq 1 : tT(N_t/t; M_t)^{-1} \geq \beta(N_t, \delta), M \in \text{Alt}(M_t) \right),$$

$$= \mathbb{P}\left( \exists t \geq 1 : \inf_r \inf_{M'_r \in \text{Alt}(M_{t,r})} \sum_{s,a} N_t(s,a) \text{KL}_{M_t|M'_r}(s,a) \geq \beta(N_t, \delta), M \in \text{Alt}(M_t) \right),$$

$$= \mathbb{P}\left( \exists t \geq 1 : \inf_{M' \in \cup_r \text{Alt}(M_{t,r})} \sum_{s,a} N_t(s,a) \text{KL}_{M_t|M'}(s,a) \geq \beta(N_t, \delta), M \in \text{Alt}(M_t) \right),$$

$$\leq \mathbb{P}\left( \exists t \geq 1 : \sum_{s,a} N_t(s,a) \text{KL}_{M_t|M}(s,a) \geq \beta(N_t, \delta) \right),$$

$$\leq \delta,$$

where the conclusion follows from [86, Prop. 1]. $\qquad\square$

## C.5  Technical Lemmas

We have the following adaptation of [87, Lemma F.4].

**Lemma C.7.** *Let $Y_1, \ldots, Y_t$ be a sequence of Bernoulli random variables. Let $\mathcal{F}_i$ be a filtration, and for $i = 1, \ldots, \lfloor \frac{t}{m} \rfloor$ and $m \in \mathbb{N}$ let $X_i = \mathbf{1}_{Y_{(i-1)m+1} + \cdots + Y_{im} > 0}$ be a sequence of Bernoulli random variables such that $Q_i = \mathbb{P}(X_i | \mathcal{F}_{i-1})$ is $\mathcal{F}_{i-1}$-measurable and $X_i$ is $\mathcal{F}_i$-measurable. Then, for $\lambda \geq 0$ we have that*

$$\mathbb{P}\left( \exists t : \sum_{i=1}^{\lfloor \frac{t}{m} \rfloor} X_i \leq \frac{1}{2} \sum_{i=1}^{\lfloor \frac{t}{m} \rfloor} Q_i - \lambda \right) \leq e^{-\lambda}. \tag{34}$$

Finally, we report the following proposition from [86, 17] that bounds the self-normalized KL process.

**Proposition C.4** (Prop. 1, [86] and Lemma 15 [17]). *Define the threshold* $\beta(N_t, \delta) := \log(1/\delta) + (S-1) \sum_{s,a} \log \left( e \left[ 1 + \frac{N_t(s,a)}{S-1} \right] \right)$. *Then, for all* $\delta \in (0, 1)$ *we have*

$$\mathbb{P} \left( \exists t \geq 1 : \sum_{s,a} N_t(s,a) \text{KL}_{M_t|M}(s,a) > \beta(N_t, \delta) \right) \leq \delta, \tag{35}$$

*with the convention that* $N_t(s,a) \text{KL}_{M_t|M}(s,a) = 0$ *whenever* $N_t(s,a) = 0$.

# D Numerical Results

In this section we present more in detail the numerical results for the tabular case and for Deep Reinforcement Learning. We also introduce additional numerical results on radio network control problems.

## D.1 Numerical Results for the tabular case

In this section we delve more into the detail of the numerical results for the tabular case. We focus on different hard-exploration tabular environments: Riverswim [54], Forked Riverswim [53], DoubleChain [22] and NArms [54] (an adaptation of SixArms to $N$ arms). We compare MR-NaS against RF-UCRL [22] (a reward-free exploration method), ID3AL [33] (a maximum entropy exploration approach) and RF-PSRL, a reward-free adaptation of PSRL [35] (posterior sampling for RL). As the baselines are not specific to our setting, we consider different sets of experiments.

We begin by describing the environments, and then present the results.

### D.1.1 Environments Details

Here we provide a brief description of the environments.

**Riverswim.** The RiverSwim environment is a classic reinforcement learning benchmark designed to test exploration [54]. It consists of a series of states arranged in a linear chain, where an agent can choose to swim right (downstream) or left (upstream). In the single-reward setting the agent can achieve a positive return by swimming right, but requires overcoming a strong current, making it a less probable event. Conversely, swimming left generally offers small to zero rewards, but is easier. This setup requires the agent to balance immediate, safer rewards with potentially higher but riskier returns. It is exponentially hard for a random uniform agent to reach the final state.

In figure fig. 3 is shown a depiction of the environment. There are $n$ states, and two main parameters, $p, p' \in (0, 1)$, and their sum $p_{\text{sum}} = p + p' < 1$. In the figure, each tuple $(a, p)$ represents the action $a$ that triggers the transition and the probability $p$ of that event. The agent starts in state $s_0$, and in every state can only take two actions $\{a_0, a_1\}$. For small values of $p$ it becomes difficult for the agent to swim right (i.e., reach $s_{n-1}$), and larger values of $p'$ can also hinder the progress. On the other hand, swimming towards the left is easier, since the probability of $P(s_{i-1}|s_i, a_0) = 1$. For the experiments, we used $n = 10, p = 0.3, p' = 0.6$.

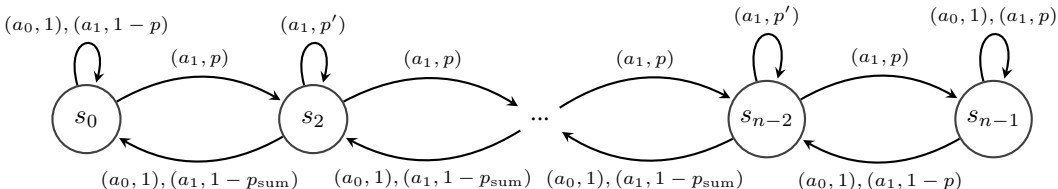

Figure 3: Riverswim environment [54]. Each tuple $(a, p)$ represents the action $a$ that triggers the transition and the probability $p$ of that event.

**Forked Riverswim.** The Forked RiverSwim environment [53] is a variation of the traditional RiverSwim reinforcement learning benchmark, designed to test more complex exploration strategies. In this variant, the state space branches into multiple paths, resembling a river that forks. At intermediate states the agent can switch between the forks, while the end states are not connected. This variant requires the agent to make more sophisticated decisions to explore the environment. This setup increases the sample complexity and challenges the agent's ability to generalize across different paths within the environment.

In fig. 4 is shown a depiction of the environment. There are a total of $2n+1$ states, and two parameters $p, p' \in (0, 1)$, so that $p_{\text{sum}} = p + p' < 1$. In the figure, each tuple $(a, p)$ represents the action $a$ that triggers the transition and the probability $p$ of that event. The agent starts in state $s_0$, and in

every state can chose between three actions $\{a_0, a_1, a_2\}$. For small values of $p$ it becomes difficult for the agent to swim right in both forks, and larger values of $p'$ can also hinder the progress. As in Riverswim, swimming towards the left is easier, since the probability of $P(s_{i-1}|s_i, a_0) = 1$. For the experiments, we used $n = 4, p = 0.3, p' = 0.6$.

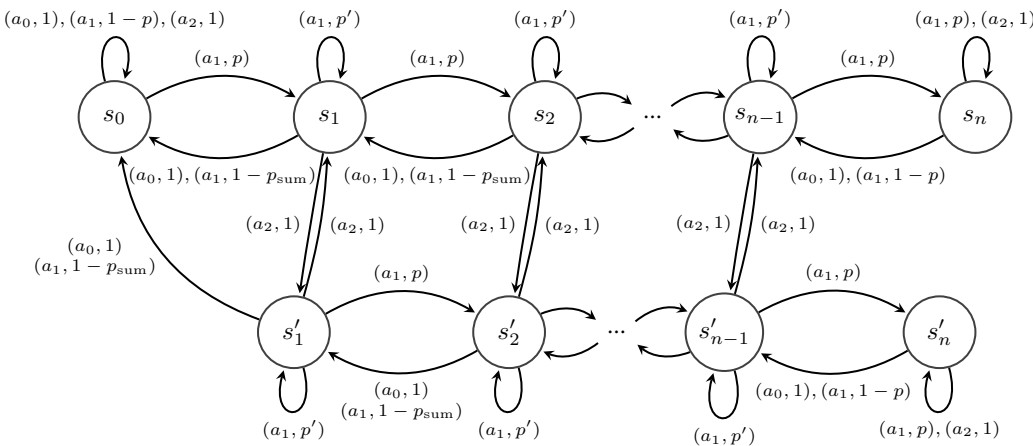

Figure 4: `Forked Riverswim` environment [53]. Each tuple $(a, p)$ represents the action $a$ that triggers the transition and the probability $p$ of that event.

**Double Chain.** The Double Chain environment [22] consists of two chains, similarly to the Forked Riverswim. The main difference consists in the fact that it is not possible to switch between the two chains, and intermediate states are transient (there is no parameter $p'$).

In fig. 5 is shown a depiction of the environment. There are a total of $2n+1$ states, and one parameters $p \in (0, 1)$. In the figure, each tuple $(a, p)$ represents the action $a$ that triggers the transition and the probability $p$ of that event. The agent starts in state $s_0$, and in every state can chose between two actions $\{a_0, a_1\}$. For small values of $p$ it becomes difficult for the agent to move to the end of the chain in both chains. For the experiments, we used $n = 6, p = 0.7$.

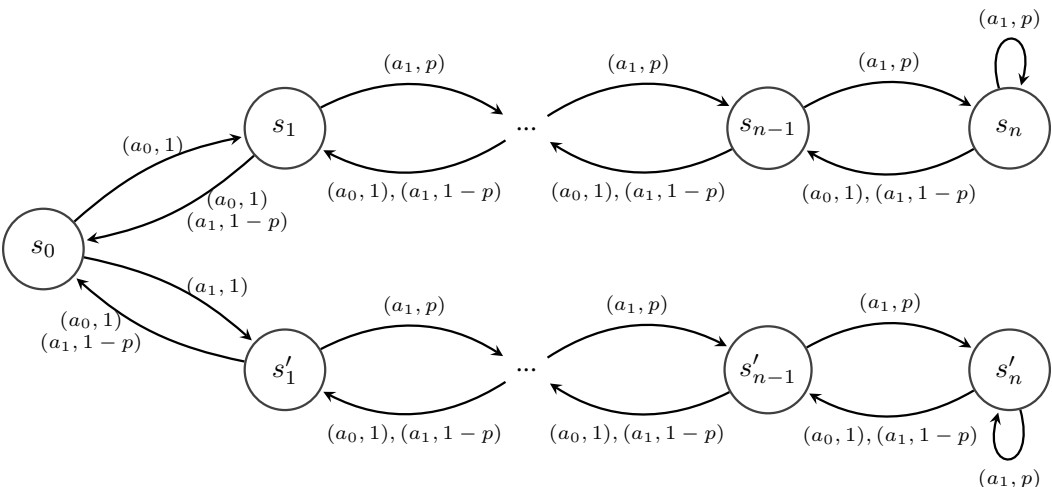

Figure 5: `Double Chain` environment [22] . Each tuple $(a, p)$ represents the action $a$ that triggers the transition and the probability $p$ of that event.

**NArms.** This environment is an adaptation to $N$ arms of the original `6Arms` environment from [54]. Differently from the previous environments, this is a bandit-like environment, where the agent is presented with $n$ different actions (or arms) to choose from. The agent starts in a state $s_0$ and selects an arm $a_i$. Upon selecting an arm, the agent may transition to corresponding state $s_i$. Certain arms are more difficult to observe, in the sense that the transition probability is lower. This property mimics the probability of collecting a reward in a bandit problem.

In fig. 6 is shown a depiction of the environment. There are a total of $n+1$ states, and one parameters $p_0 \in (0, 1)$. In the figure, each tuple $(a, p)$ represents the action $a$ that triggers the transition and the probability $p$ of that event. The notation $a_{n_0:n_0+n}$ indicates all the actions in $\{a_{n_0}, \ldots, a_{n_0+n}\}$.

The agent starts in state $s_0$, and in every state she can select between $n$ actions $\{a_0, a_1, \ldots, a_{n-1}\}$. For small values of $p_0$ it becomes difficult for the agent to move to different states. Similarly, it is harder to navigate to states $s_i$ for large values of $n$. We used $p_0 = 1$ and $n = 4$.

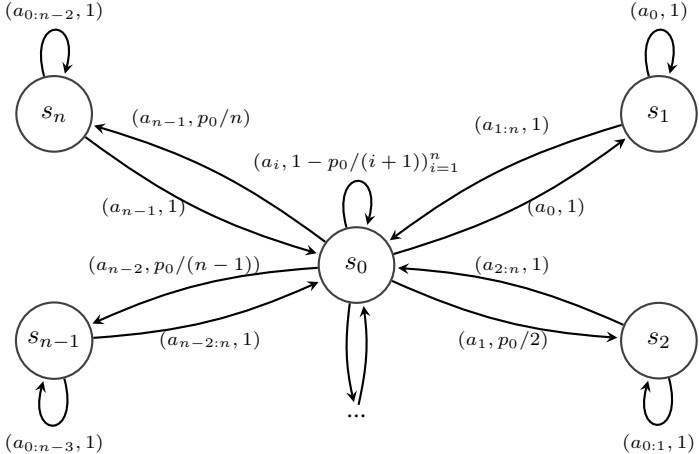

Figure 6: `NArms` environment. Each tuple $(a, p)$ represents the action $a$ that triggers the transition and the probability $p$ of that event. In the figure the notation $a_{n_0:n_0+n}$ indicates all the actions in $\{a_{n_0}, \ldots, a_{n_0+n}\}$. In state $s_0$ the probability to remain in $s_0$ for any action $a_i$ is $P(s_0|s_0, a_i) = 1 - p_0/(i+1)$, with the exception that $P(s_0|s_0, a_0) = 0$.

### D.1.2 Numerical Results

For the numerical results in the tabular case, we focus on 3 different aspects of the problem: (1) policy estimation error; (2) value estimation error; (3) exploration properties.

For the first two aspects we also consider the capacity of `MR-NaS` to generalize over rewards that do not belong to $\mathcal{R}$, specifically:

- `MR-NaS` uses the canonical basis of rewards $\mathcal{R}_{\text{canonical}}$ to compute the allocation $\omega_t^\star$ and explore the environments. We also use $\alpha = 0.99, \beta = 0.01$.

- We also test the generalization capacity of `MR-NaS` over a continuous set of rewards. At each time step we collect 30 additional rewards from $\mathcal{R}_{\text{rnd}} = \{R_1, \ldots, R_{30}\}$, which is a set of 30 reward vectors uniformly sampled from $[0, 1]^{SA}$, different for each seed. Note that these vectors are not used by `MR-NaS` to drive its exploration [8].

We consider a discount factor of $\gamma = 0.9$, confidence $\delta = 10^{-2}$ and run each algorithm for $T = 5 \cdot 10^5$ steps. In all experiments we depict the average value and its $95\%$ confidence interval over 100 seeds.

`MR-PSRL` **algorithm and** `RF-UCRL`. `MR-PSRL` is a simple adaptation of PSRL (posterior sampling for RL [35]). PSRL is an episodic algorithm that draws an MDP estimate from a posterior distribution. We adapt PSRL by sampling at the beginning of an episode a random reward vector $r$ in $[0, 1]^{SA}$,

---

[8]Since we use the canonical basis the second assumption (CH) in thm. B.3 is satisfied.

and exploring according to $M_{t,r}$ for that episode. In our case, we simply draw $r \sim \mathrm{Dir}(\mathbf{1})$ from a Dirichlet distribution, where $\mathbf{1}$ is a vector of ones of size $SA$. Since we do not consider a finite horizon setting, and both RF-UCRL and RF-PSRL are episodic algorithms, we set an horizon $H = 1/(1-\gamma)$.

**Policy estimation error.** We report in fig. 7 the estimation error of the optimal policies. In the plots we depict the average value of $e_{t,r} = \frac{|\Pi^\star(M_r) \triangle \Pi^\star(M_{t,r})|}{|\Pi^\star(M_r) \cup \Pi^\star(M_{t,r})|}$ for a reward $r$[9], where $A \triangle B$ is the symmetric difference between two sets $A, B$. Intuitively, $e_t$ accounts for both policies that are not accurately estimated and policies that are optimal in $M_{t,r}$ but not in $M_r$.

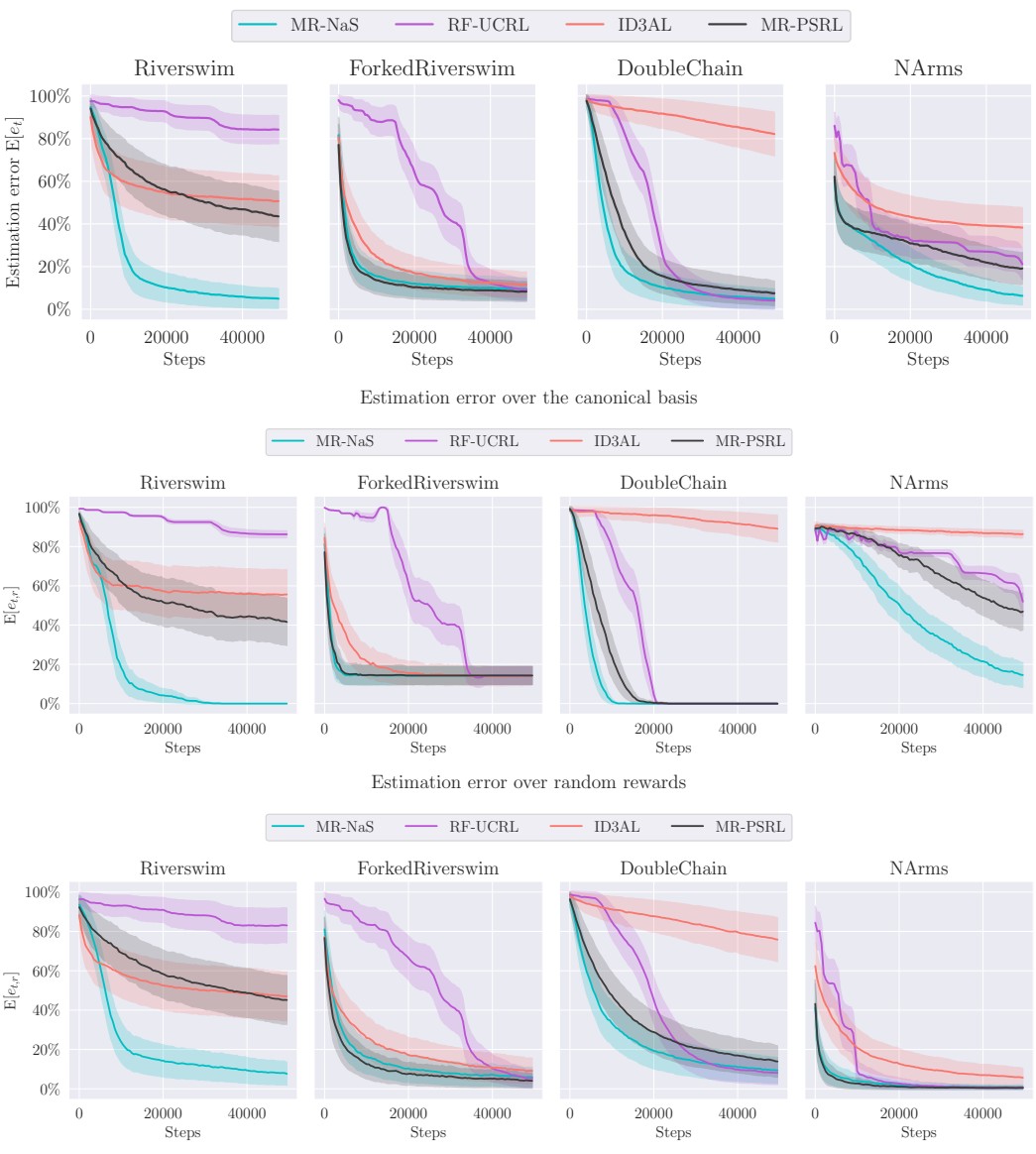

Figure 7: Estimation error of the optimal policies. The plots depict the average value of $e_{t,r} = \frac{|\Pi^\star(M_r) \triangle \Pi^\star(M_{t,r})|}{|\Pi^\star(M_r) \cup \Pi^\star(M_{t,r})|}$ ($r$ is a random quantity) and its $95\%$ confidence interval. In the first plot we combine together the results for the canonical basis $\mathcal{R}_{\mathrm{canonical}}$ and the random rewards $\mathcal{R}_{\mathrm{rnd}}$. In the second plot we only consider the canonical basis of rewards $\mathcal{R}_{\mathrm{canonical}}$, whilst in the third plot we only consider the random rewards $\mathcal{R}_{\mathrm{rnd}}$.

---

[9]In the expectation the reward $r$ is a random variable.

In the first plot we combine together the results for the canonical basis $\mathcal{R}_{\text{canonical}}$ and the random rewards $\mathcal{R}_{\text{rnd}}$. In the second plot we only consider the canonical basis of rewards $\mathcal{R}_{\text{canonical}}$, whilst in the third plot we only consider the random rewards $\mathcal{R}_{\text{rnd}}$. When consider only the canonical basis the advantage of `MR-NaS` is more noticeable, especially in the `Riverswim` and `NArms` environments (observe that for `Forked Riverswim` to bring the estimation error to $0$ requires a far greater number of samples since the environment is more difficult and the canonical basis is larger).

We also see from the last plot, the estimation error over random rewards, that `MR-NaS` is able to identify the correct optimal policies over rewards uniformly sampled form $[0,1]^{SA}$.

**Value estimation error.**    We report in fig. 8 the value estimation error over different rewards. In the plots we depict the average value of $\mathbb{E}\left[\frac{\|V_r - V_{t,r}\|_1}{S}\right]$, where the reward $r$ is a random variable when accounting for the random rewards in $\mathcal{R}_{\text{rnd}}$.

From the results we observe the good performance of `MR-NaS`, despite the fact that the objective of `MR-NaS` is not to optimize the value estimation error.

**Exploration properties.**    In this paragraph we go more into the details of the exploration properties of the various algorithms. In particular, we focus on three different metrics.

1. The deviation in visitation time, defined as $\Delta_{\text{visit},t} = \max_{s'} t_{\text{visit}}(s') - \min_s t_{\text{visit}}(s)$, where $t_{\text{visit}}(s)$ is the time step the algorithm last visited state $s$. This metric $\Delta_{\text{visit},t}$ quantifies how much an algorithms favours visitation of some states compared to others. In fig. 9 we show the results. We see that the initial transient of `MR-NaS` is relatively quick, and then converge to some steady-state value. While `MR-NaS` tends to focus on some states, it also manages to keep visiting all the other states, keeping the deviation to a small value.

2. The least visited state-action pair $\min_{s,a} N_t(s,a)$. This metric is particularly important since it is a proxy value for the value estimation error and the policy estimation error. The top figure in fig. 10 show this metric. Also in this case we see a fast initial transient for `MR-NaS`.

3. The last metric we consider is the entropy of the number of state-action visits $\mathcal{H}(N_t)$, defined as

$$\mathcal{H}(N_t) = -\sum_{s,a} p_t(s,a) \log(p_t(s,a)), \text{ where } p_t(s,a) = \frac{N_t(s,a)}{t}.$$

This metric intuitively quantifies how spread is the exploration across the state-action space. The bottom figure in fig. 10 shows this metric normalized in $[0,1]$. Larger values indicate that the exploration is more uniform across the state-action space.

**Compute resources and additional information.**    For these simulations we used 1 G5.4XLARGE AWS instance with 16 vCPUs, 64 GiB of memory and 1 A10G GPU with 24 GiB of memory. To obtain all the results 2-3 days are needed. The entire research project needed roughly 15 days of computation time for this experiment.

We set up our experiments using Python 3.11 [88] (for more information, please refer to the following link http://www.python.org), and made use of the following libraries: NumPy [89], SciPy [90], CVXPY [91], Seaborn [92], Pandas [93], Matplotlib [94]. In CVXPY we used the CLARABEL optimizer [95] and/or the ECOS optimizer [96]. Changes, and new code, are published under the MIT license. To run the code, please, read the attached README file for instructions.

### D.2   Numerical Results: Cartpole swing-up and DeepSea problems

In this section we discuss the numerical results on the Cartpole swing-up problem and the DeepSea problem.

**Libraries used in the experiments.**    We set up our experiments using Python 3.11 [88] (For more information, please refer to the following link http://www.python.org), and made use of the following libraries: Cython [97], NumPy [89], SciPy [90], PyTorch [98], Seaborn [92], Pandas [93], Matplotlib [94]. Changes, and new code, are published under the MIT license. To run the code,

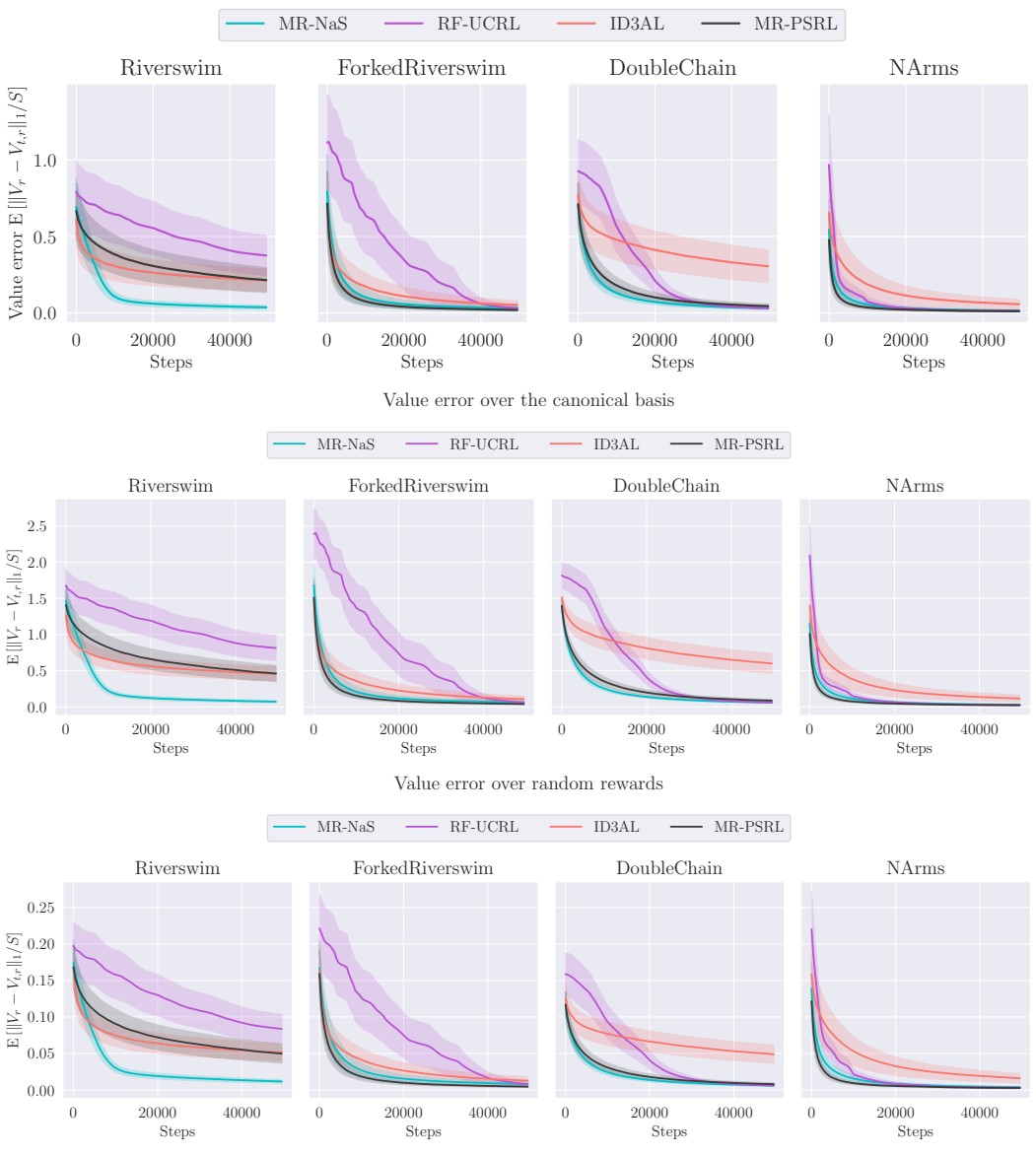

Figure 8: Value estimation error. The plots depict the average value of $\mathbb{E}\left[\frac{\|V_r - V_{t,r}\|_1}{S}\right]$,. In the first plot we combine together the results for the canonical basis $\mathcal{R}_{\text{canonical}}$ and the random rewards $\mathcal{R}_{\text{rnd}}$. In the second plot we only consider the canonical basis of rewards $\mathcal{R}_{\text{canonical}}$, whilst in the third plot we only consider the random rewards $\mathcal{R}_{\text{rnd}}$.

please, read the attached README file for instructions. In the code, we make use of some code from the Behavior suite [36], which is licensed with the APACHE 2.0 license.

### D.2.1 Cartpole swing-up

**Environment details.** The cartpole swing-up problem is a well-known challenge in control theory and reinforcement learning [99]. The objective is to balance a pole attached to a cart by an unactuated joint, where the cart moves along a frictionless track. Control is exerted by applying force to the cart. Initially, the pole hangs downward, and the goal is to swing it up into an upright position and maintain its balance there. Unlike the traditional cartpole balancing problem, this task requires both swinging the pole up and keeping it balanced once upright.

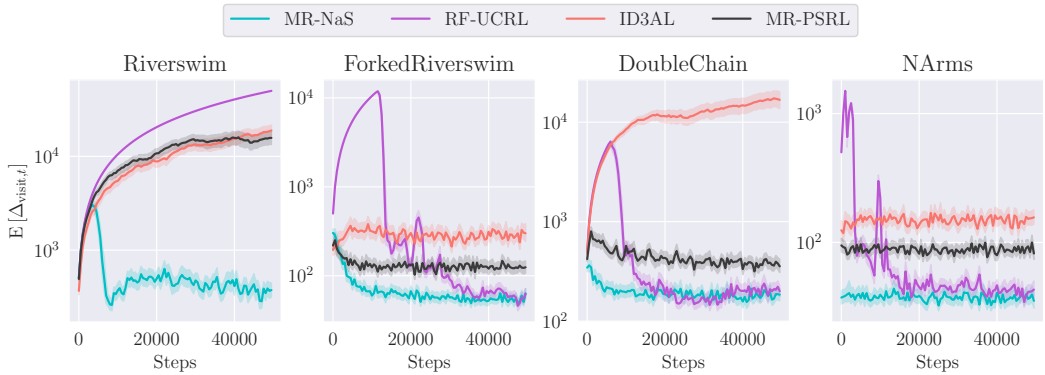

Figure 9: Average value of $\Delta_{\text{visit},t} = \max_{s'} t_{\text{visit}}(s') - \min_s t_{\text{visit}}(s)$, where $t_{\text{visit}}(s)$ is the time step the algorithm last visited state $s$.

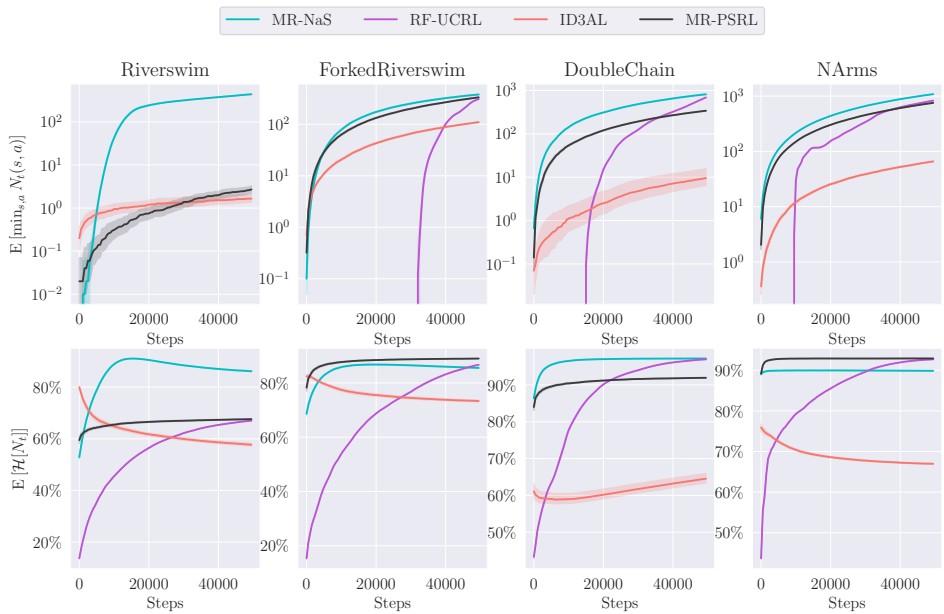

Figure 10: Top: average value of the least visited state-action pair $\min_{s,a} N_t(s, a)$. Bottom: normalized average value of the entropy of $N_t$ over time (see text for a definition).

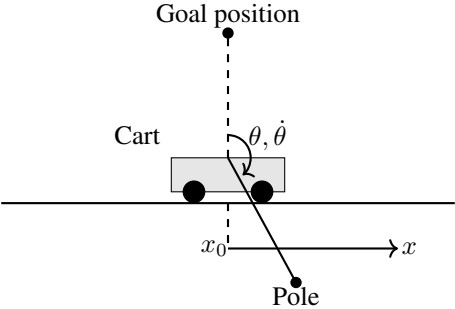

Figure 11: Diagram of the Cartpole swing-up problem.

The system's state at any given moment is defined by four variables: the cart's position $x$, the cart's velocity $\dot{x}$, the pole's angle $\theta$, and the pole's angular velocity $\dot{\theta}$. There are also four additional variables in the state, and for brevity, we direct the reader to [36].

This problem is challenging because the reward is sparse and includes a penalty for movement.

**Reward functions.** In the multi-reward variant the objective is to stabilize the pole vertically around a specified position $x_0$. The agent receives a positive reward only if the following conditions are met: (1) $\cos(\theta) > k/20$ for the pole's angle, (2) $|\dot{\theta}| \leq \theta_0$ for its angular velocity, and (3) $|x - x_0| \leq 1 - k/20$ for the cart's position, where $\theta_0$ is a constant and $k$ is an integer indicating the difficulty level. Additionally, the agent incurs a negative reward of $-0.1$ for moving, which hinders the learning process. Classical algorithms like `DQN` [100] tend to remain stationary due to this penalty.

Then, the reward at time $t$ for a given value of $x_0$ is:

$$ r_{x_0}(s_t, a_t) = -0.1 + \left( \mathbf{1}_{\cos(\theta_t) > k/20} \cdot \mathbf{1}_{|\dot{\theta}_t| \leq \theta_0} \cdot \mathbf{1}_{|x_t - x_0| \leq 1 - k/20} \right). $$

We considered a set of rewards $\mathcal{R}_{\text{base}} = (r_{x_0})_{x_0 \in \mathcal{X}}$ that consists of 5 rewards for different values for $x_0$, where $x_0$ is evenly spaced in $[-1.5, 1.5]$ (that is $\mathcal{X} = \{-1.5, -0.75, 0, 0.75, 1\}$). To evaluate `DBMR-BPI`'s ability to generalize to unseen rewards, we also evaluate the rewards collected in the set $\mathcal{R}_{\text{rnd}}$, which consists of 5 reward functions where, for each reward, $x_0$ is uniformly sampled in $[-1.5, 1.5]$. The set $\mathcal{R}_{\text{rnd}}$ is different for each seed, and observe that the random rewards in $\mathcal{R}_{\text{rnd}}$ were not used during training of `DBMR-BPI`.

**Numerical results: performance.** In figs. 12 and 13 we depict the performance of the various algorithms over $\mathcal{R}_{\text{base}}$ and $\mathcal{R}_{\text{rnd}}$ for two difficulty levels $k \in \{3, 5\}$. In tabs. 4 and 5 we report the values at the last time step $T$ of the various metrics considered.

In particular, we focus on the distribution of the following random variable

$$ X_i = \frac{1}{T} \sum_{t=1}^{T} \mathbf{1}_{r_i(s_t, a_t) > 0}, $$

where $i$ is the $i$-th reward in the corresponding set of rewards. This metric measures the average amount of information collected by the agent useful to learn how to stabilize the pole upright. In the figures, and in the tables, the quantity $\text{Avg}_{\mathcal{R}}(\{X_i\})$ (sim. the others metrics) indicates the arithmetic mean of $\{X_i\}_i$ over the rewards. We average results across 30 seeds.

|  |  | **Base rewards $\mathcal{R}$** | | | | |
|---|---|---|---|---|---|---|
|  |  | $\mathbb{E}[\text{med}_{\mathcal{R}}(X_i)]$ | $\mathbb{E}[\text{Avg}_{\mathcal{R}}(X_i)]$ | $\mathbb{E}[\text{std}_{\mathcal{R}}(X_i)]$ | $\mathbb{E}[\text{max}_{\mathcal{R}}(X_i)]$ | $\mathbb{E}[\text{min}_{\mathcal{R}}(X_i)]$ |
| $k = 3$ | DBMR-BPI | 21.8(20.9, 22.7) | 21.3(20.6, 21.9) | 6.1(5.5, 6.6) | 28.9(27.6, 30.2) | 12.8(12.0, 13.6) |
| $T = 150000$ | RND | 1.3(1.0, 1.7) | 1.3(1.0, 1.6) | 0.3(0.2, 0.3) | 1.7(1.3, 2.1) | 0.9(0.7, 1.2) |
|  | APT | 0.3(0.3, 0.4) | 0.3(0.3, 0.4) | 0.1(0.1, 0.2) | 0.5(0.4, 0.6) | 0.2(0.1, 0.2) |
|  | Disagreement | 0.9(0.7, 1.2) | 0.9(0.7, 1.2) | 0.3(0.2, 0.6) | 1.3(0.9, 2.0) | 0.5(0.4, 0.6) |
| $k = 5$ | DBMR-BPI | 21.6(20.9, 22.3) | 20.2(19.6, 20.6) | 6.4(6.0, 6.9) | 28.0(27.0, 29.0) | 11.4(10.7, 12.0) |
| $T = 200000$ | RND | 1.2(0.9, 1.4) | 1.2(0.9, 1.5) | 0.2(0.2, 0.3) | 1.5(1.2, 1.9) | 0.8(0.6, 1.1) |
|  | APT | 0.3(0.3, 0.4) | 0.3(0.3, 0.4) | 0.1(0.1, 0.1) | 0.5(0.4, 0.6) | 0.2(0.1, 0.2) |
|  | Disagreement | 0.7(0.5, 0.9) | 0.7(0.5, 1.0) | 0.3(0.2, 0.4) | 1.2(0.7, 1.6) | 0.4(0.3, 0.6) |

Table 4: Cartpole swing-up. Statistics for the random variable $X_i = \frac{1}{T} \sum_{t=1}^{T} \mathbf{1}_{r_i(s_t, a_t) > 0}$, with $r_i \in \mathcal{R}_{\text{base}}$. Values are multiplied by 100 and rounded to the first digit. In bold statistically significant results (in parentheses we indicate the 95% confidence interval).

In general, we see that `DBMR-BPI` outperforms all of the unsupervised learning baselines. Interestingly, even though `DBMR-BPI` uses the base set of rewards $\mathcal{R}_{\text{base}}$ to explore, we see better performance when collecting rewards from $\mathcal{R}_{\text{rnd}}$ (which are not used by `DBMR-BPI` during exploration). The reason may be that some rewards in $\mathcal{R}_{\text{base}}$, (e.g., those with $|x_0| = 1.5$), are harder to learn, while it is unlikely for the random rewards to have such values of $x_0$. This result seems to suggest that the data collected by `DBMR-BPI` could also be used for rewards that are similar to those in $\mathcal{R}$.

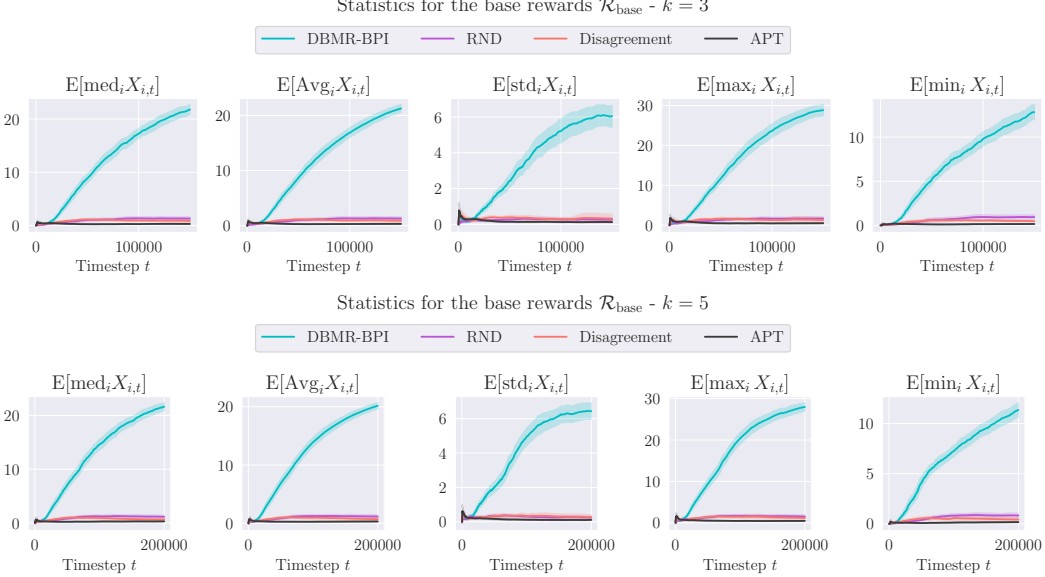

Figure 12: Cartpole swing-up. Statistics for the random variable $X_i = \frac{1}{T}\sum_{t=1}^{T} \mathbf{1}_{r_i(s_t,a_t)>0}$, with $r_i \in \mathcal{R}_{\text{base}}$. Values are multiplied by 100 and rounded to the first digit.

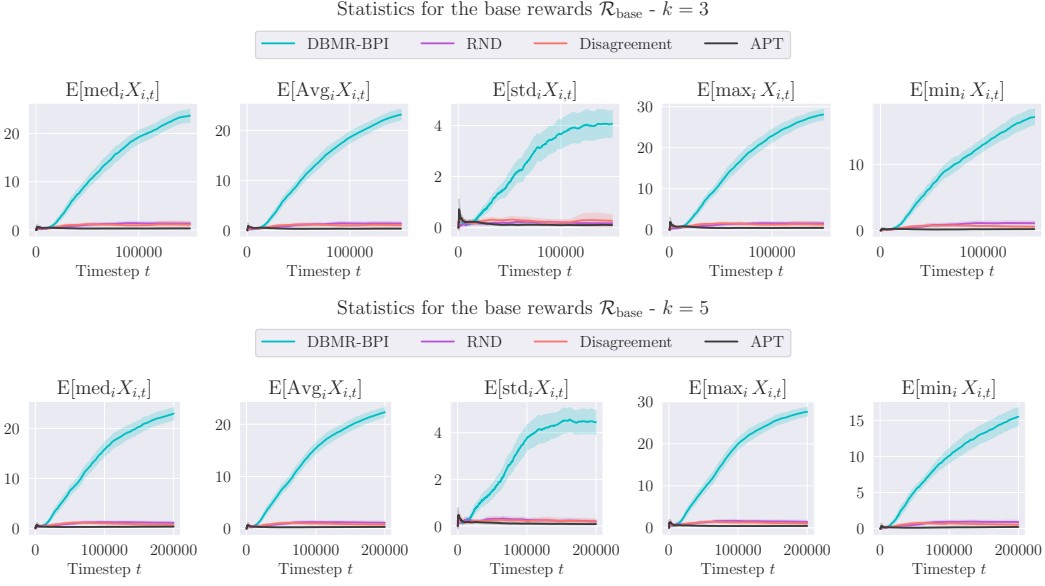

Figure 13: Cartpole swing-up. Statistics for the random variable $X_i = \frac{1}{T}\sum_{t=1}^{T} \mathbf{1}_{r_i(s_t,a_t)>0}$, with $r_i \in \mathcal{R}_{\text{rnd}}$. Values are multiplied by 100 and rounded to the first digit.

**Numerical results: rewards chosen by** `DBMR-BPI` **.** In fig. 14 we plot the value of $\Delta_{t,r}$ estimated from `DBMR-BPI` . We see that according to the algorithm the third reward, which corresponds to $x_0 = 0$, is initially the easiest to learn. This fact is also depicted in fig. 15, which shows the number of time a certain reward was picked by `DBMR-BPI` for exploration (the plot does not account the uniform exploration over rewards, or the tracking procedure that ensures each rewards is selected at a rate of $O(\sqrt{n})$, where $n$ is the number of episodes). Rewards corresponding to $|x_0| = 1.5$ are harder to learn since the episode terminates if the agent exceeds the $x$ boundary of the environment.

|  |  | **Random rewards** $\mathcal{R}_{\mathrm{rnd}}$ | | | | |
|---|---|---|---|---|---|---|
|  |  | $\mathbb{E}[\mathrm{med}_{\mathcal{R}_{\mathrm{rnd}}}(X_i)]$ | $\mathbb{E}[\mathrm{Avg}_{\mathcal{R}_{\mathrm{rnd}}}(X_i)]$ | $\mathbb{E}[\mathrm{std}_{\mathcal{R}_{\mathrm{rnd}}}(X_i)]$ | $\mathbb{E}[\mathrm{max}_{\mathcal{R}_{\mathrm{rnd}}}(X_i)]$ | $\mathbb{E}[\mathrm{min}_{\mathcal{R}_{\mathrm{rnd}}}(X_i)]$ |
| $k = 3$ | `DBMR-BPI` | 23.7(22.5, 24.9) | 23.1(22.2, 24.2) | 4.1(3.6, 4.6) | 28.1(26.9, 29.3) | 17.1(16.0, 18.2) |
| $T = 150000$ | RND | 1.3(1.1, 1.7) | 1.4(1.1, 1.7) | 0.2(0.1, 0.2) | 1.6(1.2, 2.0) | 1.1(0.9, 1.4) |
|  | APT | 0.3(0.3, 0.4) | 0.3(0.3, 0.4) | 0.1(0.1, 0.1) | 0.5(0.4, 0.6) | 0.2(0.2, 0.2) |
|  | Disagreement | 1.1(0.7, 1.8) | 1.0(0.7, 1.6) | 0.3(0.1, 0.5) | 1.3(0.8, 2.0) | 0.6(0.5, 0.7) |
| $k = 5$ | `DBMR-BPI` | 23.0(21.8, 24.1) | 22.3(21.5, 23.1) | 4.5(4.0, 4.9) | 27.6(26.7, 28.6) | 15.5(14.4, 16.7) |
| $T = 200000$ | RND | 1.1(0.9, 1.4) | 1.1(0.9, 1.4) | 0.2(0.1, 0.2) | 1.4(1.1, 1.7) | 0.9(0.7, 1.1) |
|  | APT | 0.3(0.3, 0.4) | 0.3(0.3, 0.4) | 0.1(0.1, 0.1) | 0.4(0.4, 0.5) | 0.2(0.2, 0.3) |
|  | Disagreement | 0.7(0.5, 0.9) | 0.8(0.5, 1.0) | 0.2(0.1, 0.3) | 1.0(0.7, 1.4) | 0.5(0.4, 0.7) |

Table 5: Cartpole swing-up. Statistics for the random variable $X_i = \frac{1}{T}\sum_{t=1}^{T} \mathbf{1}_{r_i(s_t,a_t)>0}$, with $r_i \in \mathcal{R}_{\mathrm{rnd}}$. Values are multiplied by 100 and rounded to the first digit. In bold statistically significant results (in parentheses we indicate the 95% confidence interval).

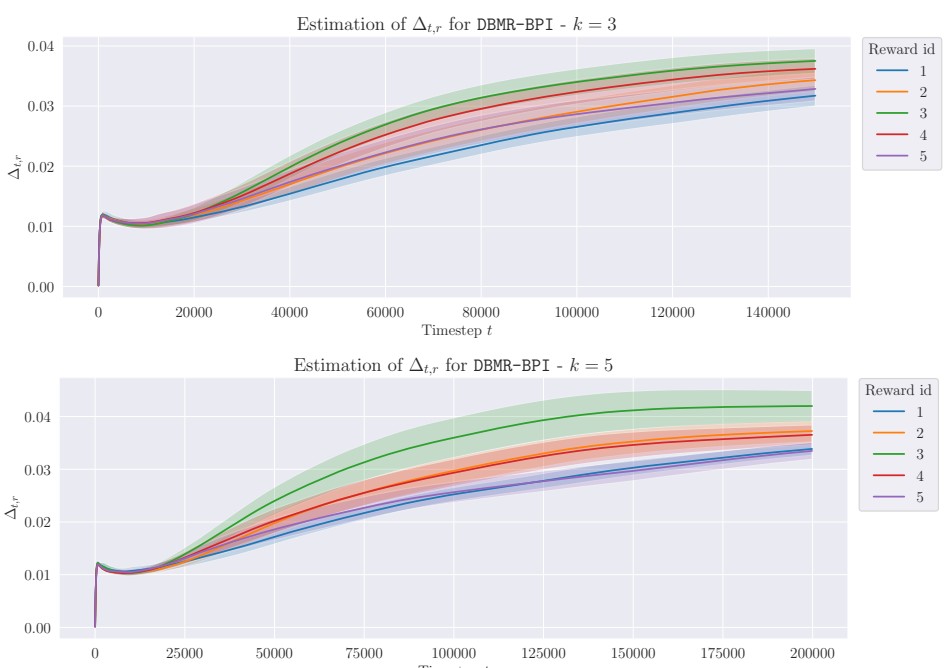

Figure 14: Estimate $\Delta_{t,r}$ of `DBMR-BPI` over time steps. The $i$-th reward corresponds to the $i$-th reward in $\mathcal{R}_{\mathrm{base}}$. Shaded area corresponds to the 95% confidence interval over 30 seeds.

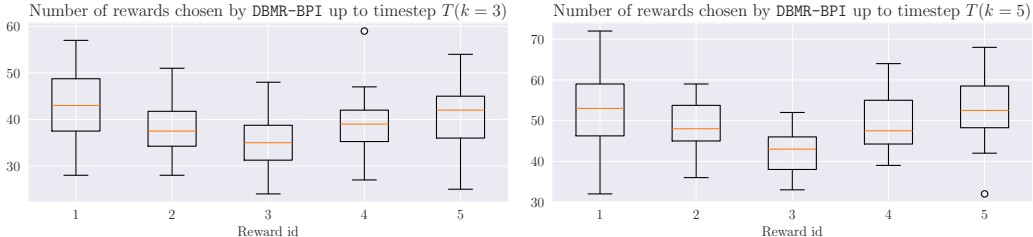

Figure 15: Distribution of the number of times a certain reward $r_i$ was picked by `DBMR-BPI` up to time step $T$. The $i$-th reward corresponds to the $i$-th reward in $\mathcal{R}_{\mathrm{base}}$. The plot does not account for rewards chosen uniformly at random or due to forced tracking.

**Parameters and compute resources.** For these simulations we used 1 G5.4XLARGE AWS instance with 16 vCPUs, 64 GiB of memory and 1 A10G GPU with 24 GiB of memory. To obtain all the results 3-4 days are needed. The entire research project needed roughly 21 days of computation time for this experiment.

The parameters are listed in tab. 6. Refer to app. E for further details on the parameters and the algorithms.

| DBMR-BPI parameters | Value | APT parameters | Value | Disagreement parameters | Value | RND parameters | Value |
|---|---|---|---|---|---|---|---|
| $N_{ensemble}$ | 20 | $N_{dqn}$ | $[128, 128]$ | $N_{dqn}$ | $[128, 128]$ | $N_{dqn}$ | $[128, 128]$ |
| $N_{dbmrbpi}$ | $[128, 128]$ | $C$ | $2 \cdot 10^5$ | $C$ | $2 \cdot 10^5$ | $C$ | $2 \cdot 10^5$ |
| $\tau$ | $10^{-2}$ | $\tau$ | $10^{-2}$ | $\tau$ | $10^{-2}$ | $\tau$ | $10^{-2}$ |
| $\alpha_Q, \alpha_M$ | $(5 \cdot 10^{-4}, 5 \cdot 10^{-5})$ | $\alpha$ | $5 \cdot 10^{-4}$ | $\alpha$ | $5 \cdot 10^{-4}$ | $\alpha$ | $5 \cdot 10^{-4}$ |
| $p_{uniform}$ | 0.5 | $N_{rep}$ | 512 | $N_{ensemble}$ | 20 | $N_{rep}$ | 512 |
| $p_{mask}$ | 0.7 | $N_{trunk}$ | $[512]$ | $N_{disag}$ | $[128]$ | $N_{rnd}$ | $[128]$ |
| $k$ | 2 | $N_F$ | $[128]$ | $\alpha_{disag}$ | $10^{-4}$ | $\alpha_{rnd}$ | $10^{-4}$ |
| $s_{prior}$ | 6 | $N_B$ | $[128]$ | | | | |
| $C$ | $2 \cdot 10^5$ | $\alpha_{apt}$ | $10^{-4}$ | | | | |
| | | $K_k$ | 12 | | | | |

Table 6: Parameters and values of the algorithms. For all algorithms we used a batch size of 128.

### D.2.2 DeepSea

**Environment details.** The DeepSea problem is a hard-exploration reinforcement learning challenge set in an $N \times N$ grid. The agent starts in the top-left corner (state $(0,0)$) and, in the classical version, aims to reach the bottom-right corner (state $(N-1, N-1)$) for a large reward. The state vector is an $N^2$-dimensional vector that one-hot encodes the agent's position in the grid. The agent can move diagonally, left or right, and down when near the wall. Reaching the bottom-right corner yields a reward of 1. We also incorporate slipping, as described in [53], so that when the agent selects an action $a$, there is a $5\%$ probability that a different action $a' \neq a$ is executed instead. Lastly, for each seed, in each state the action mapping is randomized (so that in two different states an action $a$ can correspond to go left in the first state, and go right in the second state).

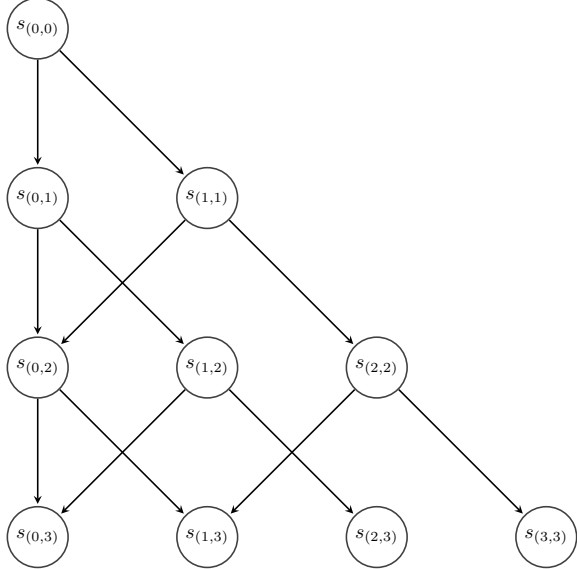

Figure 16: Depiction of the Deepsea problem as a tabular MDP with $N = 4$.

**Reward functions.** We adapt the DeepSea environment to a multi-reward setting by introducing $N$ different rewards, $\mathcal{R}_{\text{base}} = \{r_1 \cdots, r_N\}$, where $r_i$ assigns a positive reward whenever the agent reaches column $i$ in the last row of the grid, that is:

$$r_i(s_t, a_t) = \mathbf{1}_{s_t = (N-1, i)}.$$

To evaluate the capability of `DBMR-BPI` in collecting unseen rewards, we sampled 20 reward functions $\mathcal{R}_{\text{rnd}} = (r'_i)_{i=1}^{20}$ for each seed. Each reward function is essentially a vector of dimension $N(N+1)/2$, corresponding to the possible positions in the grid, and is sampled from a Dirichlet distribution with parameter $\alpha = (\alpha_i)_i$, where $\alpha_i = 1/(10N)$ for all $i = 1, \cdots, N(N+1)/2$.

When the agent is in state $s$, corresponding to row $k$, column $j$, it collects the corresponding reward $(r_i)_{j,k}$ from the reward function $r'_i$. We then computed the rewards for these additional reward functions to assess performance. Note that these rewards are not used by `DBMR-BPI` during training.

**Numerical results: performance.** In figs. 17 and 18 we depict the performance of the various algorithms over $\mathcal{R}_{\text{base}}$ and $\mathcal{R}_{\text{rnd}}$ for two sizes $N \in \{20, 30\}$. In tabs. 7 and 8 we report the values at the last time step $T$ of the various metrics considered.

In particular, we focus on the distribution of the following random variable

$$X_i = \frac{1}{T} \sum_{t=1}^{T} r_i(s_t, a_t),$$

where $i$ is the $i$-th reward in the corresponding set of rewards. This metric measures the average reward collected for the $i$-th reward in a specific instance.

In the figures, and in the tables, the quantity $\text{Avg}_{\mathcal{R}}(\{X_i\})$ (sim. the others metrics) indicates the arithmetic mean of $\{X_i\}_i$ over the rewards, while $\text{GM}_{\mathcal{R}}(\{X_i\})$ indicates the geometric mean over the rewards. We average results across 30 seeds.

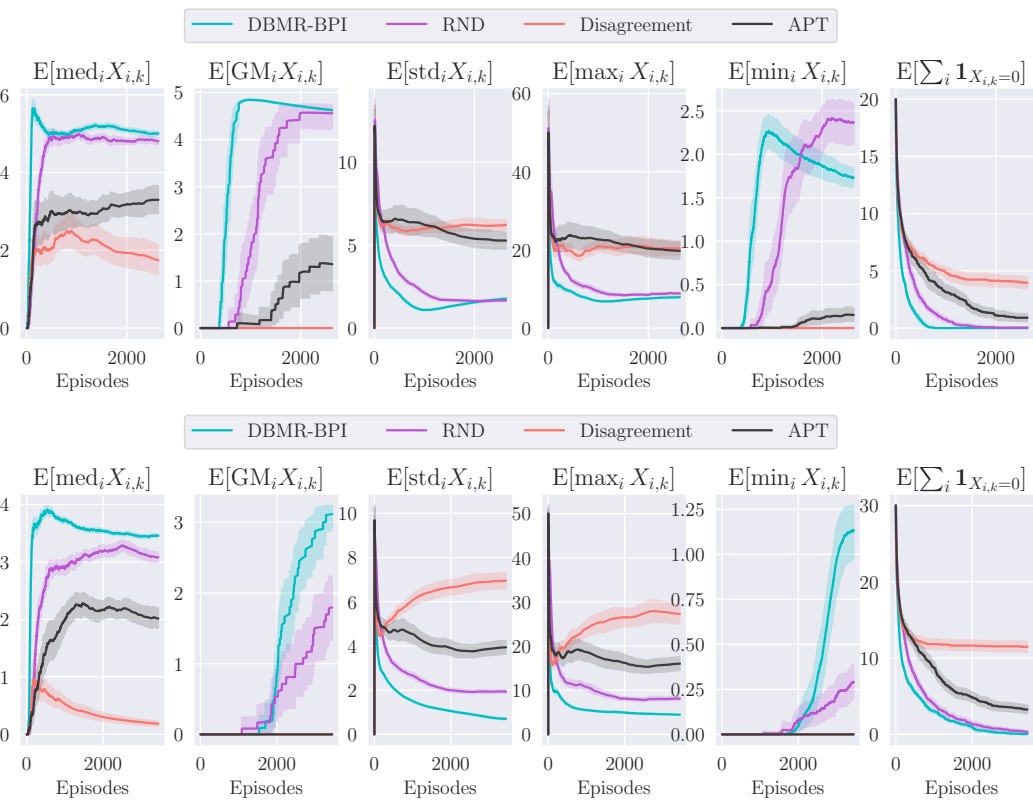

Figure 17: DeepSea environment. Statistics for the random variable $X_i = \frac{1}{T}\sum_{t=1}^{T} r_i(s_t, a_t)$ (with $r_i \in \mathcal{R}_{\text{base}}$). Values are multipled by 100 and rounded to the first digit (except for the last plot). Shaded areas depict 95% confidence intervals.

| | | Base rewards $\mathcal{R}_{\text{base}}$ | | | | | |
| | | $\mathbb{E}[\text{med}_{\mathcal{R}}(X_i)]$ | $\mathbb{E}[\text{GM}_{\mathcal{R}}(X_i)]$ | $\mathbb{E}[\text{std}_{\mathcal{R}}(X_i)]$ | $\mathbb{E}[\text{max}_{\mathcal{R}}(X_i)]$ | $\mathbb{E}[\text{min}_{\mathcal{R}}(X_i)]$ | $\mathbb{E}[\sum_i \mathbf{1}_{X_i=0}]$ |
|---|---|---|---|---|---|---|---|
| $N=20$ | DBMR-BPI | 5.0(4.9, 5.1) | 4.6(4.6, 4.6) | 1.8(1.7, 1.8) | 7.9(7.8, 8.0) | 1.7(1.6, 1.8) | 0.0(0.0, 0.0) |
| $T=50000$ | RND | 4.8(4.7, 4.9) | 4.6(4.2, 4.7) | 1.7(1.6, 1.8) | 8.9(8.5, 9.2) | 2.4(2.1, 2.6) | 0.0(0.0, 0.1) |
| | APT | 3.3(3.0, 3.6) | 1.4(0.8, 1.9) | 5.3(4.8, 5.8) | 19.7(17.7, 21.9) | 0.2(0.1, 0.2) | 0.9(0.6, 1.3) |
| | Disagreement | 1.7(1.4, 2.1) | 0.0(0.0, 0.0) | 6.2(5.9, 6.5) | 20.4(18.7, 22.2) | 0.0(0.0, 0.0) | 4.0(3.4, 4.5) |
| $N=30$ | DBMR-BPI | 3.5(3.4, 3.5) | 3.1(2.9, 3.2) | 0.7(0.7, 0.7) | 4.4(4.3, 4.6) | 1.1(1.0, 1.3) | 0.0(0.0, 0.1) |
| $T=100000$ | RND | 3.1(3.0, 3.2) | 1.8(1.3, 2.2) | 1.9(1.9, 2.0) | 8.1(7.7, 8.5) | 0.3(0.2, 0.4) | 0.3(0.2, 0.5) |
| | APT | 2.0(1.8, 2.2) | 0.0(0.0, 0.0) | 3.9(3.6, 4.2) | 16.0(14.5, 17.6) | 0.0(0.0, 0.0) | 3.3(2.8, 3.8) |
| | Disagreement | 0.2(0.1, 0.2) | 0.0(0.0, 0.0) | 7.0(6.6, 7.3) | 27.3(25.1, 29.6) | 0.0(0.0, 0.0) | 11.5(10.8, 12.2) |

Table 7: DeepSea environment. Statistics for the random variable $X_i = \frac{1}{T}\sum_{t=1}^{T} r_i(s_t, a_t)$ (with $r_i \in \mathcal{R}_{\text{base}}$). Values are multipled by 100 and rounded to the first digit. In parentheses we indicate the 95% confidence interval.

For the base case, we report the geometric mean over the rewards, since the arithmetic mean would be a constant for this specific setting (the number of visits in the last row, normalized by $t$, can be seen as a frequency, and sums up to 1). hence, for this setting the geometric mean is more appropriate [58].

We also report the metric $\sum_{i=1}^{N} \mathbf{1}_{X_i=0}$, which indicates the number of cells in the last row of the grid that have yet to be visited at a certain time step.

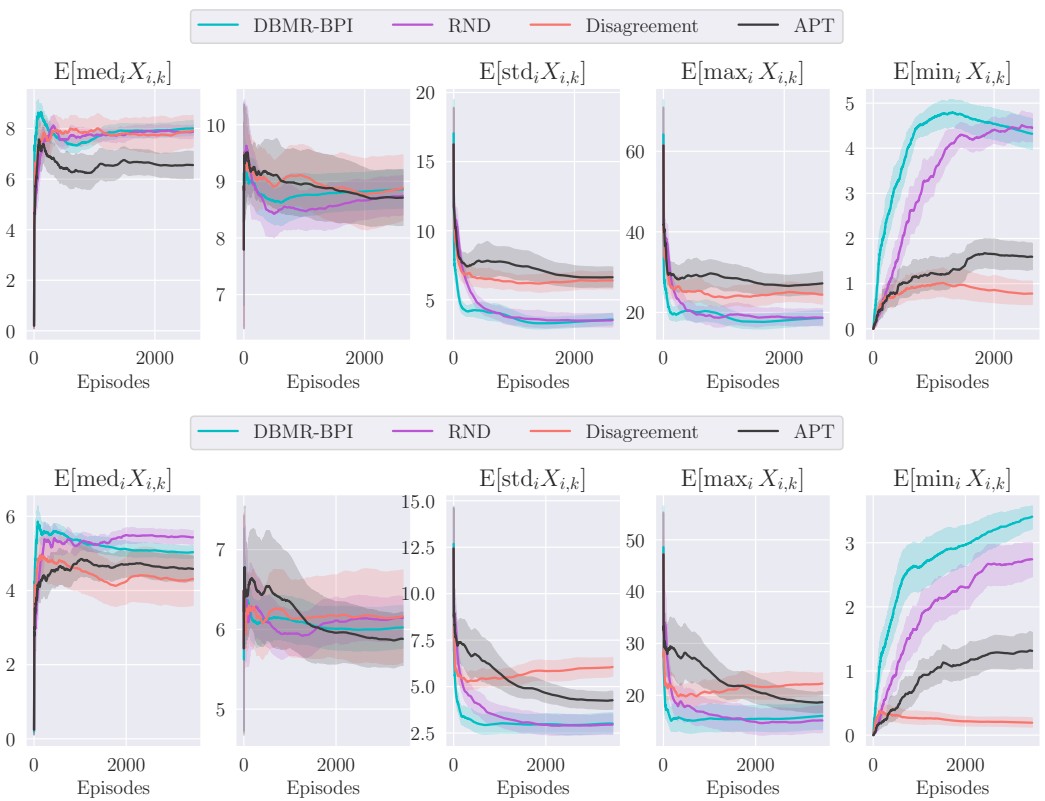

Figure 18: DeepSea environment. Statistics for the random variable $X_i = \frac{1}{T}\sum_{t=1}^{T} r_i(s_t, a_t)$ (with $r_i \in \mathcal{R}_{\mathrm{rnd}}$). Values are multiplied by $100$ and rounded to the first digit (except for the last plot). Shaded areas depict $95\%$ confidence intervals.

| | | **Random rewards $\mathcal{R}_{\mathrm{rnd}}$** | | | | |
|---|---|---|---|---|---|---|
| | | $\mathbb{E}[\mathrm{med}_{\mathcal{R}_{\mathrm{rnd}}}(X_i)]$ | $\mathbb{E}[\mathrm{Avg}_{\mathcal{R}_{\mathrm{rnd}}}(X_i)]$ | $\mathbb{E}[\mathrm{std}_{\mathcal{R}_{\mathrm{rnd}}}(X_i)]$ | $\mathbb{E}[\mathrm{max}_{\mathcal{R}_{\mathrm{rnd}}}(X_i)]$ | $\mathbb{E}[\mathrm{min}_{\mathcal{R}_{\mathrm{rnd}}}(X_i)]$ |
| $N = 20$ | DBMR-BPI | $8.0(7.7, 8.3)$ | $8.9(8.5, 9.2)$ | $3.6(3.2, 4.0)$ | $18.7(17.0, 20.5)$ | $4.3(4.0, 4.6)$ |
| $T = 50000$ | RND | $7.9(7.6, 8.1)$ | $8.7(8.4, 9.1)$ | $3.5(3.1, 4.0)$ | $18.6(16.7, 20.8)$ | $4.5(4.1, 4.8)$ |
| | APT | $6.6(6.0, 7.1)$ | $8.7(8.2, 9.2)$ | $6.6(5.9, 7.4)$ | $27.2(24.6, 30.1)$ | $1.6(1.3, 1.9)$ |
| | Disagreement | $7.9(7.3, 8.5)$ | $8.9(8.3, 9.5)$ | $6.4(5.8, 7.1)$ | $24.4(22.1, 26.8)$ | $0.8(0.5, 1.0)$ |
| $N = 30$ | DBMR-BPI | $5.0(4.9, 5.2)$ | $6.0(5.8, 6.3)$ | $3.0(2.5, 3.6)$ | $16.0(13.4, 18.8)$ | $3.4(3.2, 3.6)$ |
| $T = 100000$ | RND | $5.4(5.3, 5.6)$ | $6.1(5.9, 6.4)$ | $3.0(2.4, 3.5)$ | $15.1(12.7, 17.9)$ | $2.7(2.5, 3.0)$ |
| | APT | $4.6(4.3, 4.9)$ | $5.9(5.6, 6.2)$ | $4.3(3.8, 4.7)$ | $18.6(16.7, 20.7)$ | $1.3(1.0, 1.6)$ |
| | Disagreement | $4.3(3.6, 5.1)$ | $6.2(5.6, 6.7)$ | $6.1(5.6, 6.6)$ | $22.2(20.2, 24.4)$ | $0.2(0.1, 0.3)$ |

Table 8: DeepSea environment. Statistics for the random variable $X_i = \frac{1}{T}\sum_{t=1}^{T} r_i(s_t, a_t)$ (with $r_i \in \mathcal{R}_{\mathrm{rnd}}$). Values are multiplied by $100$ and rounded to the first digit. In parentheses we indicate the $95\%$ confidence interval.

When considering the base set of rewards $\mathcal{R}_{\mathrm{base}}$ used by DBMR-BPI , we see that APT and Disagremeent do not match the performance of DBMR-BPI and RND, focusing most of the time on a few rewards. On the other hand, while DBMR-BPI and RND show similar performance for $N = 20$, the difference increases for $N = 30$ on the base set of rewards. We also note that DBMR-BPI tend to focus less on specific rewards (on average $\max_i X_i$ is smaller). For larger $N$ DBMR-BPI seems also to explore more uniformly (lower standard deviation, and higher minimum). When considering the total reward collected from the set $\mathcal{R}_{\mathrm{rnd}}$, we see that DBMR-BPI achieves performance similar to RND, which confirms the capability of DBMR-BPI to collect reward from unseen reward functions.

**Numerical results: rewards chosen by DBMR-BPI.** In fig. 19 we plot the value of $\Delta_{t,r}$ estimated from DBMR-BPI. According to DBMR-BPI the rewards in $\mathcal{R}_{\mathrm{base}}$ are initially the hardest one to learn

($\Delta_{t,r}$ is small for those rewards), but then, as the algorithm eventually reaches those states, the estimated gap grows larger, and the hardest reward to learn is actually the first one, corresponding to the one that yields positive reward in state $(N-1, 0)$. This is indeed expected, since there are different paths in the MDP that the agent can take to reach that state, while to reach state $(N-1, N-1)$ there is only one possible sequence of actions (and one can verify that the sub-optimality gap is also larger).

Fig. 20 shows the number of time a certain reward was picked by `DBMR-BPI` for exploration (the plot does not account the uniform exploration over rewards, or the tracking procedure that ensures each rewards is selected at a rate of $O(\sqrt{n})$, where $n$ is the number of episodes). The last rewards (those with high id value) have been chosen the most since their sub-optimality gap was smaller for a longer period of time.

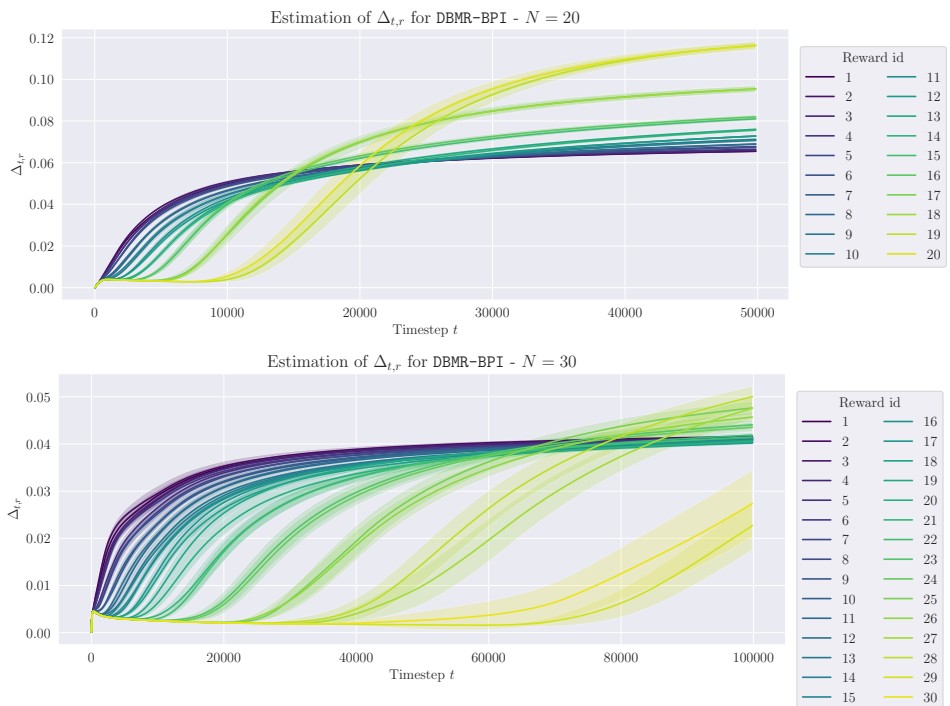

Figure 19: Estimate $\Delta_{t,r}$ of `DBMR-BPI` over time steps. The $i$-th reward corresponds to the $i$-th reward in $\mathcal{R}_{\mathrm{base}}$. Shaded area corresponds to the $95\%$ confidence interval over 30 seeds.

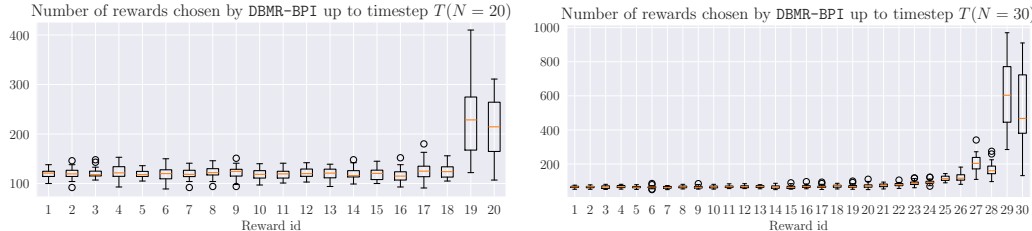

Figure 20: Distribution of the number of times a certain reward $r_i$ was picked by `DBMR-BPI` up to time step $T$. The $i$-th reward corresponds to the $i$-th reward in $\mathcal{R}_{\mathrm{base}}$. The plot does not account for rewards chosen uniformly at random or due to forced tracking.

**Parameters and compute resources.** For these simulations we used 1 G5.4XLARGE AWS instance with 16 vCPUs, 64 GiB of memory and 1 A10G GPU with 24 GiB of memory. To obtain all the results 2-3 days are needed. The entire research project needed roughly 18 days of computation time for this experiment.

The parameters are listed in tab. 9. Refer to app. E for further details on the parameters and the algorithms.

| `DBMR-BPI` parameters | Value | `APT` parameters | Value | `Disagreement` parameters | Value | RND parameters | Value |
|---|---|---|---|---|---|---|---|
| $N_{ensemble}$ | 20 | $N_{dqn}$ | $[64, 64]$ | $N_{dqn}$ | $[64, 64]$ | $N_{dqn}$ | $[64, 64]$ |
| $N_{dbmrbpi}$ | $[64, 64]$ | $C$ | $10^5$ | $C$ | $10^5$ | $C$ | $10^5$ |
| $\tau$ | $10^{-2}$ | $\tau$ | $10^{-2}$ | $\tau$ | $10^{-2}$ | $\tau$ | $10^{-2}$ |
| $\alpha_Q, \alpha_M$ | $(5 \cdot 10^{-4}, 5 \cdot 10^{-5})$ | $\alpha$ | $5 \cdot 10^{-4}$ | $\alpha$ | $5 \cdot 10^{-4}$ | $\alpha$ | $5 \cdot 10^{-4}$ |
| $p_{uniform}$ | 0.5 | $N_{rep}$ | 512 | $N_{ensemble}$ | 20 | $N_{rep}$ | 512 |
| $p_{mask}$ | 0.5 | $N_{trunk}$ | $[512]$ | $N_{disag}$ | $[64]$ | $N_{rnd}$ | $[64, 64]$ |
| $k$ | 1 | $N_F$ | $[64]$ | $\alpha_{disag}$ | $10^{-4}$ | $\alpha_{rnd}$ | $10^{-4}$ |
| $s_{prior}$ | 8 | $N_B$ | $[64]$ | | | | |
| $C$ | $10^5$ | $\alpha_{apt}$ | $10^{-4}$ | | | | |
| | | $K_k$ | 12 | | | | |

Table 9: Parameters and values of the algorithms. For all algorithms we used a batch size of 128.

## D.3 Radio network control problems

### D.3.1 Downlink link adaptation problem

In this section we describe how the `DBMR-BPI` can be applied to the Downlink link adaptation problem [37] in radio networks.

**Overview.** Link adaptation, or adaptive modulation and coding (AMC), is a technique used in wireless communication systems, such as the 3GPP HSDPA, LTE or NG-RAN, to dynamically adapt the transmission rate of a communication link to the time- and frequency-varying channel conditions [37]. Modulation and Coding Schemes (MCS) effectively adapt the transmission rate by matching the modulation and coding parameters used for communication (such as modulation order, code rate, etc.) to the conditions of the radio link, such as propagation loss, channel strength, interference from other signals concurrently transmitted in the same radio resources, etc. Link adaptation is a dynamic process that acts potentially as frequently as each transmission time interval (e.g., on a sub-millisecond time-scale in the 3GPP NG-RAN system) wherein a communication is scheduled between a base station (a radio node that facilitates wireless communication for user equipment within its coverage area) and a user equipment (UE, a mobile device such as a smartphone or tablet).

Link adaptation algorithms attempt to optimally adapt the transmission data rate chosen for a link to the current channel and interference conditions of the link. These algorithms select the most appropriate modulation order and coding rate (also known as modulation and coding scheme, MCS) based on the latest information available about the state of the communication system as well as the state of the individual communication link (e.g., Channel Quality Indicator, CQI) for which the MCS value is selected.

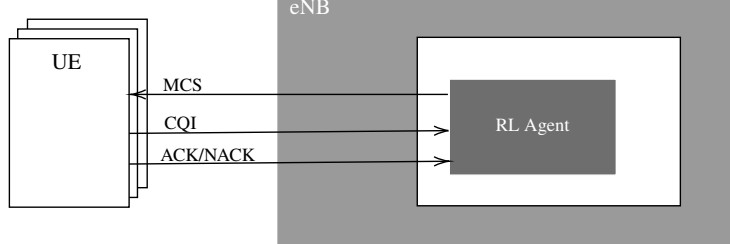

Figure 21: Downlink link adaptation diagram.

**Problem Objective.** In wireless communication systems, there is a fundamental trade-off between throughput and spectral efficiency, which can be understood in terms of resource elements. Resource elements are the smallest units of time-frequency resources allocated for transmission, encompassing a specific duration of time and a specific frequency bandwidth. Our objective is to explore the environment using `DBMR-BPI` to design policies that optimize this trade-off based on the desired intent.

Higher throughput, the amount of data transmitted successfully per unit of time, often requires higher-order modulation schemes and denser coding rates, which can reduce spectral efficiency—the effective use of available spectral resources. Conversely, enhancing spectral efficiency through lower-order modulation schemes and robust coding rates can ensure more reliable transmissions but may limit throughput.

Since measuring throughput directly is challenging due to various influencing factors, we consider Transport Block Size (TBS) as a proxy. TBS represents the amount of data that can be transmitted in a given time interval based on the selected Modulation and Coding Scheme (MCS). However, increasing TBS does not necessarily result in higher throughput, as a larger TBS can lead to a higher Block Error Rate (BLER)—the percentage of transport blocks that fail to be decoded correctly—impacting the reliability of data transmission and reducing the actual amount of successfully received data.

**Environment, RL formulation and setup.** We consider a Time-Division Duplexing (TDD) 5G system operating at a 3.5GHz carrier frequency, with PHY layer numerology $\mu = 0$ of the 3GPP technical specifications 38.211 and single-user Multi-Input Multi-Output (SU-MIMO) transmission. Each base station is configured as massive MIMO (mMIMO) with an 8x4x2 antenna array. We consider 1 site, 3 sectors, and a fixed number of 10 UEs with full buffer traffic, thereby experiencing stable interference conditions. We generated 150 randomized scenarios during training that randomize across several variables (e.g., UEs movement and traffic). Each scenario lasted 5 seconds.

The downlink link adaptation problem can be formulated as a multiple-rewards RL problem as follows:

- The base station represents the RL agent that, depending on the channel measurements transmitted by the UEs, chooses the modulation and coding scheme (MCS).
  - The state includes current CQI values, number of the antennas of the UE, estimated speed of the UE, rank indicators, and other relevant metrics that affect link quality, including a windowed history of these values.
  - Actions involve selecting different MCS values, which dictate the modulation order and coding rate for the packet transmission.
- The transmission of a packet from a UE is modeled as an episodic MDP of at most 5 steps. At each time step $t$, the agent chooses an action $a_t$ (the MCS). If the transmission is successful, the agent receives an ACK and the episode terminates. Otherwise, the agent receives a NACK and observes the new channel measurements (next state) until the episode is done.

The environment was simulated using a proprietary simulator[10], the code for which cannot be disclosed. We evaluated the `DBMR-BPI` algorithm along with `RND`, `APT` and `Disagreement`.

**Reward design.** We consider two rewards, that optimize for different intents. For each episode we let $z_t$ be a boolean value indicating if the packet was successfully transmitted (ACK) or not (NACK). We also let $TBS_n$ indicate the transport block size for the $n$-th transmission, and $N_{re,n}$ the number of resource elements for the $n$-th transmission.

1. Transport block size: $r_{1,n} = TBS_n \cdot \mathbf{1}_{z_n=1}$, this reward encourages the agent to choose actions that maximize the transport block size (at the expense of other metrics, i.e., error rate, spectral efficiency), measured as the number of bits of successfully transmitted packets.
2. Spectral efficiency: $r_{2,n} = 0$ if the transition is not terminal; otherwise, $r_{2,n} = -\sum_{k \leq n} N_{re,k}$, representing the total number of resource elements used up to the final time step $n$ of the episode.

These rewards, if weighted together, can optimize for different intents, between throughput, efficiency and transmission rate.

**Numerical results: performance.** In fig. 22 we report the performance of the various algorithms in terms of (1) throughput, (2) BLER (block error rate) and (3) spectral efficiency (which measures the number of resource elements used).

---

[10]The environment design and MDP formulation come, or are inspired, from a forthcoming publication [101].

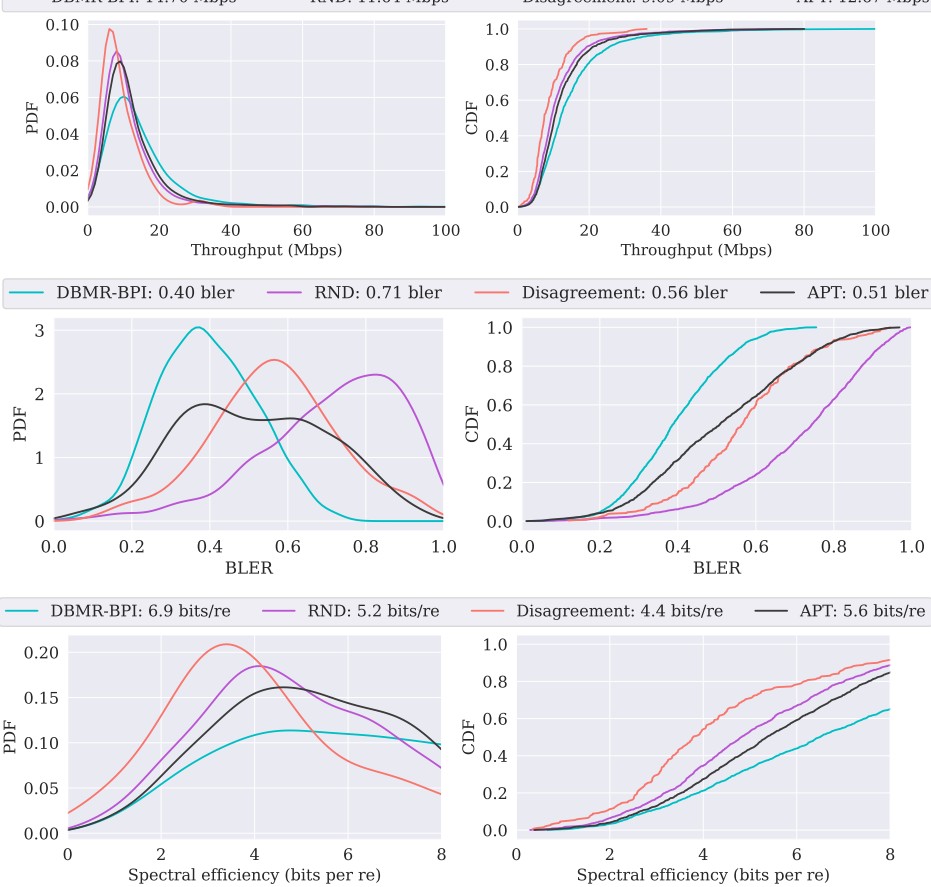

Figure 22: From top to bottom: (1) throughput distribution achieved during training; (2) BLER distribution during training; (3) distribution of the number of bits transmitted per resource elements (spectral efficiency) during training. In the legend we also write the average value for each algorithm.

In fig. 23 we present the distribution of the number of transmissions attempted per packet and the action selection (MCS index) for each algorithm.

We observe that `DBMR-BPI` achieves better performance both in terms of throughput and spectral efficiency. The MCS index distribution reveals that `DBMR-BPI` tends to use all the available actions during the first transmission (note that only the last 5 index values are used during re-transmissions), which seems to indicate that it is effectively learning which action to take depending on the state features. On the other hand `RND` and `Disagreement` tend only to focus on the initial MCS index values during the first transmission.

We also note a lower BLER target for `DBMR-BPI`, as evidenced also by the fact that the average number of transmissions is lower for `DBMR-BPI` compared to the other methods.

In general, these performances appear to be primarily due to optimized usage of resource elements, as shown in fig. 24. This figure illustrates the distribution of rewards collected during training for each episode. While the TBS is similar across all algorithms, `DBMR-BPI` clearly focuses on optimizing the number of resource elements.

This type of analysis is valuable for practitioners to understand the properties and impacts of each reward function.

**Numerical results: rewards chosen by `DBMR-BPI`.** In fig. 25 we plot the estimate $\Delta_{t,r}$ for both rewards (left plot) and the number of times `DBMR-BPI` selected each reward (right plot; does not account for forced tracking or uniform exploration). From the left plot we see that optimizing

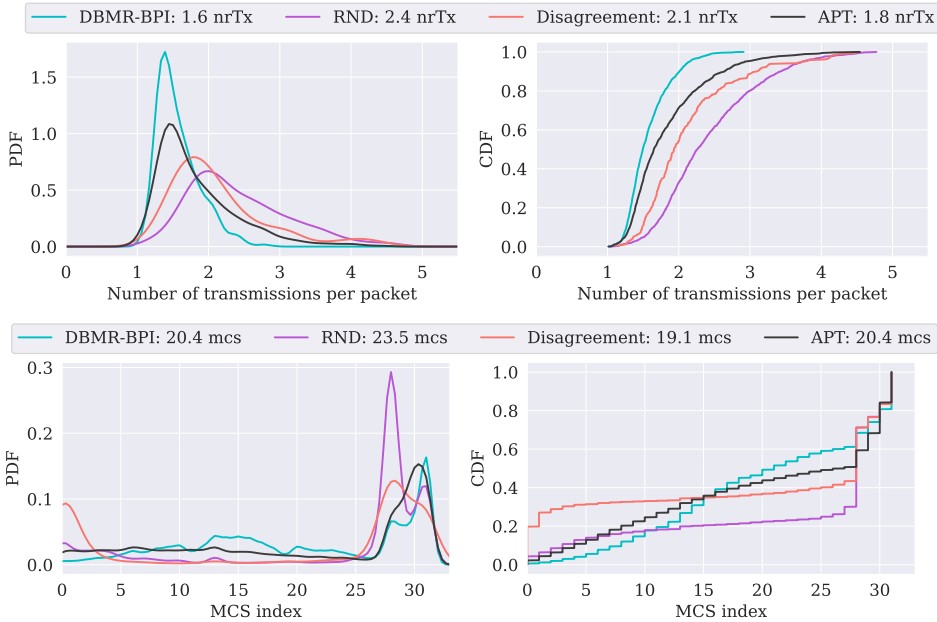

Figure 23: From top to bottom: (1) distribution of the number of re-transmissions per packet during training; (2) MCS index selected by the algorithms during training (note that only the last 5 values are used during re-transmissions). In the legend we also write the average value for each algorithm.

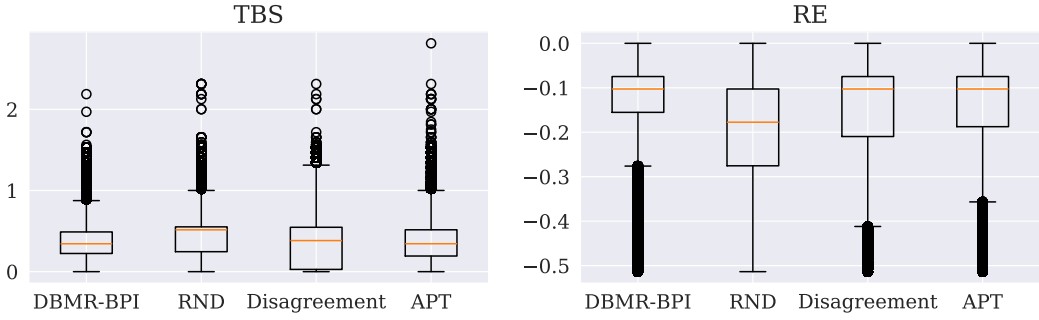

Figure 24: Distribution of the normalized rewards collected during training by the algorithms.

the number of resource elements seems to be a harder task due to the lower value of $\Delta_{t,r}$, which explains why `DBMR-BPI` focuses more on optimizing the number of resource elements used per packet transmission.

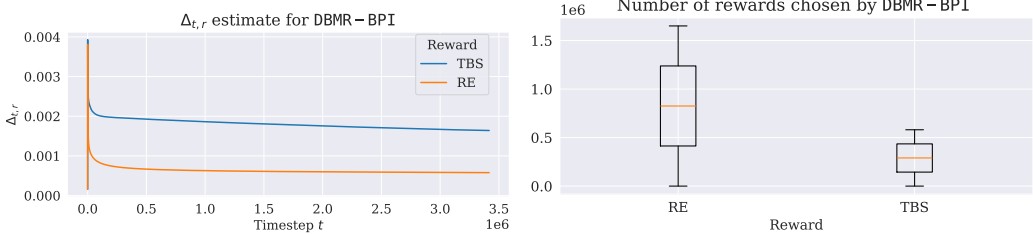

Figure 25: Left plot: value of $\Delta_{t,r}$ estimated by `DBMR-BPI` during training for both rewards. Right plot: distribution of the rewards chosen by `DBMR-BPI` during training.

**Parameters, compute resources and libraries.** For these simulations we used 4 private workstation each with 24 vCPUs, 128 GiB of memory and 1 A100 NVIDIA GPU. To obtain all the results 6 days of computation time are needed. The entire research project needed roughly 24 days of computation time for this experiment.

The parameters are listed in tab. 10. Refer to app. E for further details on the parameters and the algorithms.

| DBMR-BPI parameters | Value | APT parameters | Value | Disagreement parameters | Value | RND parameters | Value |
|---|---|---|---|---|---|---|---|
| $N_{ensemble}$ | 20 | $N_{dqn}$ | $[128,]*7$ | $N_{dqn}$ | $[128,]*7$ | $N_{dqn}$ | $[128,]*7$ |
| $N_{dbmrbpi}$ | $[256,]*6$ | $C$ | $7 \cdot 10^6$ | $C$ | $7 \cdot 10^6$ | $C$ | $7 \cdot 10^6$ |
| $\tau$ | $10^{-3}$ | $\tau$ | $10^{-3}$ | $\tau$ | $10^{-3}$ | $\tau$ | $10^{-3}$ |
| $\alpha_Q, \alpha_M$ | $(5 \cdot 10^{-5}, 5 \cdot 10^{-5})$ | $\alpha$ | $5 \cdot 10^{-5}$ | $\alpha$ | $5 \cdot 10^{-5}$ | $\alpha$ | $5 \cdot 10^{-4}$ |
| $p_{uniform}$ | 0.5 | $N_{rep}$ | 512 | $N_{ensemble}$ | 20 | $N_{rep}$ | 512 |
| $p_{mask}$ | 0.5 | $N_{trunk}$ | $[256, 256]$ | $N_{disag}$ | $[256, 256, 128, 128, 128]$ | $N_{rnd}$ | $[256, 256, 256, 128, 128, 128]$ |
| $k$ | 1 | $N_F$ | $[256, 128, 128, 128]$ | $\alpha_{disag}$ | $5 \cdot 10^{-5}$ | $\alpha_{rnd}$ | $5 \cdot 10^{-5}$ |
| $s_{prior}$ | 10 | $N_B$ | $[256, 128, 128, 128]$ | | | | |
| $C$ | $7 \cdot 10^6$ | $\alpha_{apt}$ | $5 \cdot 10^{-5}$ | | | | |
| | | $K_k$ | 15 | | | | |

Table 10: Parameters and values of the algorithms. For all algorithms the batch size was $512$. The notation $[x,] * k$ indicates a stack of $k$ layers of size $x$.

We set up our experiments using Python 3.10 [88] (For more information, please refer to the following link http://www.python.org), and made use of the following libraries: NumPy [89], SciPy [90], PyTorch [98], Seaborn [92], Pandas [93], Matplotlib [94].

# E  Deep-RL Algorithms Details

## E.1  `DBMR-BPI` algorithm

**Overview.**   As mentioned in sec. 4, extending `MR-NaS` to Deep-RL is not straightforward. Firstly `MR-NaS` is a model-based method, secondly there is no closed-form solution to $\arg\inf_{\omega \in \Omega} U^\star(\omega; M)$. To circumvent these issues, we adapt the DBMF-BPI algorithm from [53], where the authors propose `DBMF-BPI`, a deep-RL exploration approach for single-reward BPI. In [53], the authors suggest that a model-free solution can be achieved by (i) using the *generative solution* $\arg\inf_{\omega \in \Delta(S \times A)} U^\star(\omega; M)$, which can be computed in closed form, and (ii) by estimating the parametric uncertainty of the MDP-specific quantities (sub-optimality gaps, etc.). In other words, the generative solution may be close enough to the model-based one. Nevertheless, this solution may be biased because we do not know what are the true sub-optimality gaps, and therefore we need to take into account the parametric uncertainty that we have in the estimates.

Hence, we propose `DBMR-BPI`, an extension of `DBMF-BPI`, which learns an ensemble of $Q$-functions of the optimal policy and, similarly, the $k$-th moment of that $Q$-function (called $M$-function). More precisely, the extension of `DBMF-BPI` to a finite set of rewards, mainly consists of few modifications that we describe in the following paragraphs.

**1.  Value learning for different rewards.**   The $Q$-function in `DBMR-BPI` (sim. the $M$-function) learns the value for multiple rewards. In the finite action setting this implies that there are $AR$ values to estimate in every state. Alternatively, we also tried to feed the networks with a one-hot encoding of the chosen reward, but this resulted to be less effective in terms of model capacity.

**2. Reward exploration.**   We explore according to the difficulty of the reward function. For the sake of stability, at the beginning of each episode (or, for example, every $\approx 1/(1-\gamma)$ steps) we sample a reward $r \sim \mathcal{R}$ to explore, and apply `DBMF-BPI` on this reward. To estimate the exploration difficulty, we simply use the estimate at time $t$ of the minimum sub-optimality gap $\Delta_{t,r}$ (see paragraph below), and explore a reward $r$ with probability $\propto 1/\Delta_{t,r}^2$.

In addition to the mechanism described above, we also introduce a random uniform exploration of the rewards and forced tracking of the rewards:

- *Random exploration*: with probability $p_{uniform}$ we choose a reward uniformly at random.

- *Forced tracking*: this mechanis ensures that each reward is explored sufficiently often. We let $U_t = \{r \in \mathcal{R} : \sqrt{t} - N_{r,t}/2 < 0\}$ be the set of under-sampled rewards at time $t$, where $N_{r,t}$ is the number of times the algorithm has chosen reward $r$ in the previous $t$ steps, and choose $r = \arg\min_r N_{r,t}$ whenever $U_t$ is non-empty (otherwise, we sample it as in Alg. 2). This procedure guarantees that each reward is sampled at a rate of $O(\sqrt{t})$ [20].

---

**Algorithm 3** `DBMR-BPI` - Reward selection

1: **if** $X \le p_{uniform}$ **then**
2:     Choose a reward uniformly at random.
3: **else if** $U_t$ is not empty **then**
4:     Choose reward $\arg\min_r N_{r,t}$.
5: **else**
6:     Choose reward $r$ with probability $\propto \frac{1}{\Delta_{t,r}^2}$
7: **end if**

---

**3. Estimation of the sub-optimality gaps $\Delta_{t,r}$.**   To estimate the minimum sub-optimality gap for a fixed reward $r$, we initially used the method described in `DBMF-BPI`. However, their estimates are often overly conservative. In `DBMF-BPI`, the minimum gap for each batch across all models in the ensemble is calculated and added to a rolling window. The smallest value in this window is then used as the target for stochastic approximation, employing a decreasing learning rate.

We instead compute $\Delta_{t,r}$ as follows:

1. For each reward in the batch, we calculate the gaps and take the minimum across the models in the ensemble. We then average these minimum gaps over the transitions in the batch to obtain an estimate $\Delta_{t,r}$. This approach is less conservative than taking the minimum across both transitions and models in the ensemble.

2. We treat $\Delta_{t,r}$ as a learnable parameter and use the estimated gap $\hat{\Delta}_{t,r}$ as the target value. We compute the Huber loss between $\Delta_{t,r}$ and $\hat{\Delta}_{t,r}$, and perform a gradient step to update $\Delta_{t,r}$.

3. To further stabilize the estimate of $\Delta_{t,r}$, we can apply Polyak averaging over time steps.

The main parameters for `DBMR-BPI` are as follows:

| Symbol | Description |
|--------|-------------|
| $N_{ensemble}$ | Ensemble size |
| $N_{dbmrbpi}$ | Hidden layers sizes for each model in the ensemble |
| $\tau$ | Rate at which the target network updates |
| $\alpha_Q, \alpha_M$ | Learning rates of Adam for the $Q,M$-networks |
| $p_{uniform}$ | Uniform random reward probability |
| $p_{mask}$ | Mask probability used by `DBMF-BPI` |
| $k$ | $k$-value used by `DBMF-BPI` |
| $s_{prior}$ | prior scale value used by `DBMF-BPI` |
| $C$ | Maximum size of the replay buffer |

Table 11: `DBMR-BPI` parameters.

## E.2 Other algorithms

For `APT`, `Disagreement` and `RND` we used the open-source implementation from [24]. However, the implementation in [24] only works for continuous action spaces, therefore we adapted the implementation to discrete action spaces as described below. For all the algorithms we used the same architecture as in the original papers, and only did a slight tuning of the hyper-parameters.

`DQN` **agent.** Each algorithm uses a `Double DQN` agent to learn the $Q$-values using the intrinsic reward computed by the corresponding algorithm. The main parameters for this `DQN` agent are

| Symbol | Description |
|--------|-------------|
| $N_{dqn}$ | Hidden layers sizes |
| $C$ | Maximum size of the replay buffer |
| $\tau$ | Rate at which the target network updates |
| $\alpha$ | Learning rate of Adam |

Table 12: `DQN` parameters.

`APT` **Agent.** This agent uses a forward network $F_\theta$ and a backward network $B_\theta$ to learn a representation that is used to compute the intrinsic reward. The forward network predicts a representation of the next state $\hat{s}_{t+1}$ given the current state $s_t$ and action $a_t$, while the backward network predicts the action that was taken $\hat{a}_t$ given two consecutive state representations $s_t, s_{t+1}$.

The representation is $N_{rep}$-dimensional, and is learnt by a trunk network $M_\theta$.

The loss is computed by averaging over a batch $\mathcal{B}$ the forward and backward errors. The forward error is computed as in [24]

$$e_{forward}(\mathcal{B}) = \frac{1}{|\mathcal{B}|} \sum_{(s_t,a_t,s_{t+1})\in\mathcal{B}} \|F_\theta(M_\theta(s_t))_{a_t} - M_\theta(s_{t+1})\|_2,$$

where $F_\theta(\cdot)_a$ is the $a$-th predicted representation.

We adapt the backward error to discrete action spaces by using the cross-entropy loss

$$e_{backward}(\mathcal{B}) = -\frac{1}{|\mathcal{B}|} \sum_{(s_t,a_t,s_{t+1})\in\mathcal{B}} \sum_a \mathbf{1}_{(a_t=a)} \log(B_\theta(M_\theta(s_t), M_\theta(s_{t+1}))_a),$$

where $B_\theta(\cdot, \cdot)_a$ is the predicted probability of action $a$ by the network. The overall loss over a batch $\mathcal{B}$ is $e(\mathcal{B}) = e_{forward}(\mathcal{B}) + e_{backward}(\mathcal{B})$.

For this agent, in addition to the parameters in tab. 12, we also have the following parameters

| Symbol | Description |
|---|---|
| $N_{rep}$ | Representation dimension |
| $N_{trunk}$ | Hidden layers sizes for the trunk network |
| $N_F$ | Hidden layers sizes for the forward network |
| $N_B$ | Hidden layers sizes for the backward network |
| $\alpha_{apt}$ | Learning rate for APT |
| $K_k$ | Number of neighbors used to compute the particle based entropy |

Table 13: APT parameters.

Disagreement **agent.** This agent uses an ensemble of models to predict the next state. The variance across the models is then used to quantify the parametric uncertainty and compute the intrinsic reward. We kept the same implementation as in [24]. For this agent, in addition to the parameters in tab. 12, we also have the following parameters

| Symbol | Description |
|---|---|
| $N_{ensemble}$ | Ensemble size |
| $N_{disag}$ | Hidden layers sizes for each model in the ensemble |
| $\alpha_{disag}$ | Learning rate for Disagremeent |

Table 14: Disagreement parameters.

RND **agent.** This agent uses a predictor and target network to estimate the parametric uncertainty, which is then used as intrinsic reward. This is similar to the mechanism employed by prior networks in [102]. Simply, for a given state both the predictor and target networks output a representation of dimension $N_{rep}$. The predictor is then trained to minimize the $\ell_2$ norm distance between the predictor representation and the target representation.

For this agent, in addition to the parameters in tab. 12, we also have the following parameters

| Symbol | Description |
|---|---|
| $N_{rep}$ | Representation dimension |
| $N_{rnd}$ | Hidden layers sizes for the predictor/target networks |
| $\alpha_{rnd}$ | Learning rate for RND |

Table 15: RND parameters.

