# OpenReview forum: "Multi-Reward Best Policy Identification"
_NeurIPS.cc/2024/Conference — NeurIPS 2024 poster_

### Official Review · Reviewer_YFyE · 2024-07-10

**Soundness:** 4
**Presentation:** 2
**Contribution:** 3
**Rating:** 7
**Confidence:** 2

**Summary:**

The present article extends the track-and-stop approach of Garivier et al. to a multi-reward MDP setup. Given an MDP problem with a finite number of reward functions the aim is to develop an algorithm that learns optimal policies for all reward functions simultaneously. Under (drastic) assumptions the authors present a sample based variant of track-and-stop in the multi reward setup. The algorithm is based on replacing the theoretical model complexity $T^*$ (in a multi reward variant) from Garivier et al. by an upper bound $U^*$ that can be estimated. Estimating $U^*$ during the exploration phase results in a practically implementable termination condition that results in worse complexity than the theoretical termination condition. The algorithm is tested on a few (very similar) tabular examples and compared to simple algorithms. An educated guess is performed to design a deep variant of the algorithm that is tested on a multi reward card pole variant and deep sea. Results beat easily the (terrible) results of the benchmark algorithms used.

**Strengths:**

The article is extremely dense with information, perhaps it would have been better to split the article in 2 or 3. The theoretical development is an extension of several earlier articles of the authors with very detailed mathematics. While the tabular examples are of limited practical interest (the assumptions are just too strong) the deep variant is interesting. While I am not so sure about the relevance of the tabular setting, the relevance in the deep setting is obvious (and was realised of the authors for cardpole). The ability of generalisation of NN makes it interesting to train the network simultaneously on perturbations of the true reward function in order to learn a more stable policy that might work well for even more reward settings.

The scientific quality of the article is very high. There is hardly any typo. I appreciate a lot the critic discussion of limitations, this is far above the rather low scientific standard in ML. I could not check the entire proofs of the appendix, what I read seemed solid.

Good article! I am curious to follow up on the future development of the robust DQN variant.

**Weaknesses:**

- I did not enjoy reading the article very much as I was pushed into first reading other articles to get a rough understanding of what is going on. Even reading the basis of the present article ([17], [40]) was not enough. To get an idea why $T^*$ is considered one needs to go all the way back to [20], and even further. It feels a bit like a community wants to stay among each other, the usual RL researcher is excluded by the style of writing. I would suggest to completely skip the discussion in the end (almost a page) and instead use the space to explain the reader what the technique is about and why the theoretical estimate naturally lead to Algorithm 1.
- The assumptions are drastic and should be discussed more clearly. I guess the authors have bandit background and skip discussions of issues that are typical for bandits. In particular, assuming knowledge of rewards and/or the reward gaps. While this is not unusual in bandits it is very much in RL. I am perfectly fine with such theoretical results in particular, as the authors implemented an educated guess algorithm in the deep setting that addresses this issue with the reward gaps.

**Questions:**

Here is a number of questions and comments.

- $\alpha$ should also be an input for the algorithm 1. Why is there no dependence on $\alpha$ in the theoretical results?
- How do you compute U in Algorithm 1 without knowing the rewards gaps? You estimate M, but what about the gaps? They are estimated in the deep setup, why not in the tabular setting?
- I think the algorithms are not sufficiently explained. In the tabular case $M_t$ should be explained in more detail. What shall the reader get from "use estimate $M_t$ of the model", "update $M_t$" without knowing $M_t$? In the present form the article is hardly comprehensible (even for me as a Mathematician). In that generality (what estimate is used?) I am a bit sceptical about the validity of Theorem 3.3, but as said, I could not try to understand all proofs.
- The use of Borel-Cantelli in the proofs is not correct as it is. The events are not independent and this direction of Borel-Cantelli requires (pairwise) independence. I am pretty sure the SARSA paper contains the details on how to use a conditional independence version of Borel-Cantelli.
- I am a bit puzzled about the connection of Theorem 3.1 and 3.3. If I am not wrong, Theorem 3.1 is a bandit theorem, requiring to play independent "arms" $(s,a)$ while Theorem 3.3 (the algorithm) is an RL theorem requiring rollouts $(s_0,a_0, s_1,a_1,...)$. The latter is obviously much harder and depends a lot on the exploration method. Unfortunately, I have six 50pages NeurIPS papers on the table and cannot go fully through every proof. Could you add two sentences why the theorem should hold? For instance, SARSA convergence holds by rewriting SARSA as Q-learning plus an error that goes to zero by the exploration condition. Here, the situation is much harder.
- A crucial point is to replace $T$ by $U$. In line 169 you mention an upper bound for $U$, which has an additional $1/\Delta^2$ compared to the theoretical bound. Is there a better theoretical bound? The authors should at least discuss briefly in the main text that there is something non-trivial going on (as in the SARSA paper where on-policy exploration is essentially as good as off-policy exploration). As it is it one quickly overlooks the difference in the algorithm to the "allocation".
- I cannot see a scaling in the number of rewards. Is there any theoretical understanding on how $\tau$ scales in the number of rewards?
- How does your algorithm compare to just train for one reward after the other? Does that make sense? The training time for card pole seems quite high but I might be wrong. Typically card pole plots show the number of episode (ca. 200-300), not the number of steps. Is it clear your algorithm is faster than just training 5x card pole? How about comparing your algorithm to training 5x card pole with your different reward settings but keeping the neurone network from the reward before (and using some replay buffer to avoid overfitting)? My feeling is that such a naive approach might be better than the benchmarks used.
- Similar question: Comparing your deep algorithm to performing several deep trainings for different rewards. Is your approach particularly suitable to avoid overfitting? Does your algorithm help for generalisation (as you see in the cardpole example?).

**Limitations:**

The tabular setting requires drastic assumption (but gives a clean analysis), the deep setting is an interesting educated guess but does not allow a mathematical analysis. I would say that is quite normal for RL, except, perhaps, the deviation of the educated guess Algorithm 2  from the tabular algorithm 1 is pretty big.

There is no good benchmark to compare to, this makes the numerical results a bit boring. Perhaps a better benchmark would be to compare to training individually the policies. While that might even beat the proposed algorithm in the tabular case (for few rewards) I doubt the same holds for the deep case as the NN would only remember to solve the last reward problem.

---

> ### Author Rebuttal · Authors · 2024-08-06
>
> We thank the reviewer for their thoughtful review and the time spent on our paper. Below, we address each concern in detail.
>
> >I was pushed into first reading other articles to get a rough understanding of what is going on
>
> We acknowledge  the current format assumes familiarity with related literature. We will add a background section to make the technique accessible to a broader audience.
>
> > How do you compute $U$ in Algorithm 1 without knowing the rewards gaps?
>
> We briefly explain the procedure in lines 178-182 and we do not assume knowledge of the gaps in the algorithm.
>
> We use the current estimate of the model $M_t$ and the set of rewards to compute the sub-optimality gaps. We use the data gathered up to time $t$ to estimate the transition function $P_t$ and perform value iteration for each reward to find the sub-optimality gaps. These estimates are used in place of the true gaps to compute $U$.
>
> This technique is known as "certainty-equivalence" principle, meaning that we use the current estimates as if they were exact when computing $\omega_t^\star$. Note that, as more data is gathered, the estimate of the transition function improves, and $\omega_t^\star$ tends to $\omega^\star$.
>
> > I think the algorithms are not sufficiently explained. In the tabular case $M_t$ should be explained in more detail
>
> Currently, the main text provides a brief description: lines 178-182 outline how $M_t$ is updated by estimating the transition function $P_t$, and lines 183-189 explain how to compute the instance-dependent quantities to determine $U$.
>
> In the revised manuscript we will provide a more detailed explanation of the algorithm.
>
> >  Why is there no dependence on $\alpha$ in the theoretical results?
>
> The main theoretical results are asymptotic in $\delta$, causing $\alpha$ dependence to vanish. Intermediate results do show $\alpha$ dependency (e.g., see Remark C.2, line 1099).
>
> > The use of Borel-Cantelli in the proofs is not correct as it is. The events are not independent
>
> We rely on Observation 1 in [70] (in the text of the proof there is a typo, since there is no Lemma 4 in [70]). We use the  fact that there exists another stochastic process (the one provided by the forcing policy) such that the sum of the probabilities with which action $a$ is chosen is infinite. This sum lower bounds the sequence of probabilities of selecting action $a$ and satisfies the independence conditions required by the Borel-Cantelli Lemma.
>
> > What is the connection of Theorem 3.1 and 3.3?
>
> Both theorems are for MDPs. Deriving Theorem 3.1 requires additional steps compared to the bandit setting, and the optimal allocation $\omega_{opt}$ must satisfy the forward Chapman Kolmogorov equation.
>
> To explain how the two theorems are connected, we consider an information-theoretical approach.
>
> Firstly, note that the optimal exploration strategy induces the stationary distribution $\omega_{opt} =\arg\sup_{\omega \in \Omega(M)} \inf_{r,M'\in Alt(M_r)} \mathbb{E}\_{(s,a)\sim\omega}[{\rm KL}\_{M|M'}(s,a)]$, and $T^\star$ can be interpreted as the average amount of information extracted per time-step from the MDP at stationarity under $\omega_{opt}$.
>
> Secondly, as $T^\star$ is a hard non-convex optimization problem, we instead use $U^\star$ in the design of our algorithm.
>
> As more data is collected, the estimate of the model $M_t$ tends to $M$, and one can show that $\omega_t^\star$ approaches the optimal stationary distribution $\omega^\star$. Asymptotically, the algorithm will visit the MDP according to $\omega^\star$, so the amount of information extracted approaches $U^\star$. Hence, with a proper stopping rule, this is roughly  why, as $\delta \to 0$, the sample complexity of the algorithm approaches the  value $U^\star$ in Theorem 3.3.
>
> > You mention an upper bound for $U$, which has an additional $1/\Delta^2$ compared to the theoretical bound. Is there a better  bound?
>
> Please note that also $U$ has a dependency on $1/\Delta_r^2$. In general, we believe that it is hard to find a better dependency in the gaps that does not depend on some specific property of the MDP.
>
> >Is there any theoretical understanding on how $\tau$ scales in the number of rewards?
>
> The result in Theorem 3.1 indicates that the scaling depends on the "worst" reward in the considered set. Practically, according to $U^\star$, the sample complexity is dictated by the  minimum sub-optimality gap $1/(\inf_{r\in {\cal R}} \Delta_r^2)$ (and $\tau$ will scale as $U^\star \ln(1/\delta)$). This scaling is consistent with the scaling obtained in classical reward-free exploration, see  for example [17; A]. In [A], the authors show a similar result in a different setting, where the sample complexity also scales according to the worst reward (with a similar $1/\Delta^2$ scaling, though their gap definition is slightly different).
>
> > How does your algorithm compare to just train for one reward after the other? Typically card pole plots show the number of episode, not the number of steps. Does your algorithm help for generalisation?
>
> The optimization problem formulated to compute $\omega^\star$ shows that the scaling depends on the worst-case reward in the set (i.e., the reward with the minimum sub-optimality gap) rather than the number of rewards. Consequently, a sequential exploration process would generally not be optimal (since that would scale according to the number of rewards). Such an approach would likely increase training time and associated costs.
>
> Regarding the Cartpole swing-up experiments, each episode may have a different number of steps. Further note that we use a modified version of the Cartpole swing-up problem,  designed to assess the exploration properties of RL algorithms.
>
> Lastly, with a sequential training the ability to generalize may be limited, especially for very different rewards, reducing the effectiveness of such an approach.
>
> **Other ref.**
>
> [A]. Jedra et al. "Nearly optimal latent state decoding in block mdps." AISTATS 2023.

---

> > ### Comment · Reviewer_YFyE · 2024-08-09
> > **Thanks for the answers - I will keep my 7.**
> >
> > Thanks for your answers! I believe the article is really good and I would even consider an 8 if the article would be much better understandable for more than a handful of experts in the field. In your own interest in spreading your ideas to a wider group of researchers I would suggest to really use the revision for more explanations.

---

> > > ### Author Response · Authors · 2024-08-10
> > >
> > > Thank you again for your support! We completely agree with you. In the revision, we will make sure to provide clearer explanations of the background and main ideas behind this family of techniques.

---

### Official Review · Reviewer_N1yn · 2024-07-12

**Soundness:** 2
**Presentation:** 2
**Contribution:** 2
**Rating:** 4
**Confidence:** 3

**Summary:**

This paper studies the problem of best policy identification for RL with multiple rewards. The goal is to efficiently identify the best policy for given rewards with a high-level confidence. Authors provide an instance-dependent lower bound for the studied problem and introduce a provably-correct algorithm for a convex approximation of the original problem in the tabular setting. Extensions to deep RL is discussed with numerical results.

**Strengths:**

The authors demonstrate how to efficiently identify optimal policies across multiple rewards. The strengths of this work are summarized as follows:

1. The studied setting is interesting and of practical concern when we seek to optimize the performance across a set of rewards.

2. An instance-dependent lower bound is identified for any Probably-Correct algorithm in the setting of multi-reward best policy identification problem.

**Weaknesses:**

While this paper clearly articulates the idea of performing, here are some problems and potential improvements that authors are suggested to address and consider. More specifically,

1. Environments for the deep RL part is too simple. The Cartpole swing-up and DeepSea environments cannot fully demonstrate the performance of the proposed algorithm in more complex, real-world scenarios. It would be beneficial to include experiments on more challenging benchmark environments for better assessment of scalability and practical applicability.

2. The policy optimality is only considered and defined for stationary deterministic policy (as in definition 2.1), which can be too restrictive. It is not clear when considering the set of Markovian policies (which can be stochastic), whether the proposed lower bound still holds, and whether the performance of the algorithm is still optimal.

3. Theoretical guarantees for the deep RL extension is unavailable. Sample complexity bounds are only provided for tabular settings, which leads to the dependencies on the cardinality of state and action space. And empirical studies for the deep RL settings are not sufficiently convincing due to the simplicity of the environments.

4. In terms of the theoretical results of the lower bound, the proof structure closely follows the prior results (e.g. Marjani et al. 2021, Taupin et al. 2022) in single-reward RL. It is not quite obvious what are the main technical challenges and novelties of extending the prior results (e.g. Marjani et al. 2021) from single-reward RL to multiple-reward RL.

5. While the studied setting can be interesting, the relationship between the MR-BPI problem and reward-free RL as well as multi-objective RL somewhat remains unclear throughout the context. Though discussion has been provided in Section 1 and 5, I am not fully convinced that reward-free RL cannot solve the concerned practical scenario in multi-reward best policy identification problem. Indeed, reward-free RL assumes rewards are unknown, whereas the studied settings assume the knowledge of rewards. As a result, it is not surprising to see that properly utilizing the knowledge of rewards can lead to better performance as shown in the numerical results: the proposed algorithm (MR-NaS) significantly outperforms RF exploration strategies (RF-UCRL, ID3AL). However, reward-free RL is a more general type of algorithms and can be particularly useful in practice when it is hard to accurately learn rewards or when rewards are sparse.  As such, the emphasis of the two settings are rather different. It might not be a fair comparison, and it is desirable to provide the fundamental reasons that can explain such performance improvement in numerical experiments. Therefore, more thoughtful insights should be provided to clearly explain the difference and relationship between these settings.

6. Some minor aspects:
    - Grammatical errors need to be addressed, e.g. Line 354, line 91.
    - In line 75-80, if only deterministic policies are been considered, it is more appropriate to write $a_t = \pi(\cdot|s_t)$ etc. Do not use the probability simplex notation for policies.

**Questions:**

1. Do we treat MR-BPI as dedicated exploration strategies for multi-objective RL?

2. Most reward-free RL focuses on episodic settings, whereas this paper studies discounted settings for multi-reward MDPs. Is there any particular reason for choosing this (simpler) setting? Do you foresee any technical challenges that can be difficult to resolve in episodic multi-reward settings?

3. In Line 81 - 86 (Set of rewards), for each reward vector, each coordinate $i$ represents the reward for the $i$-th state-action pair, which is a scalar, why do we need the canonical basis $\mathcal{R_{canonical}}$ of rewards, where each element is a vector? (When you write $\mathbb{R}^{SA}$, I assume each element is a real number, not a vector). Do you assume for each $(s, a)$, there is a different reward function? Could you provide a concrete example of your definition of "set of rewards" with your notation? I assume you intended to say there exists $m$ (global) reward functions, and this set of functions $\mathcal{R}$ is thus in $\mathbb{R}^{m \times SA}$.

**Limitations:**

There are no unaddressed ethical considerations in the studied context.

---

> ### Author Rebuttal · Authors · 2024-08-05
>
> We thank the reviewer for their feedback and for the time and effort spent reviewing our paper. Below we provide detailed responses to the main concerns raised by the reviewer.
>
> > Simple environments for deep RL
>
> We appreciate the reviewer's concern. However, the selected environments are intentionally designed as hard exploration challenges, as referenced in [36,40,43]
> Additionally, we  included  experiments for a real-world application in appendix D.3 (a radio network control problem). These  experiments show the effectiveness of our algorithm  in a realistic environment using an industry-level network simulator.
>
> > Optimality is only considered and defined for stationary deterministic policy
>
> For finite MDPs, the optimal policy is stationary and deterministic (see Theorems 6.2.9 and 6.2.10 in [39]). Therefore, in this context, the lower bound holds, and the performance of our algorithm is asymptotically optimal  with respect to $U^\star$.
>
> > Theoretical guarantees for the deep RL extension
>
> We acknowledge the absence of theoretical guarantees for the deep RL extension.
> However,  our primary goal was to establish a foundational understanding of the multi-reward setting for finite MDPs, which is crucial for building towards more complex cases.
>
> Providing theoretical guarantees for deep RL  would merit its own manuscript.
>
> > Main technical challenges and novelties of extending the prior results  from single-reward RL to multiple-reward RL.
>
> For the lower bound, as in prior work, the proof builds upon classical change of measure arguments. The main difference with respect to the single-reward setting lies in the additional optimization over the set of rewards.
>
> However, there are several differences compared to prior work, highlighted throughout the paper. For example:
>
> 1. Concerning the relaxed characteristic rate, deriving a tractable  optimization problem for $U^\star$ in presence of a continuous set of rewards is challenging, since the minimum sub-optimality gap can be non-convex (see Sec. B.4 of the appendix). To address this issue, we show that it is sufficient to consider a finite set of rewards. This finding is  used in our numerical results (see Sec. D.1.2 in the appendix or Sec. 3.4 in the main text).
>
> 2.  Regarding the algorithm, we remove the need for an MDP-specific constant in the forced exploration (see line 195), which  limited the usability of NaS [17]. We further provide a novel high-probability forced exploration result (see also remark C.2, page 35).
>
> 3. Regarding the extension to Deep-RL,   prior works in BPI mostly consider the tabular case. Furthemore, the extension of [40] to the multi-reward setting is non-trivial. First, by inspecting $U^\star$,  we design an exploration procedure that explores according to the difficulty of the rewards. Secondly, we experimented with various architectural changes of the neural  networks to accommodate the multi-reward setting.
>
>
> > I am not fully convinced that reward-free RL cannot solve the concerned  scenario  [...] reward-free RL is more general  and can be useful when it is hard to accurately learn rewards or when rewards are sparse.
>
> As correctly noted by the reviewer, considering a set of known rewards can provide superior performance with respect to the more general reward-free RL setting and this is the main motivation behind our setting.
>
> Determining the best policy for any possible reward  is a harder problem that is unlikely to occur in some practical problems. Hence, for applications where there are only a finite number of rewards to optimize, the multi-reward setting would provide a better model choice w.r.t. the reward-free setting. We provide a concrete example of this scenario in Appendix D.3.
>
> Regarding the sparsity of the reward, our  method can also handle scenarios with sparse rewards, as demonstrated in our numerical results.
>
> Lastly, while the comparison may not be completely fair, there are currently no other pure exploration algorithms in the multi-reward setting (and reward-free RL is one of the closest to our setting). For this reason, we compare to different reward-free methods, as well as an adaptation to the multi-reward setting of PSRL [35] that we provide in the appendix.
>
> > Do we treat MR-BPI as dedicated exploration strategies for multi-objective RL?
>
> The multi-reward setting, as well as reward-free RL, share some similarities to multi-objective RL. In multi-objective RL, one seeks to identify the Pareto frontier of non-dominated policies. This objective is, in general, a significantly harder challenge compared to identifying the optimal policies for a finite set of rewards.
>
> >  Is there any particular reason for choosing the discounted setting instead of the episodic one?
>
> While  most reward-free approaches focus on episodic settings, in practice, many algorithms used in industry employ the discounted setting. Our goal was to strike a balance between providing theoretical results and developing a practical algorithm that practitioners can readily apply.
>
> We believe the discounted setting can be extended to episodic, leveraging recent advances in [A].
>
> > Why do we need the canonical basis of rewards? Could you provide a concrete example  of ”set of rewards” with your notation?
>
> Each element of the canonical basis is a vector of size $SA$. For example, $e_1$ represents a sparse reward where $r(s_1,a_1)=1$ and $0$ otherwise, expressed in vector form as $e_1 = \begin{bmatrix} 1 & 0 & \dots & 0 \end{bmatrix}^\top$. Similarly, $e_2$ is a reward vector that assigns a reward of $1$ to the second state-action pair and $0$ otherwise.
>
> In some experiments we explored according to this canonical basis of rewards (as previously explained, see also Sec B.4 in the appendix).  For a concrete example, we also refer the reviewer  to section B.4.1 of the appendix.
>
> **Additional references**
>
> A. Al-Marjani et al. "Towards instance-optimality in online pac reinforcement learning." arXiv:2311.05638 (2023).

---

> > ### Comment · Reviewer_N1yn · 2024-08-13
> >
> > I thank the authors for their detailed response.
> >
> > As pointed out the authors, the studied multi-reward setting, reward-free RL and multi-objective RL does share similarities with different emphasis respectively. Hence, it would be important to provide a rigorous comparison between all these settings to clearly credit their own strengths and articulate the differences. It is essential to acknowledge such difference especially when it is not a fair comparison, so as to avoid misunderstanding for future studies. Effort should be made to provide a better understanding.
> >
> > As such, I prefer maintaining my current score while not championing for a reject. I do think results of the multi-reward setting could be interesting, but more careful revision would be extremetly beneficial.

---

> ### Author Response · Authors · 2024-08-13
>
> We appreciate the reviewer's feedback and understanding.
>
> However, we have already acknowledged the differences between our work and these other settings (see lines 33-42 and the discussion in Section 5), and we are committed to further clarify the distinction.
>
> We have already extensively compared our approach with reward-free RL in both the tabular and deep settings (both in the main text and in the appendix), as it is the closest existing framework to ours in terms of exploration objective.
>
> Regarding multi-objective RL,  we remark that our focus is on a different exploration objective, since we are not interested in identifying the Pareto frontier of non-dominated policies. Currently, there are  no best policy identification techniques available for multi-objective RL, making a direct comparison impractical (which would merit a separate, dedicated study).
>
> We hope these clarifications address the reviewer's concerns and help in understanding the scope and contributions of our work.

---

### Official Review · Reviewer_1UCp · 2024-07-29

**Soundness:** 3
**Presentation:** 4
**Contribution:** 2
**Rating:** 6
**Confidence:** 3

**Summary:**

The paper addresses the challenge of identifying the best policy in RL when there are multiple rewards. The authors get a lower bound on the sample complexity and design an optimal exploration policy. The authors propose two algorithms: MR-NaS for tabular environments and DBMR-BPI for Deep RL. These algorithms efficiently identify the best policy across multiple rewards in RL.

**Strengths:**

The paper presents a comprehensive and well-balanced analysis of theoretical and empirical results. The appendix provides supplementary evidence that strengthens the authors' arguments.

**Weaknesses:**

1. The paper  is inspired by [17], but it would be better to explicitly acknowledge this inspiration in the main part, rather than mention it at Remark C.2. Furthermore, a more in-depth discussion of the challenges in proof caused by the novel forcing policy would strengthen the paper's contribution.

2. Even though more details is covered in the appendix, the paper should provide some details on the convex optimization. For example, a discussion of the computational costs associated with these methods would provide valuable context for readers. Sometimes, the computational cost of convex optimization methods are high.


3. The abstract would be better if providing a more impressive motivation for multi-reward RL, emphasizing its significance and potential impact.

4. Didn’t put some innovative aspects in the main text, leaving vital details to be discovered in the appendix. This can lead to important contributions being overlooked.

**Questions:**

Please address the issues mentioned in the weaknesses.

**Limitations:**

As noted in the paper, the assumption of a unique optimal policy limits the method's applicability to a broader range of scenarios. The computational cost of convex optimization may prevent it to deal with large-scale dataset.

---

> ### Author Rebuttal · Authors · 2024-08-05
>
> Thank you for your comments and valuable feedback. We appreciate the time and effort spent reviewing our paper, as well as your positive comments on the comprehensive analysis of theoretical and empirical results.
>
> Below, we address each of your concerns and outline the corresponding revisions we plan to make to improve the paper.
>
> > The paper is inspired by [17], but it would be better to explicitly acknowledge this inspiration in the main part [...] a more in-depth discussion of the challenges in proof caused by the novel forcing policy would strengthen the paper’s contribution.
>
> We would like to clarify that we acknowledge this inspiration in multiple points, including at the beginning of Sec. 3 (line 118), Sec. 3.1 (lines 133-134), and Sec. 3.3 (line 172). To further highlight this point we plan to include this sentence in the introduction:
> "Our work is inspired by [17] for single-reward BPI and extends the results and analysis to the multi-reward setting".
>
> Regarding the challenges in the proof, we agree with the reviewer that expanding on these in the main text would enhance the paper and highlight an important contribution. We have already briefly described how our approach removes the need for an MDP-specific constant in the forced exploration (discussed at line 195), which previously limited the usability of NaS [17]. Due to lack of space, more details on the novelty of the forcing policy were left in the appendix. We will make sure to include these in the main part of the paper by using the extra page in the final version of the paper. More precisely, we will further enhance our discussion by highlighting the challenges addressed in our proof and emphasizing our improved result for the high-probability forced exploration theorem, which we believe is of independent interest.
>
> > Even though more details is covered in the appendix, the paper should provide some details on
> the convex optimization. For example, a discussion of the computational costs [...]. The computational cost of convex optimization may prevent it to deal with large-scale dataset.
>
> We agree with the reviewer that the paper should provide more details on the convex optimization problem defining $U^\star$.
> We plan to include a subsection discussing the technical aspects of solving the  optimization problem in the paper (note that currently in  appendix D we provide the total runtime of our simulations), including implementation details and further results on the computational cost.
>
> Lastly, for Deep-RL, our algorithm DBMR-BPI has similar computational complexity as  DBMF-BPI [40], or Boostrapped DQN [43] at inference and training time (due to  a  vectorized implementation of DBMR-BPI).
>
> > The abstract would be better if providing a more impressive motivation for multi-reward RL, emphasizing its significance and potential impact.
>
> We appreciate the suggestion and we agree that the abstract could better emphasize the motivation for multi-reward RL. In the introduction, we already highlight the practical importance and potential impact of Multi-Reward RL and its relevance with respect to existing similar settings. For instance, reward-free RL might not always applicable in industry. In some applications, there are only a finite number of rewards to optimize.
>
> To illustrate this aspect concretely,  in Section D.3 of the appendix, we present an example of applying DBMR-BPI to a radio network control problem using an industry-level network simulator, demonstrating how current reward-free exploration techniques are limited when the number of rewards is finite.
>
> We will make sure to highlight these aspects in the abstract and include these consideration in the final version of the paper.
>
> > Didn’t put some innovative aspects in the main text, leaving vital details to be discovered in the appendix.
>
> We agree with the reviewer that, due to the limitation in space, some important aspects were left to the appendix. We will integrate the main innovative aspects into a dedicated section by using the extra page in the final version of the paper. This section will include a summary of the theoretical and practical contributions, as well as their implications.
>
> > The assumption of a unique optimal policy limits the method’s applicability to a broader range of scenarios.
>
> While we agree that the uniqueness assumption may be limiting for a continuous set of rewards, we believe that it is less restrictive when the set of rewards is finite, as it is generally easier to guarantee the uniqueness of an optimal policy in such cases.
>
> Furthermore, note that, as discussed in Appendix A (Limitations section), the assumption on the uniqueness of the optimal policy is common in the BPI literature [18, 16, 17, 40, 20]. Addressing MDPs with multiple optimal policies or identifying $\varepsilon$-optimal policies  necessitates the use of more complex overlapping hypothesis tests [67]. We refer the reader to this Appendix for more details and further consideration on this extension.
>
> Lastly, while the theoretical results may require this assumption, in practice we experienced both MR-NaS and DBMR-BPI to perform well in the numerical experiments even in presence of multiple optimal rewards.

---

> > ### Comment · Reviewer_1UCp · 2024-08-13
> >
> > Thank you for your rebuttal, I've revised my score to 6.

---

> > > ### Author Response · Authors · 2024-08-13
> > >
> > > Thank you for your time and feedback! We are glad our response was able to clarify the concerns you raised, and we appreciate you taking the time to reconsider and update the score accordingly.

---

### Author Rebuttal · Authors · 2024-08-06

We would like to thank the reviewers for their comprehensive  reviews.  We are pleased that you recognize the scientific quality and the rigor of our theoretical and empirical contributions.

Our work strives to bring a comprehensive and well-balanced analysis, bridging both theoretical analysis and practical methods.  Our work provides not only fundamental theoretical results in the tabular setting but also practical extensions to deep reinforcement learning (DeepRL).
The numerical experiments for both the tabular and DeepRL cases are conducted on  environments designed to be challenging for exploration, highlighting the effectiveness of our approach across different settings.

Lastly, we present a real-world application in Section D.3 of the appendix, where we apply DBMR-BPI to a radio network control problem using an industry-level network simulator. This example emphasizes the relevance and applicability of our contributions.

We hope our responses clarify and address the reviewers' concerns, and we are committed to revising the paper based on their feedback in our final submission.

---

### Decision · Program_Chairs · 2024-09-25

**Decision:**

Accept (poster)

**Comment:**

This paper presents a comprehensive study on the Multi-Reward Best Policy Identification (MR-BPI) problem in Reinforcement Learning (RL), where the objective is to identify the best policy for a set of rewards with minimal sample complexity and high confidence. The authors provide an instance-specific lower bound on the sample complexity required by any Probably Correct (PC) algorithm and introduce a convex approximation to design efficient, sample-optimal algorithms. The proposed algorithms, MR-NaS for tabular settings and DBMR-BPI for Deep RL, are evaluated on both synthetic and real-world tasks, showing competitive performance, particularly in hard-exploration environments.

Pros & Cons:
+ A significant theoretical advancement by deriving an instance-specific lower bound and extending classical RL concepts to the multi-reward setting.
+ Algorithm design is supported by both theory and experiments.

- Some reviewers expressed concerns that the environments used for the deep RL experiments, such as Cartpole swing-up and DeepSea, maybe too simplistic.
- It may need more clarification for readers to understand the dense details.

Summary:
The majority of reviewers appreciate the depth and rigor of the paper’s contributions, particularly in advancing theoretical understanding and proposing novel algorithms in the multi-reward RL setting. However, concerns about the clarity of the presentation and the complexity of the experimental validation were raised. Overall, the AC recommends acceptance of the paper.